*Resource*

# The global phosphorylation landscape of mouse oocytes during meiotic maturation

Hongzheng Sun[1,4], Longsen Han[1,4], Yueshuai Guo[1,4], Huiqing An[1,4], Bing Wang[1,4], Xiangzheng Zhang[1], Jiashuo Li[1], Yingtong Jiang[1], Yue Wang[1], Guangyi Sun[1], Shuai Zhu[1], Shoubin Tang[1], Juan Ge[1], Minjian Chen [1], Xuejiang Guo [1,2]✉ & Qiang Wang [1,3]✉

## Abstract

**Phosphorylation is a key post-translational modification regulating protein function and biological outcomes. However, the phosphorylation dynamics orchestrating mammalian oocyte development remains poorly understood. In the present study, we apply high-resolution mass spectrometry-based phosphoproteomics to obtain the first global in vivo quantification of mouse oocyte phosphorylation. Of more than 8000 phosphosites, 75% significantly oscillate and 64% exhibit marked upregulation during meiotic maturation, indicative of the dominant regulatory role. Moreover, we identify numerous novel phosphosites on oocyte proteins and a few highly conserved phosphosites in oocytes from different species. Through functional perturbations, we demonstrate that phosphorylation status of specific sites participates in modulating critical events including metabolism, translation, and RNA processing during meiosis. Finally, we combine inhibitor screening and enzyme-substrate network prediction to discover previously unexplored kinases and phosphatases that are essential for oocyte maturation. In sum, our data define landscape of the oocyte phosphoproteome, enabling in-depth mechanistic insights into developmental control of germ cells.**

**Keywords** Oocyte; Meiosis; Phosphorylation; Kinase; Phosphatase
**Subject Categories** Development; Post-translational Modifications & Proteolysis; Proteomics

## Introduction

During oogenesis, changes in protein phosphorylation occur (Bornslaeger et al, 1986; Bornslaeger et al, 1988; Schultz et al, 1983), participating in diverse biological events. Misregulation of substrate phosphorylation induced by kinase mutation could result in impaired oocyte quality and female infertility (Sang et al, 2018;

Zhang et al, 2021). Well-balanced phosphorylation and dephosphorylation are therefore necessary for producing developmentally competent eggs (Hormanseder et al, 2013; Neant et al, 1989; Zhang et al, 2023). Despite these links, temporal characterization of phosphorylation dynamics during mammalian oocyte maturation has been challenging due to the technical limitations and material scarcity.

Mammalian oocytes, originating from primordial germ cells (PGCs), undergo a process of mitosis where they transition from oogonia to primordial oocytes. At around the time of birth, the oocytes are arrested in prophase of the first meiotic division, commonly referred to as the germinal vesicle (GV) stage. Upon the stimulation of hormonal surge, immature oocytes resume meiosis characterized by the breakdown of germinal vesicle (GVBD). Concomitant with chromatin condensation and microtubule organization, oocytes progress through meiosis I (MI) division, eventually arresting at metaphase II (MII) stage until fertilization ensues. The protein phosphorylation in oocytes was investigated as early as the 1970s (Morrill and Murphy, 1972; Wallace, 1974). Employing *Xenopus laevis* as a model, researchers observed a surge in protein phosphorylation during meiotic resumption and the extensive dephosphorylation after fertilization. Following that, a succession of seminal discoveries concerning mammalian oocyte phosphorylation has emerged, primarily using kinase inhibitors and genetically engineered animal models. For instance, maturation-promoting factor (MPF) is a heterodimer composed of a catalytic CDK1 and the Cyclin B regulatory subunit (Han and Conti, 2006; Levasseur et al, 2019). Mice with oocyte-specific deletion of CDK1 demonstrated an inadequate ability to drive the resumption of meiosis (Adhikari et al, 2012; Kalous et al, 2006). Inhibition of mitogen-activated protein kinase cascade (MAPK) has been shown to disrupt meiotic apparatus and oocyte maturation (Howard et al, 1999; Posada and Cooper, 1992). ERK1/2 directly phosphorylates and activates RNA binding protein-CPEB1, facilitating the poly(A) extension and thereby translational activation of *Ccnb1* in mouse oocytes (Cao et al, 2020; Sha et al, 2017). Kinase mutations are found to be associated with the occurrence of infertility in women, such as homozygous mutations in *Wee2* and biallelic mutations in *Mos* (Sang et al, 2018; Zhang et al, 2021). In

[1]State Key Laboratory of Reproductive Medicine and Offspring Health, Changzhou Maternity and Child Health Care Hospital, Changzhou Medical Center, Nanjing Medical University, 211166 Nanjing, China. [2]Department of Histology and Embryology, Nanjing Medical University, 211166 Nanjing, China. [3]Center for Global Health, School of Public Health, Nanjing Medical University, 211166 Nanjing, China. [4]These authors contributed equally: Hongzheng Sun, Longsen Han, Yueshuai Guo, Huiqing An, Bing Wang. ✉E-mail: guo_xuejiang@njmu.edu.cn; qwang2012@njmu.edu.cn

addition, the phosphorylation of non-kinase proteins also involves in the modulation of oogenesis. For example, histone H3 phosphorylation participates in chromosome condensation and anaphase onset during meiosis (Wang et al, 2016). Phosphorylation-mediated activation of PDE3A and CDC25B significantly contributes to the initiation of meiotic resumption (Han et al, 2006; Pirino et al, 2009). While the existing studies identified numerous phosphorylation sites on individual proteins, we believe that a global search for phosphorylation patterns could lead to more complete understanding of oocyte development.

In the present study, we applied quantitative MS-based phosphoproteomics to characterize mouse oocyte phosphorylation dynamics during in vivo maturation by isolating a substantial number of cells at three key stages (GV, GVBD and MII). In combination with in-depth proteomics, we obtained the most comprehensive stage-resolved phosphoproteome and proteome landscape of mammalian oocytes to date. Meanwhile, we revealed the novel molecular mechanisms controlling meiotic maturation by experimentally manipulating the specific phosphosites in oocytes. In conclusion, our data represent a valuable resource for exploring protein phosphorylation in germ cells, enabling the further mechanistic insights into developmental control.

# Results

## In-depth phosphoproteomics profiling of mouse oocyte maturation

While the significance of protein phosphorylation during oocyte development has been implicated, a comprehensive investigation of phosphoproteome dynamics during in vivo oocyte maturation is still lacking. This is largely attributed to the limited amounts of experimental materials. Here, we isolated a substantial quantity of mouse oocytes at three pivotal time points during meiotic maturation (arrested GV stage, meiotic resumption GVBD stage, and ovulated MII oocyte; 10,000 oocytes for each stage with 5 replicates), and obtained the intracellular phosphoproteome utilizing multiplex tandem mass tag (TMT) labelling coupled with liquid chromatography–mass spectrometry (LC–MS) (Figs. 1A, EV1A). Quality control of the phosphoproteome and proteome data was firstly performed. The results showed good separation of each stage samples, high reproducibility between biological replicates, as well as a wide quantitative range of phosphorylation levels (Fig. EV1B–EV1I). In total, we achieved high-quality quantification of 8090 distinct phosphosites (Class I, >0.75 localization probability) located on more than 2600 proteins (Fig. 1B; Dataset EV1). More than 30% of oocyte proteins (2268 phosphoproteins in total 6700 proteins) are estimated to be phosphorylated (Fig. EV1J,EV1K; Dataset EV2). We next examined the distribution patterns of identified phosphosites and observed mostly Ser phosphorylation(86.5%, pS), followed by Thr (13.0%, pT) and Tyr (0.5%, pY) (Fig. 1B). The phosphotyrosine abundance was relatively diminished in oocytes compared to mouse organs (0.5% vs. 2.5%) (Giansanti et al, 2022; Huttlin et al, 2010), indicating that the proportion of tyrosine phosphorylation in germ cells may be lower than that in somatic cells.

It has been well recognized that the active sites on MAPK1 (T183/Y185), MAPK3 (T203/Y205), and CDK1 (T161) undergo phosphorylation during oocyte meiosis, while the phosphorylation of two inhibitory residues (T14/Y15) on CDK1 is decreased after GVBD (Adhikari and Liu, 2014; Lemonnier et al, 2020; Tong et al, 2003). Our phosphoproteomic results completely confirmed these findings (Fig. 1C,D), indictive of the dataset reliability. Quantitative analysis of phosphosites showed that 58% (1508/2604) of phosphorylated proteins identified in this study contained two or more phosphosites, and about 10% (250/2604) proteins contained more than 6 phosphosites, implying that multi-phosphorylation is ubiquitous in meiotic oocytes (Fig. 1E). These phosphosites are predominantly enriched for proteins related to autophagy signaling, mTOR pathway, and RNA processing etc (Fig. 1F). Together, the results provide a broad resource of phosphoproteome landscape during oocyte maturation. Below we use the phosphoproteomics data as an entree to uncover the novel phosphorylation features and to elaborate on possible consequences of the phosphorylation changes for oocyte development.

## Phosphorylation dynamics during oocyte maturation

While the phosphorylation changes in certain proteins during oocyte maturation have been reported, the precise dynamics of alterations in the total phosphoproteome remain unknown. Herein, we found that more than 75% of phosphosites, on nearly 85% of phosphoproteins, were significantly regulated during maturation (Fig. 2A,B; Dataset EV3). An increase was observed in the number of upregulated phosphosites and phosphoproteins throughout the meiotic process, with the majority of regulation occurring at the phosphorylation level (Fig. EV2A,EV2B), whereas changes in protein abundance were less pronounced (Fig. EV2C,EV2D). Moreover, only 295 proteins overlapped between regulated phosphoproteome ($n = 2204$) and altered proteome ($n = 929$) (Fig. EV2E–EV2G; Dataset EV4). The broader distribution of the magnitude of changes to the phosphosites was observed in comparison to the proteome (GVBD/GV and MII/GV; Fig. 2C,D). These findings unveil that phosphorylation modification serves as an additional layer of molecular control for oocyte maturation, complementing the proteome landscape we discovered previously (Sun et al, 2023). To further characterize phosphoproteome fluctuations in meiotic oocytes, the regulated phosphosites were grouped in five clusters based on their dynamics using fuzzy c-means algorithm (Fig. 2E; Dataset EV5). Phosphorylation of cluster 1 sites ($n = 385$) showed a steady decrease during maturation, primarily enriching in histone ubiquitination (i.e. CUL4B-T106, HUWE-T1905) and protein dephosphorylation (i.e. CDC25B-S319, DUSP7-S220) (Fig. 2F). Sites within cluster 2 ($n = 566$) displayed the elevated phosphorylation upon meiotic resumption followed by a dramatic downregulation (Fig. 2G), specifically involving in nuclear division (i.e. CEP192-T674, CENPE-S2414) and nuclear pore organization (i.e. NUP98-T553, NUP133-S44), both critical for germinal vesicle breakdown. In contrast, phosphosites in clusters 3, 4, and 5 ($n = 5179$) were enriched in cell cycle (i.e. CHEK1-S301. BRCA1-T1199), cytoskeleton organization (i.e. MYO5A-S600, KIF23-S125), and mRNA processing (i.e. PABPC1L-S109, SRSF2-S189) (Fig. 2H–J), exhibiting marked upregulation in ovulated oocytes. In addition, the 2604 phosphoproteins identified in oocytes were found on different subunits of the majority of protein complexes (63 out of 107; Appendix Fig. S1A; Dataset EV6), highlighting the molecule

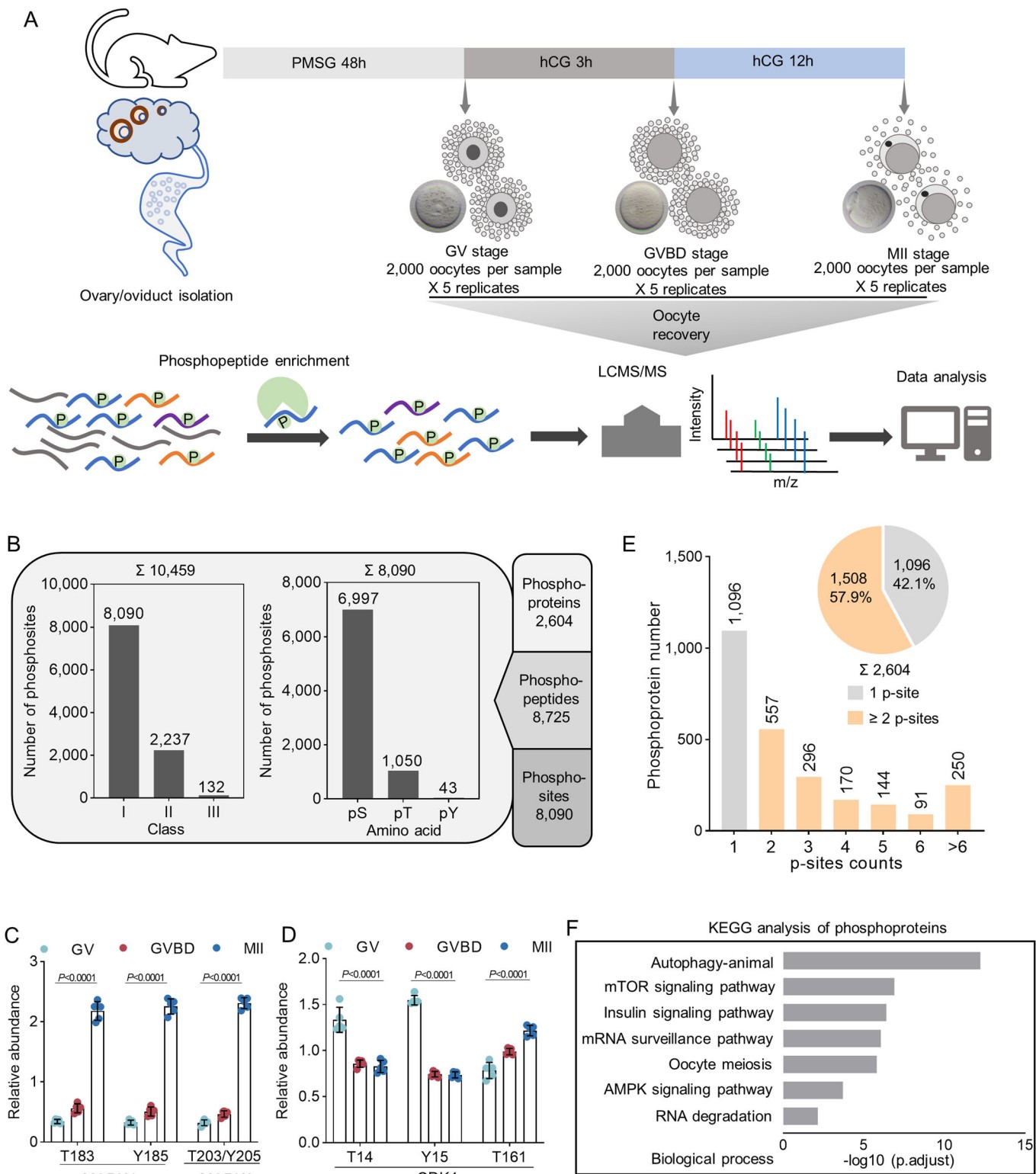

cascades related to phosphorylation signal. For example, the significant changes in phosphorylation were detected in components of MAPK signaling pathway (B-Ksr1-MEK-MAPK-14-3-3 complex), histone modifications (Ikaros complex), as well as

anaphase-promoting complex (Appendix Fig. S1B,S1C). Collectively, protein phosphorylation in oocytes is highly dynamic during maturation, independent of protein abundance, and a crucial layer of developmental control.

**Figure 1. Phosphoproteomics profiling of mouse oocyte maturation.**

(A) Illustration of in vivo isolation of mouse oocytes at GV, GVBD, and MII stage. (B) Number of identified phosphoproteins, phosphopeptides, and phosphosites in all measured samples. (Left) Number of phosphorylated residues from different classes according to localization probability: class I (probability > 75%), class II (probability = 50 to 75%), and class III (probability <50%). (Middle) Distribution of phosphorylated amino acids [serine (pS), threonine (pT), and tyrosine (pY)]. (C, D) Phosphorylation dynamics of key residues on CDK1, MAPK1, and MAPK3 during oocyte maturation. Data are expressed as mean percentage ±SD from five independent replicates. Two-tailed Student's *t* test was used for statistical analysis, comparing to GV oocytes. The *P* value is labeled in the figure. (E) Bar graph and pie chart depicting the number of phosphosites per protein. (F) Bar graph showing the statistically enriched KEGG pathways in the phosphoproteome. Benjamini–Hochberg (BH) corrected *P* value adjustment (*P*.adjust) was used for the enrichment analyses. Source data are available online for this figure.

## Identification of novel phosphosites in mouse oocytes

Oocytes play a unique role in female reproduction. Identification of oocyte-specific phosphorylation sites would be helpful for understanding its specialized developmental program. Notably, compared with Gygi's mouse multi-tissue phosphoproteomes (Villen and Gygi, 2008) and PhosphoSitePlus database (Hornbeck et al, 2019), we found that more than 22% (1842 out of 8090) of phosphosites on 931 proteins in oocytes were undetected in other tissues or cells (Fig. 3A,B; Dataset EV7). Of them, we observed a plethora of epigenetic regulators (i.e. DNA methylation modifiers and histone modification enzymes) harboring novel phosphosites (Fig. 3C–F; Appendix Fig. S2A–S2D). In specific, a novel phosphosite on TET3 (S1522; Fig. 3C) was identified within low-complexity domain that is essential for protecting oocyte genome from oxidative demethylation (Zhang et al, 2024). DNMT1, UHRF1, and DPPA3 have been implicated in the establishment and maintenance of DNA methylation during oogenesis (Li et al, 2018). Our data revealed several new phosphosites in their functional domains, such as S311/356/S882 on DNMT1 (Fig. 3D), S18/501/502 on UHRF1 (Fig. 3E), as well as T29 and S124 on DPPA3. In addition, we also discovered numerous phosphosites that have not been reported before on histone methylation erasers (KDM6B-S973, Appendix Fig. S2A) and writers (KMT2B-S352, KMT2C-S679/S4059, and KMT2D-S652/S655; Appendix Fig. S2B–S2D). The above observations indicate that these novel phosphosites are likely to meet the unique regulatory requirements for epigenetic control in oocyte development.

Many phosphosites were also identified for the first time in oocyte-specific expressed proteins (Fig. 3G–J), such as S86/92 and T176 on NOBOX, S257/183/144 and T149 on ZAR1L, T44 and S40/47 on OBOX2, and S276/279 on OBOX5. In particular, we noticed that the components within the subcortical maternal complex (SCMC) generally experience the multi-site phosphorylation, exemplified by NLRP5, which features 24 distinct phosphosites (Fig. 3K; Dataset EV7). The dynamic phosphorylation changes in these specific sites on maternal effectors may act as key signaling mechanisms, orchestrating downstream molecular events essential for oogenesis and embryonic development. In addition to the previously identified phosphosites (i.e., YBX2-T67/78) (Medvedev et al, 2008), substantial novel sites of phosphorylation were detected (i.e., YBX2-S203/206/351, CUL4B-S155/147, CSNK1A1-T321 and SRPK1-S37; Fig. 3L–O; Appendix Fig. S2). The function of these oocyte-specific phosphosites deserves in-depth exploration.

### BTG4 phosphorylation is required for maternal mRNA degradation

BTG4, a maternal effector, has been demonstrated to be responsible for the degradation of oocyte mRNA (Sha et al, 2020; Yu et al, 2016). Here, we identified three consecutive novel phosphosites on BTG4 (T145, S146 and S147), showing the progressive increase in phosphorylation during oocyte maturation (Fig. 4A). These three sites are predicted to be conserved across multiple species and are located within an intrinsically disordered region (IDR) known for its susceptibility to phosphorylation (Iakoucheva et al, 2004) (Fig. 4B,C; Appendix Fig. S3A). To determine the functional involvement of phosphorylation, BTG4 mutants were constructed in which the T145, S146 and S147 residues (*Btg4*^wt) were changed to alanine (*Btg4* triple mutations, *Btg4*^tm) via site-directed mutagenesis (Fig. 4D; Appendix Fig. S3B,S3C). Next, transcriptome sequencing (RNA-seq) and poly(A) tail assay (PAT) were conducted to assess the transcript changes in oocytes when phosphomutant was expressed (Fig. 4E). Polyadenylation levels of two representative maternal transcripts (*Zp3* and *Gtsf1*) were determined using PAT assay (Fig. 4F). Consistent with previous report (Liu et al, 2016), poly(A) tails were shortened in control maturing oocytes, but, remarkably, this process was blocked in *Btg4*^tm oocytes (Fig. 4G–I), and the corresponding transcripts were thus resistant to degradation (Fig. 4J,K). In line with this observation, RNA-seq analysis revealed the significantly upregulated transcripts in *Btg4*^tm oocytes compared to that in *Btg4*^wt oocytes (Dataset EV8). Importantly, 2461 out of the 2802 (87.8%) increased transcripts in *Btg4*^tm oocytes were those being degraded in normal MII oocytes (Fig. 4L; Dataset EV9). Furthermore, more than 50% of these upregulated transcripts were overlapped between *Btg4*^tm and *Btg4*^ko oocytes (Fig. 4M; Dataset EV10). These data strongly suggest that the phosphomutant compromises the ability of BTG4 for mRNA degradation. In addition, it has been reported that BTG4 functions through interacting with PANPN1L, and such an interaction was mainly mediated by the C-terminal domain of BTG4 (Appendix Fig. S3D) (Zhao et al, 2020). Our prediction based on structural model indicated that phosphorylation status of these three sites on BTG4 is likely to alter the surface charge of its C-terminal domain (Appendix Fig. S3E). Co-immunoprecipitation assay further corroborated that phospho-deficient mutant of BTG4 partially disrupts its interaction with PABPN1L (Appendix Fig. S3F). Cumulatively, our data identified three previously unknown phosphosites on BTG4 that crucial for its function during oocyte maturation.

## Discovery of conserved phosphosites in oocytes across multiple species

Phosphoproteomic profiling of oocytes from non-mammalian model organisms (*Drosophila*, *Sea star*, and *Xenopus*) has been established (Presler et al, 2017; Swartz et al, 2021; Zhang et al, 2019). Here, to discover the conserved phosphosites in oocytes across different species, we carried out a comparative analysis between these published datasets and our results via a two-step strategy (Fig. 5A). First, we selected the phosphorylated peptides (13 amino acid residues flanking the phosphosite) exhibiting a sequence similarity greater than 69% (9/13 amino acid) compared with mouse oocyte data in at least two other

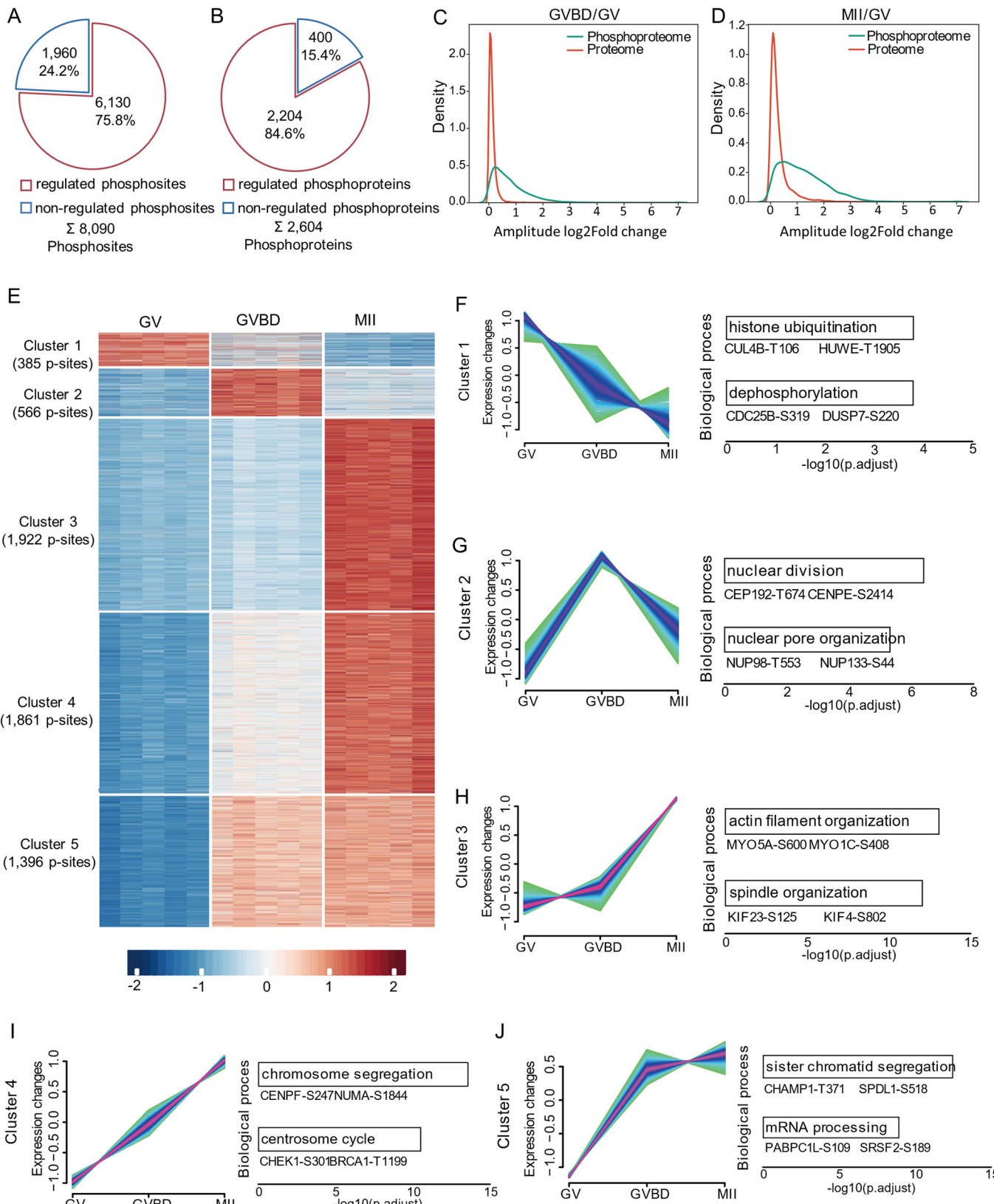

**Figure 2. Phosphorylation dynamic during meiotic maturation.**

(A) Pie chart showing the percentage of regulated phosphosites during oocyte maturation. (B) Pie chart showing the percentage of regulated phosphoproteins during oocyte maturation. (C, D) Density plots comparing the amplitudes of the regulated phosphoproteome and the corresponding proteome. (E) Heatmap illustrating the dynamic changes in regulated phosphosites during oocyte maturation. Fuzzy c-means clustering organized the phosphosites into five distinct clusters. The numbers of phosphosites corresponding to each cluster are indicated. (F–J) Five distinct clusters of regulated phosphoproteome. Each line indicates the relative abundance of individual phosphosites. Representative biological processes and phosphosites are show on the right. Benjamini–Hochberg (BH) corrected *P* value adjustment (*P*.adjust) was used for the enrichment analyses.

species. Next, proteins containing these peptides were individually examined for their functional conservation (see details in "Methods" section). Finally, a total of 76 highly conserved phosphosites on 56 proteins were identified in oocytes (Fig. 5B; Appendix Fig. S4A–S4C; Datasets EV11–EV15). On one hand, we found the phosphosites (MAPK1-T183, ARPP19-S62, and CDK1-T14/Y15) (Appendix Fig. S4D–S4F) that has been well recognized in different species and demonstrated to be important for oocyte maturation (Adhikari et al, 2016; Hached et al, 2019; Liang et al, 2007). On the other hand, we also discovered some fully conserved phosphosites across four species, which have never been reported in oocytes (i.e., HDAC2-S422/S424, NUP98-S888, PDCD4-S457) (Fig. 5C–E). These findings suggest such an evolutionarily conserved phosphorylation mechanism may be involved in the critical events in oocytes. GO analysis further revealed that oocyte proteins with conserved phosphosites were mainly enriched in cell cycle, metabolism, translation and kinase activity (Fig. 5F; Appendix Fig. S4G). For example, many conserved phosphosites were found to locate within the active residues of kinases (Appendix Fig. S4H–S4J), indicative of the importance in enzyme regulation. In the following study, we determined whether these conserved phosphosites are functional in mouse oocytes. Among them, MDH1-S241 (Fig. 5G) and RPL12-S38 (Fig. EV3D) are of great interest because of their high conservation and potential involvement in metabolic and translational control, respectively.

### Phosphorylation of Ser241 on MDH1 is essential for amino acid metabolism in oocytes

Well-balanced metabolism is essential for producing a high-quality oocyte. Many conserved phosphorylation events occurring on metabolic enzymes become particularly noteworthy given the substantial remodeling of the metabolic framework during oocyte maturation (Li et al, 2020; Yang et al, 2010). MDH1 is a highly conserved protein with roles in malate aspartate shuttle (Fig. 5G,H; Appendix Fig. S5A,S5B). Moreover, Ser241 is located within the crucial functional domains of MDH1, including NAD binding domain, malate binding domain and dimer interface (Appendix Fig. S5C,S5D). To investigate the role of MDH1 phosphorylation, a mutation by changing Ser241 to Ala (*Mdh1*mut) was introduced (Fig. 5I,J), and then metabolome analysis for both control group (*Mdh1*wt) and phosphomutant group (*Mdh1*mut) was conducted (Fig. 5K; Dataset EV16). Robust orthogonal partial least-squares-discriminant analysis (OPLS-DA) clearly demonstrated the separation of two groups (Appendix Fig. S6A). A total of 15 differential metabolites was identified following MDH1 mutation (Fig. 5L). As expected, disruption of malate-aspartate shuttle induced a significant oxaloacetate reduction and malate accumulation in *Mdh1*mut oocytes (Appendix Fig. S6B,S6C). More importantly, phosphomutant Ser241 remarkably altered the metabolic flux of aspartate, promoting the synthesis of related amino acids (i.e., threonine, glutamine, guanidoacetic acid, and asparagine)

(Fig. 5M,N; Appendix Fig. S6D–S6G). In sum, the phosphorylation at the conserved S241 residue appears to serve as a regulatory switch of MDH1 activity, participating in the specialized metabolic programming within oocytes.

### RPL12-Ser38 phosphorylation participates in maintaining translational homeostasis in oocytes

The phosphorylation of ribosome protein RPS6 and S6 kinase activity have been extensively examined in oocytes (Adhikari et al, 2009; Adhikari et al, 2010; Reddy et al, 2010). However, whether and how the phosphorylation of other ribosomal proteins functions during oocyte maturation remains to be explored. In the present study, we detected 18 phosphosites on 11 ribosome proteins, belonging to 8 large ribosomal subunits and 3 small ribosomal subunits (Fig. EV3A,EV3B). Of them, a critical factor of ribosome p-stalk protein, RPL12-S38 is highly conserved in oocytes across mouse, *sea star*, and *Xenopus* (Fig. EV3C,EV3D). Furthermore, RPL12-S38 phosphorylation experienced a significant increase during mouse oocyte maturation (Fig. EV3E). To evaluate the potential involvement of RPL12-S38 phosphorylation in translation, we carried out overexpression experiments by microinjecting FLAG-tagged wild-type RPL12 (*Rpl12*wt) and non-phosphorylatable S38A mutant (*Rpl12*mut) mRNA into oocytes, followed by proteomic analysis (Fig. EV3F–EV3H). 197 proteins (a total of 4540 proteins identified) displayed differential accumulation in *Rpl12*mut oocytes compared with *Rpl12*wt oocytes (Fig. EV3I; Dataset EV17). Strikingly, the majority of these proteins (85%, 168/197) was upregulated in oocytes when RPL12-S38 phosphorylation was abolished (Fig. EV3J). Functional annotation indicated that they are primarily involved in cell cycle regulation, protein ubiquitination and organelle organization. Together, these observations suggest that ribosomal protein RPL12 (S38) phosphorylation does not globally impact translation during meiosis, but may regulate the expression of specific subsets of mRNAs.

## Exploring oocyte's kinome

Kinases, as regulators of cellular signaling, play key roles in orchestrating the complex processes of oocyte maturation, fertilization, and embryonic development. The phosphorylation events observed in oocyte phosphoproteome suggest a dynamic interplay between kinases and their substrates. Hence, understanding the intricacies of oocyte phosphorylation necessitates a comprehensive examination of the kinase landscape.

### Oocyte is a kinase-rich repository

To date, a total of 536 kinases has been identified in different tissues and cells based on kinmap database (Eid et al, 2017). It is worth noting that 224 kinases (~40%, 224/536) can be detected in mouse oocytes from proteome and phosphoproteome. Of them, 128

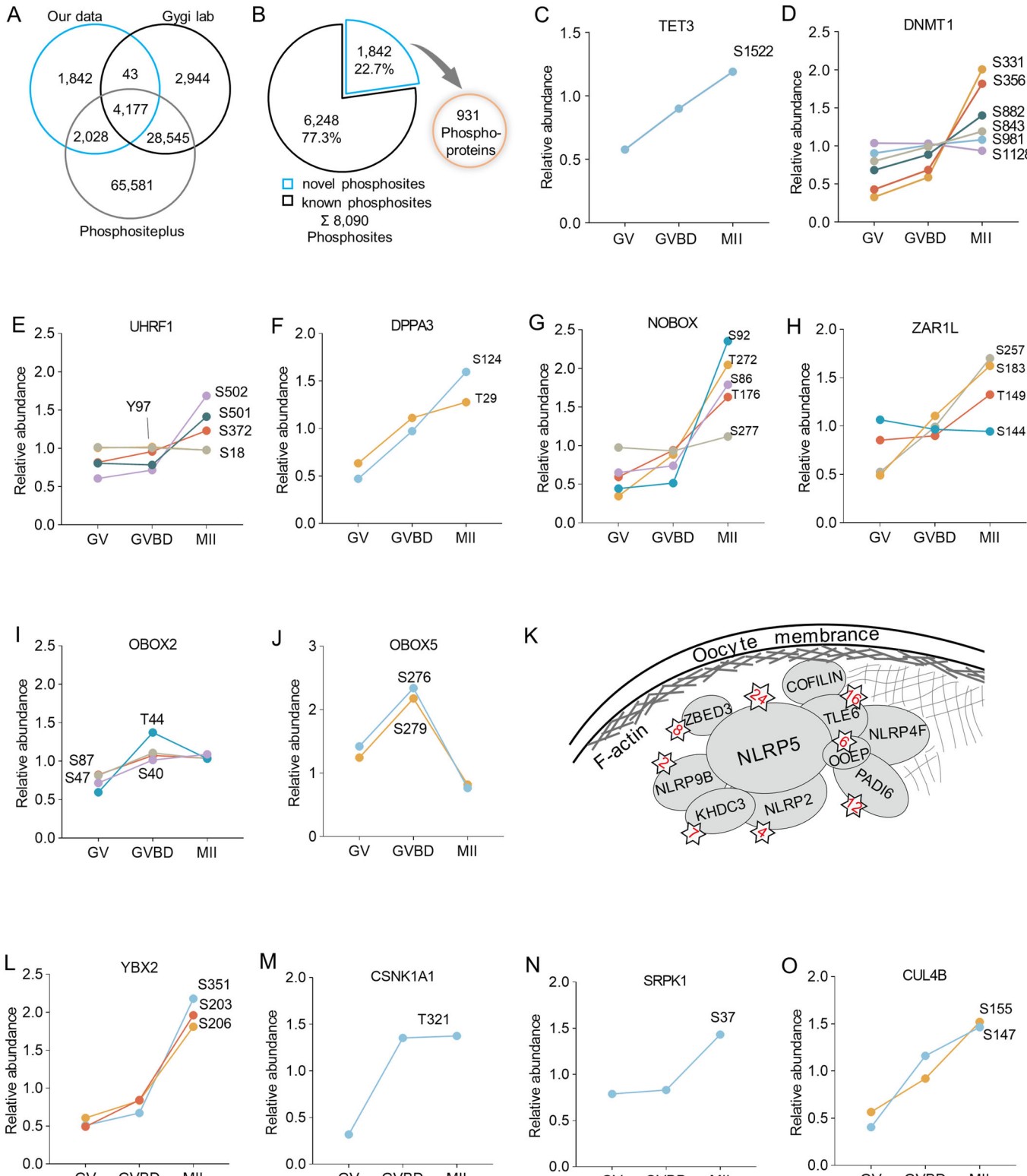

**Figure 3. Novel phosphosites identified in mouse oocytes.**

(A) The overlap of phosphorylation sites among our study, Gygi's data and PhosphoSitePlus database. (B) Pie chart showing the percentage of novel phosphosites identified in oocytes and their corresponding proteins. (C–J) Relative abundance of novel phosphosites in representative proteins at different stages. (K) Schematic model of core subcortical maternal complex, with the number of phosphosites indicated for each component. (L–O) Relative abundance of novel phosphosites in representative proteins at different stages. Source data are available online for this figure.

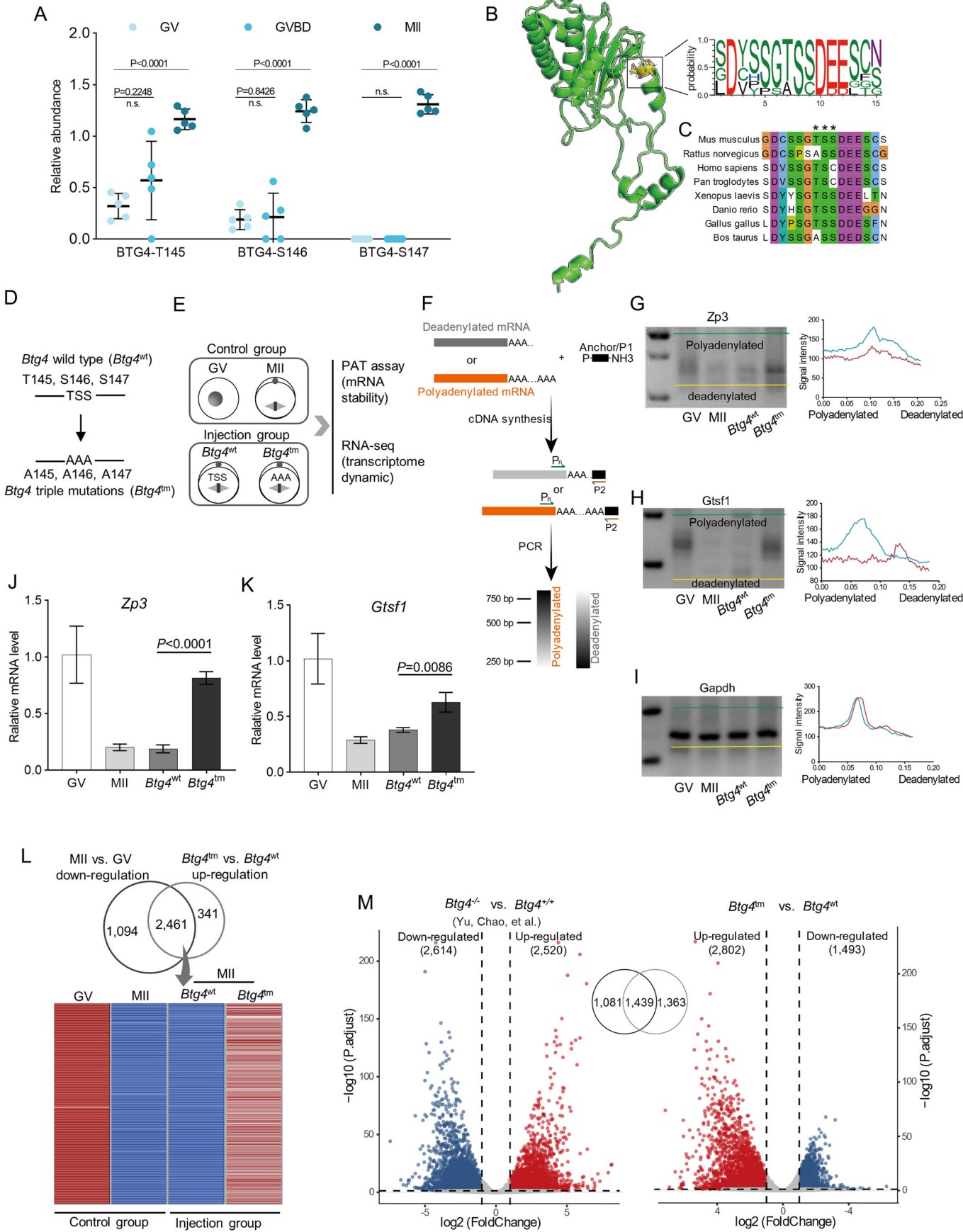

**Figure 4. BTG4 phosphorylation is required for maternal mRNA degradation.**

(A) Relative abundance of three phosphosites (Thr145, Ser146, and Ser147, abbreviated as "TSS") on BTG4 protein during oocyte maturation. Data are expressed as mean percentage ±SD from five independent replicates. Welch's *t* test was used to analyze the difference between GV and GVBD at the BTG4-T115 p-site, while Two-tailed Student's *t* test was used for statistical analysis between other data sets, comparing to GV oocytes. The *p* value is labeled in the figure. (B) BTG4 protein structure is predicted by Robetta and TSS phosphosites are indicated by yellow residues. Sequence logo illustrating the phosphorylation cluster (15-amino acid flanking regions of TSS phosphosites) across multiple species. (C) Alignment of BTG4 phosphorylation cluster across different species. Asterisks indicate TSS. (D) Schematic presentation of phosphomutant design of BTG4-TSS. (E) Schematic presentation of sample collection and RNA analysis. GV and MII oocytes in control group were collected from in vitro-cultured oocytes. *Btg4*^wt and *Btg4*^tm oocytes were collected from MII oocyte with wild-type or mutant BTG4 overexpression. (F) Diagram showing the strategy of the mRNA poly(A) tail (PAT) assay. P1 anchor primer, P2 P1-antisense primer, Pn gene-specific primer. (G–I) PAT assay showing changes in the poly(A)-tail length for the indicated transcripts in control group (GV and MII oocytes) and injection group (*Btg4*^wt and *Btg4*^tm oocytes). (J, K) Relative abundance of the indicated transcripts in control group (GV and MII oocytes) and injection group (*Btg4*^wt and *Btg4*^tm oocytes), determined by RT-qPCR. Data are expressed as mean percentage ±SD. *n* = 4 independent replicates. In total, 50 oocytes were analyzed for each group. Two-tailed Student's *t* test and Welch's *t* test were used for the statistical analysis of *Zp3* and *Gtsf1*, respectively. The *P* value is labeled in the figure. (L) Heatmap illustrating the downregulated transcripts in MII vs. GV oocytes and the aberrantly accumulated transcripts in *Btg4*^tm vs. *Btg4*^wt oocytes, respectively. (M) Volcano plot showing the differentially expressed genes in *Btg4* knockout oocytes and *Btg4*^tm oocytes, respectively. Student's *t* test followed by Benjamini–Hochberg (BH) *P* value adjustment (*P*.adjust) was used for the statistical analysis. Source data are available online for this figure.

kinases (~60%; 128/224) were found to be phosphorylated, and phosphorylation of 105 kinases (~80%; 105/224) experienced the dynamic regulation during oocyte maturation (termed regulated kinases in this paper, Fig. 6A). These regulated kinases belong to all major kinase families, with a higher representation of CAMK (Calcium/calmodulin-dependent protein kinases) and AGC (cAMP-dependent, cGMP-dependent and protein kinase C) family (Fig. 6B,C). In particular, only 8 out of 95 regulated kinases simultaneously displayed the differential accumulation at the protein level (Dataset EV4), reflecting that phosphorylation, rather than protein abundance, modulates the temporal kinase function in meiotic oocytes. Interestingly, among these regulated kinases, we discovered numerous previously-unnoticed kinases in oocytes (Fig. 6D–F; Appendix Fig. S7A–S7L). For example, the phosphorylation levels of MAST4 (S2478/S1906/S1781), WNK1 (T12/T17/T71) and PDPK1 (S38/S25/S244) were dramatically elevated accompanying with the meiotic resumption, indicative of the altered kinase activity. In addition, our data not only verified the known phosphosites on oocyte kinase (i.e. MAP4K4-S795/629/646 and MAP3K4-S1241/424/449), but also implicated that the novel sites on these enzymes (i.e. MAP4K4-S5 and MAP3K4-S114/461/T459) undergo phosphorylation and possibly control their activity (Appendix Fig. S7M,S7N). Above findings together reflect the dynamic and fine-tuning regulation of kinase activities in oocytes.

### Unraveling the effect of kinase inhibition on oocyte maturation

Previous studies have shown that phosphorylation can significantly impact kinase function (Aerts et al, 2015; Chong et al, 2001; Johnson et al, 1996). In the kinase analyses, we observed changes in kinase phosphorylation, which may suggest altered kinase activity. Next, we evaluated the potential effects of 13 representative uncharacterized kinases (WNK1, CSNK1A1, DYRK1A, GRK6, CSNK1E, PDPK1, RIPK2, RSK2, SRPK1, STK3, VRK1, SMG1, and PBK) on oocyte maturation by inhibitor treatment (Figs. 6G and EV4A). Markedly, suppression of each individual kinase was able to reduce the first polar body (Pb1) extrusion (Figs. 6G and EV4B–EV4M), strongly suggesting their functional involvement in oocyte maturation. In particular, we examined the impact of WNK1 inhibition on oocyte development in detail. First, fully grown GV oocytes were cultured in medium supplemented with various concentrations of WNK1 inhibitor (WNK-IN-11), and then relevant maturation phenotypes were assessed at the indicated time points (Fig. 6H). As shown in Fig. 6I–K, WNK-IN-11

treatment caused the delayed GVBD and diminished Pb1 extrusion in a dose-dependent manner. Confocal scanning microscopy further revealed that WNK1 inhibition induced a high frequency of meiotic defects in metaphase I oocytes including spindle disorganization and chromosome misalignment (Fig. 6L,M), indicative of the deficient assembly of meiotic apparatus. Moreover, we evaluated the in vivo effects of WNK-IN-11 on oocyte development through the tail vein injection in mice (Appendix Fig. S8A). As shown in Appendix Fig. S8B,S8C, similar to the in vitro studies, the maturation rate of those ovulated oocytes was significantly lowered once WNK1 activity was suppressed. These findings demonstrated that kinase WNK1 is required for orderly meiosis in oocytes.

### Kinase-substrate network analysis

Kinase recognizes the sequence motif surrounding the substrate phosphorylation sites. We next sought to explore the kinases with enriched substrate motifs for each cluster, based on the significantly altered phosphoproteomes mentioned above (Fig. 2E). Using GPS5.0 (Wang et al, 2020), the kinase-substrate phosphorylation relationship was annotated, and the oocyte kinase-substrate phosphorylation network (KSPN) was constructed (Dataset EV18). The activity regulation for 101 kinases (termed predicted kinases here) was estimated according to the phosphorylation changes in their known substrates (Fig. EV5A). Consistent with previous observations (Fig. 6C), more than 50% of predicted kinases belong to the CMGC, AGC, and CAMK family, with no kinases belonging to the TK (Tyrosine kinase) family (Fig. EV5B). Importantly, we noted that CSNK1A/1D, RIPK2, CHUK and WNK1 are the major kinases controlling the phosphorylation dynamics in cluster 1, and RPS6KA2/A3/A4/A5/B2, PRKCH/CO/G2/D2, VRK1 and SMG1 primarily modulate the phosphorylation changes in cluster 3 (Fig. EV5C). For instance, the phosphorylation levels of CSNK1A substrates clearly suggested the progressive decline of its activity from GV to MII stage (Fig. EV5D). CDK1 and PDE3A, as two predicted substrates of CSNK1A (Fig. EV5E), have been widely reported to be associated with meiotic resumption in oocytes (Adhikari et al, 2016; Han et al, 2006). Hence, these data indicate that CSNK1A-mediated CDK1 and PDE3A phosphorylation status might participate in the control of meiotic maturation. By contrast, MAPKs, CDKs, GSK3A/3B and DYRK1A/2 seem responsible for the phosphorylation dynamics in cluster 2, 4 and 5 (Fig. EV5C). Given the phosphorylation changes of substrates in cluster 5, ERK

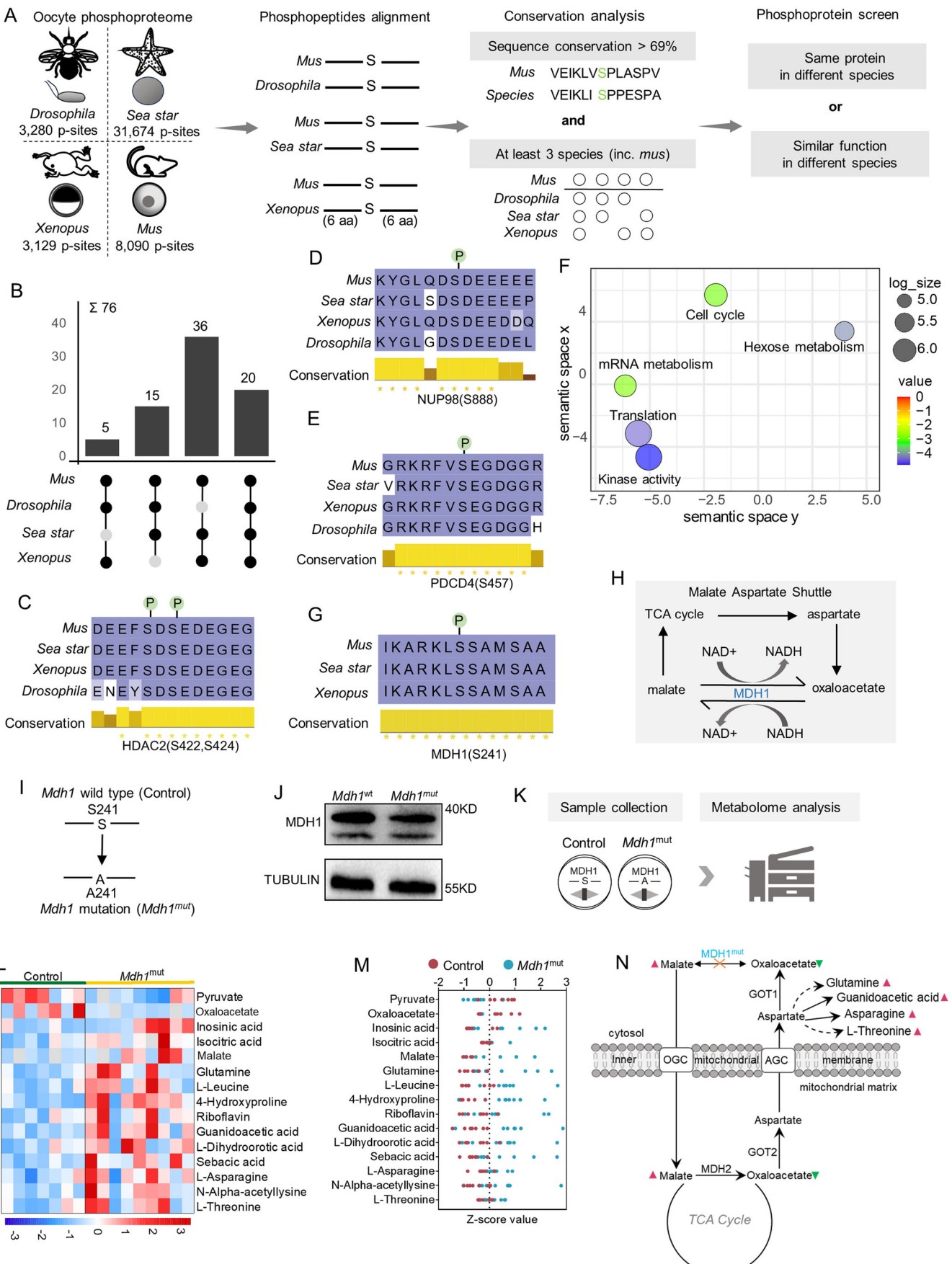

◀ **Figure 5. Conserved phosphosites across multiple species.**

(A) The flowchart illustrating the process of screening for conserved phosphosites across the four species (*Drosophila, Sea star, Xenopus,* and *mus musculus*). (B) Upset plot showing the identified 76 conserved phosphosites across the four species. (C–E) Alignment of sequences surrounding conserved phosphosites in HDAC2 (C), NUP98 (D), and PDCD4 (E). (F) REVIGO clusters of significantly overrepresented (*P* value < 0.01) GO terms for proteins with conserved phosphosites. (G) Alignment of sequences surrounding conserved phosphosites (S241) in MDH1. (H) Schematic diagram of Malate-Aspartate Shuttle. (I) Schematic representation of the MDH1 phosphomutant design. (J) Immunoblotting showing the overexpression of exogenous MDH1 (MDH1$^{wt}$ and MDH1$^{mut}$) protein in oocytes. (100 oocytes per lane). (K) Diagram showing the sample collection for metabolic analysis. (L) Heatmap showing the dynamics of 15 differential metabolites between *Mdh1*$^{mut}$ group and control group. (M) Z-score plot of 15 differential metabolites compared between *Mdh1*$^{mut}$ group and control group. (N) Schematic diagram of malate-aspartate shuttle, TCA cycle and the relevant metabolic pathways. Metabolites increased in *Mdh1*$^{mut}$ oocytes are indicated by red filled triangles, and metabolites decreased in *Mdh1*$^{mut}$ oocytes are indicated by green filled triangles. AGC aspartate–glutamate carrier, OGC malate-2-oxoglutarate carrier, GOT1 Glutamic-Oxaloacetic Transaminase 1, GOT2 Glutamic-Oxaloacetic Transaminase 2. Source data are available online for this figure.

activity was predicted to gradually elevate during oocyte maturation (Fig. EV5F). Although ERK has been implicated in diverse events in oocyte development (Fan and Sun, 2004), its exact targets are largely unknown. Here, by performing kinase-substrate network analysis, we discovered numerous potential substrates and phosphosites (i.e. TSC1-S321, FLNA-T1750, and PAPOLG-S556) of ERK, which perhaps mediate the different biological process in oocytes (Fig. EV5G).

## Deciphering phosphatase dynamics during oocyte maturation

About 1000 sites on proteins experienced dephosphorylation during oocyte maturation (Fig. 2E), suggesting the involvement of protein phosphatases. 67 phosphatases were detected in oocytes, and phosphorylation occurred in 29 of them. About 90% (26/29) of phosphatase showed the dynamic phosphorylation (called regulated phosphorylation here) in meiotic oocytes, reflecting the potential changes in activity (Fig. 7A,B). Overall, we identified 77 regulated phosphosites on these 26 phosphatases, with the majority of sites displaying the upregulated phosphorylation (Fig. 7C–F; Appendix Fig. S9A–S9H). CDC25B has been recognized as an important phosphatase required for resumption of meiosis in mammalian oocytes (Lincoln et al, 2002). Here we discovered several novel phosphosites (Fig. 7G, S12, S209, S78, S87) in addition to the well-known sites (Fig. 7C; i.e. S15, S158, and S351) on CDC25B (Swain et al, 2003; Wang et al, 2010; Zhao et al, 2015). Further exploration is warranted to determine whether these sites are functional. Besides, we found numerous tyrosine phosphatases (PTPs) (i.e. DUSP7, MTMR family, PTPN family, and PTPR family) undergoing the drastic changes in phosphorylation (Fig. 7D–F; Appendix Fig. S9A). For example, the newly identified sites on MTMR14 (i.e. S479, T620, S501, S516) showed the progressive increase in phosphorylation during oocyte maturation (Fig. 7D,H), whereas the phosphorylation levels of MTMR-S629, DUSP7-S220 and PTPN13-S1286 were significantly reduced (Fig. 7I). It is worth pointing out that these phosphatases have never been examined in oocytes. Therefore, how they control the phosphorylation kinetics during maturation deserves further investigation.

## Discussion

Protein phosphorylation is closely associated with oocyte development. However, the precise phosphorylation dynamics in mammalian oocyte meiosis are still unknown. In the current study, we conducted an integrated analysis of phosphoproteomics and proteomics to delineate the phosphorylation landscape of oocytes during in vivo maturation. We systematically analyzed specific phosphosites in mouse oocytes, as well as conserved phosphosites across multiple species. Through functional approaches, we revealed the involvement of several crucial phosphosites in diverse biological processes during maturation. Moreover, we discovered numerous previously unexplored kinases and phosphatases in oocytes experiencing significant changes in phosphorylation.

## Phosphorylation landscape during oocyte maturation

For an extended period, *Xenopus laevis* has served as an ideal model for studying the dynamics of phosphorylation during oocyte maturation. Due to the scarcity of experimental material, protein phosphorylation in mammalian oocyte has been investigated mainly by using knockout model in combination with inhibitor treatment. Studying the phosphoproteome of mouse oocytes holds paramount significance in unraveling the intricate regulatory mechanisms governing mammalian oogenesis and even embryogenesis. Here, we therefore established temporal phosphoproteome profiles during in vivo maturation. In parallel, we conducted quantitative proteomic studies to bolster the phosphoproteomics data. More than 8000 phosphosites were accurately identified. Our study, together with another work (Cheng et al, 2022), provide a comprehensive description of the dynamics of phosphorylation in mammalian oocytes. The observation of a significantly lower quantity of phosphorylated tyrosine residues in oocytes compared to mouse somatic cells (Giansanti et al, 2022) and testis (Qi et al, 2014) suggests the distinctive regulatory mechanisms. Based on the integrated profiles of proteome and phosphoproteome, our analyses reveal an array of key phosphosites and phosphoproteins. Notably, the upregulated phosphosites exhibited a higher abundance compared to the downregulated ones. Such a dynamic phosphorylation may play a crucial role in orchestrating intricate signaling cascades necessary for successful oocyte development (Presler et al, 2017). This extensive phosphorylation dynamic occurred independently of changes in the overall protein levels, reflecting the functional specificity and importance of phosphorylation in meiotic oocytes. In addition, protein phosphorylation in meiotic oocytes displayed the differing change tendency, which appears responsible for the modulation of different cellular events (Fig. 2). In future, more research should be focused on understanding the developmental significance of those phosphorylation changes. The findings may provide valuable insights into the molecular mechanisms

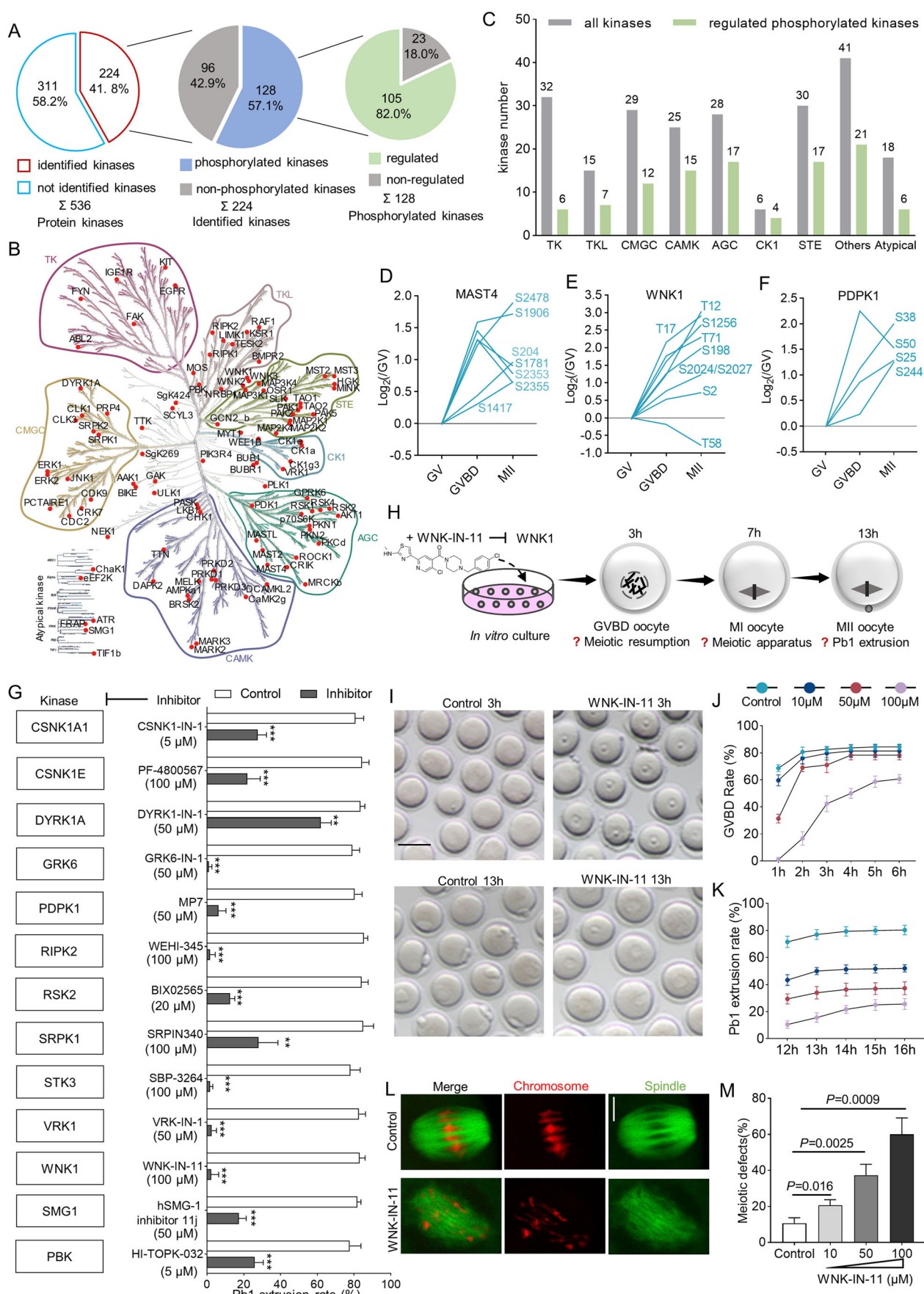

**Figure 6.   Unveiling crucial kinases orchestrating oocyte maturation.**

(A) Pie chart showing the numbers of total kinases (left), identified kinases (middle), and phosphorylated kinases (right). (B) Regulated phosphorylated kinases annotated to the major kinase families using www.kinhub.org. TK tyrosine kinases, TKL tyrosine kinase-like, STE homologs of the yeast STE7, STE11, and STE20 genes, CK1 casein/cell kinase 1 family, AGC protein kinase A, G, C families, CAMK calmodulin/calcium-regulated kinases and some non-calcium-regulated families, CMGC is CDK, MAPK, GSK3, and CLK kinase families. (C) Bar plot representing the percentage of all kinases (gray) and regulated phosphorylated kinases (green) quantified in oocytes from each of the major mammalian kinase families. (D–F) Regulated phosphosites on MAST4 (D), WNK1 (E), and PDPK1 (F). (G) Effects of different kinase inhibitors on the first polar body extrusion. Data are expressed as mean percentage ± SD from three independent replicates in which at least 100 oocytes were analyzed for each group. Two-tailed Student's $t$ test was used for statistical analysis. **$P < 0.01$, ***$P < 0.001$. (H) Schematic presentation of the WNK-IN-11 treatment experiments. (I) Bright-field images of control and WNK-IN-11-treated oocytes. Scale bars, 50 µm. (J) Quantitative analysis of GVBD rate in oocytes treated with WNK-IN-11 at different concentrations. Data are expressed as mean percentage ± SD from three independent replicates in which at least 100 oocytes were analyzed for each group. (K) Quantitative analysis of Pb1 extrusion in oocytes treated with WNK-IN-11 at different concentrations. Data are expressed as mean percentage ± SD from three independent replicates in which at least 100 oocytes were analyzed for each group. (L) Representative confocal images of control and WNK-IN-11-treated oocytes stained with a-tubulin antibody to visualize the spindle (green) and with propidium iodide to visualize chromosomes (red). Scale bars, 10 µm. (M) Quantitative analysis of meiotic defects in control and WNK-IN-11-treated oocytes. Data are expressed as mean percentage ±SD from three independent replicates in which at least 100 oocytes were analyzed for each group. Two-tailed Student's $t$ test was used for statistical analysis. Source data are available online for this figure.

driving meiotic maturation and pave the way for targeted interventions or improvements in assisted reproductive technologies.

## Phosphosites in mouse oocytes

In this study, a pivotal focus lies in the identification and characterization of novel phosphosites in oocytes. Over 20% of phosphosites we identified are novel for mouse oocytes, and absent in somatic cells, suggesting the presence of unique and oocyte-specific phosphorylation events. For instance, DPPA3, a maternal effector, participates in protection of DNA methylation by preventing conversion of 5mC to 5hmC in the maternal pronucleus (Han et al, 2019). Two novel phosphosites were found in its functional domains (T29 and S124) (Nakamura et al, 2012) (Fig. 3), raising the possibility that they may be essential for epigenetic control. Three previously unknown phosphosites (T145/S146/S147) were identified in BTG4, and phosphomutant compromises its ability for mRNA degradation in oocytes (Fig. 4). Nevertheless, expression of exogenous BTG4wt and BTG4mut seemed to reduce the accumulation of endogenous BTG4 (Appendix Fig. S3C), which needs to be clarified in the future. Moreover, we found that the novel phosphosites occur not only on maternal effectors, but also on some proteins that are widely expressed across multiple tissue and cell types, such as YBX2, CUL4B and ZFP57. This implies that ubiquitously expressed proteins can be differentially regulated by phosphorylation in an oocyte-specific manner, playing a unique role.

The resumption and maturation of oocyte meiosis in diverse species may be governed by analogous protein phosphorylation pathways, such as CDK1, PKA, and MAPK signaling. Exploring conserved phosphosites in oocytes across multiple species is important for gaining insights into the reproductive processes in humans. Therefore, we conducted a comparative analysis of the oocyte phosphoproteomes of four model organisms: mouse, *Xenopus*, *Drosophila*, and *Starfish*, and successfully discovered 76 conserved phosphosites. Analysis of these sites indicate that translational control and metabolic modulation may be the evolutionarily conserved events controlled by phosphorylation modification in female germ cells. For example, we found the perturbations in oocyte metabolism resulting from the mutation of the conserved site S241 in MDH1 protein. Moreover, the phosphomutant in MDH1 not only significantly affects its direct substrates (isocitrate and malate), but also has a broad impact

on the synthesis of amino acids (Fig. 5). MDH1 phosphomutant represents a new metabolic defect in the malate–aspartate shuttle, not only in mouse oocytes but also perhaps in human oocytes. The systematic study of post-translational modifications in ribosomal proteins of mammalian oocytes has not been conducted. Our phosphoproteomics analyses identified numerous phosphosites in ribosomal proteins, particularly in the p-stalk complex, including RPLP0, RPLP1, RPLP2, and RPL12. Follow-up phosphomutant experiments confirmed that RPL12 phosphorylation regulates the translation of specific subsets of mRNAs during meiosis. In addition, we also noticed several conserved phosphosites within essential proteins such as CAMK2G-T278 and SRPK1-S51. Nonetheless, in order to uncover universal principles underlying reproductive processes, the functions of these sites need to clarified in the future.

## Kinase and phosphatase in oocytes

Kinases are the key regulators for oocyte development. Here, we conducted an analysis of kinase activity based on phosphoproteome data. It's well-accepted that the phosphorylation status of a kinase is intricately linked to its enzymatic activity, and modifications at specific phosphorylation sites can either activate or inhibit the kinase (Ardito et al, 2017). Therefore, monitoring the phosphorylation changes of a kinase provides valuable insights into its functional regulation. Firstly, we detected more than 120 phosphorylated kinases (24% of total kinome) and found that 82% (105 kinases) of them show regulated phosphorylation of least one residue (Fig. 6). Based on changes in phosphorylation of their annotated substrates, we estimated their activity dynamics in mouse oocytes (Fig. EV5) and discovered that many unexplored kinases are essential for oocyte maturation (Fig. 6). Increasing evidence indicate that kinase mutations in oocytes can result in abnormal maturation and embryo developmental failure (Sang et al, 2018; Zhang et al, 2021). Hence, systematic screening the critical kinases and their phosphosites in oocytes would be helpful for understanding and preventing women infertility and birth defects.

Compared to kinases, the exploration of phosphatases in the process of oocyte maturation has been relatively limited. The dephosphorylation of key protein sites by phosphatases is necessary for meiosis (Hattersley et al, 2016). In this study, a comprehensive list of identified phosphatases with dynamically changing phosphorylation is presented. For example, MTMR14, a

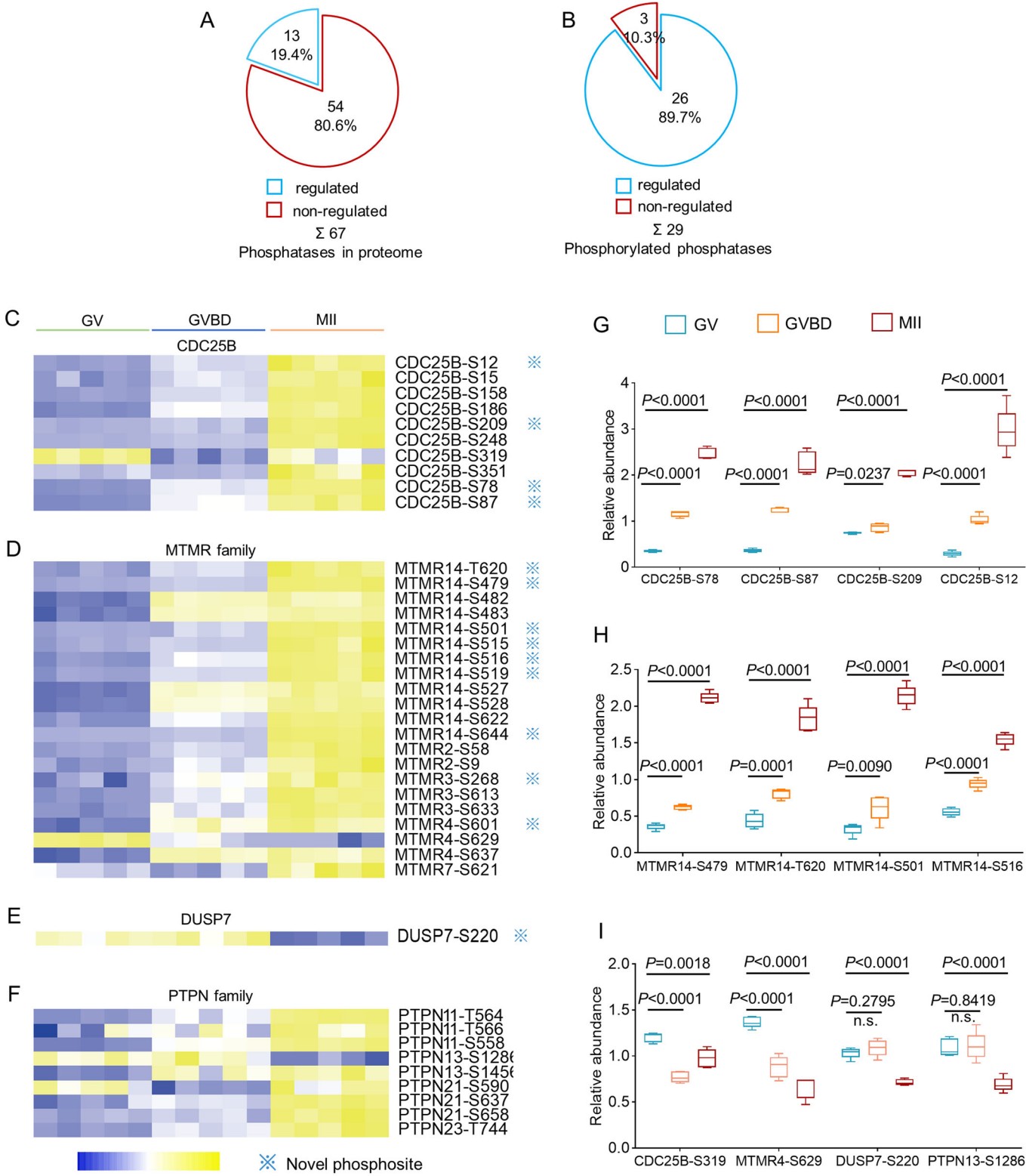

**Figure 7. Phosphorylation dynamics of phosphatases during oocyte maturation.**

(A) Pie chart showing the numbers of regulated and non-regulated phosphatases in oocyte proteome. (B) Pie chart showing the numbers of regulated and non-regulated phosphorylated phosphatases. (C–F) Heatmap depicting the phosphorylation dynamics of phosphorylated phosphatases categorized into distinct families. Novel phosphosites identified in each phosphatase are indicated by blue asterisk. (G) Relative phosphorylation levels of the novel phosphosites identified in CDC25B. Box plots: centerlines show the medians; box limits indicate the 25th and 75th percentiles; whiskers extend to the minimum and maximum. Data are expressed as mean percentage ± SD from five independent replicates. Two-tailed Student's $t$ test was used for statistical analysis, comparing to GV oocytes. The $P$ value is labeled in the figure. (H) Relative phosphorylation levels of the representative MTMR14 tyrosine phosphatase. Box plots: centerlines show the medians; box limits indicate the 25th and 75th percentiles; whiskers extend to the minimum and maximum. Data are expressed as mean percentage ± SD from five independent replicates. Two-tailed Student's $t$ test was used for statistical analysis, comparing to GV oocytes. The $P$ value is labeled in the figure. (I) Representative phosphatases with downregulated phosphorylation levels at specific phosphosite. Box plots: centerlines show the medians; box limits indicate the 25th and 75th percentiles; whiskers extend to the minimum and maximum. Data are expressed as mean percentage ±SD from five independent replicates. Two-tailed Student's $t$ test was used for statistical analysis, comparing to GV oocytes. The $P$ value is labeled in the figure. Source data are available online for this figure.

phosphoinositide phosphatase, displayed phosphorylation at numerous residues, with half of them being novel phosphosites in oocytes. Ongoing research in the lab is to determine the role of these phosphosites on phosphatases in mammalian oocytes.

Due to the relatively low abundance of phosphoproteins and the technical challenges associated with phosphoproteomic analysis of low-input samples, a large number of oocytes were used in this study. In our preliminary experiment, ~2500 phosphosites were detected from a pool of 2000 mouse oocytes (~16 µg of protein). According to a previous report (Humphrey et al, 2018), at least 200 µg of protein starting material is likely required for the identification of about 10,000 phosphosites. This number of phosphosites can largely meet the needs of data analysis. Hence, a total of 30,000 oocytes (~240 µg of protein), isolated from 950 mice, were used for the phosphoproteomic profiling here. 10,000 oocytes were retrieved for each stage (GV/GVBD/MII), and were divided into five replicates (2000 oocytes for each). Simultaneously, other parts of these mice, including granulosa cells, muscle, liver, and adipose tissues, were collected for different research projects in the lab.

A notable strength of our study lies in the comprehensive exploration of in vivo oocyte phosphoproteome during maturation. Over 8000 phosphosites provide valuable resource for probing the phosphorylation modification in oocyte meiosis. However, numerous oocytes were utilized for phosphoproteomics profiling. With the advancements in mass spectrometry and acquisition modes (Bortel et al, 2024), employing minute samples or even single-cell phosphoproteomics could substantially reduce the number of oocytes required for similar investigations. On the other hand, our data may not reflect the exact fold changes in proteome and phosphoproteome due to the ratio compression inevitably caused by Tandem Mass Tag (TMT) quantification (Karp et al, 2010). Technical improvements (i.e., the implementation of a causal model of ion interference) may help resolve this issue (Madern et al, 2024). In addition, various inhibitors were utilized to evaluate the effects of kinase activity on oocyte maturation in current study. We observed the significantly reduced Pb1 extrusion in oocytes treated with CSNK1A1 inhibitor (CSNK1-IN-1) (Fig. EV4B), consistent with a previous report (Wang et al, 2013). Nevertheless, knockdown via RNAi or microinjection of CKIα antibodies did not block the maturation progression of mouse oocytes (Gross et al, 1997; Qi et al, 2015). Such a discrepancy may arise from the differential effects of different approaches on enzymatic activity or abundance. Knockout targeting the specific kinase is helpful in clarifying its function. Besides, the datasets presented here were derived from superovulated oocytes, and hormone stimulation might influence oocyte phosphoproteome. For functional analysis of specific phosphosite in oocyte, transgenic or site-specific mutant mouse would be a perfect model.

## Methods

**Reagents and tools table**

| Reagent/resource | Reference or source | Identifier or catalog number |
|---|---|---|
| **Antibodies** | | |
| Rabbit monoclonal anti-BTG4 | Abcam | Cat# ab206914; RRID:AB_2861140 |
| Mouse monoclonal anti-Myc | Abcam | Cat# ab18185; RRID:AB_444307 |
| Mouse monoclonal anti-α-tubulin-FITC | Sigma | Cat# F2168; RRID:AB_476967 |
| Mouse monoclonal anti-Flag | Proteintech | Cat# 66008-4-Ig; RPID: AB_2918475 |
| Goat anti-rabbit IgG-HRP | Proteintech | Cat#SA00001-2; RRID: AB_2722564 |
| Goat anti-Mouse IgG-HRP | Proteintech | Cat# SA00001-1; RRID: AB_2722565 |
| Mouse monoclonal anti-α-tubulin-FITC | Sigma | Cat# F2168; RRID:AB_476967 |
| pan Phospho-Serine/Threonine Rabbit Polyclonal Antibody | Beyotime | Cat# AF5725 |
| **Bacterial and virus strains** | | |
| TOP10 Competent E. coli | TIANGEN | Cat# CB104 |
| **Experimental models** | | |
| Mice | Nanjing Medical University | N/A |
| Oocyte | This study | N/A |
| **Chemicals, enzymes and other reagents** | | |
| M16 medium | Nanjing Luanchuang Co., China | Cat# M02-B |

| Reagent/resource | Reference or source | Identifier or catalog number |
|---|---|---|
| M2 medium | Nanjing Luanchuang Co., China | Cat# M01-B |
| Milrinone | Sigma-Aldrich | Cat# M4659 |
| Mineral oil | Sigma-Aldrich | Cat# M8410 |
| propidium iodide | Thermo Fisher | Cat# P3566 |
| T4 RNA Ligase | Thermo Fisher | Cat# AM2141 |
| AscI endonuclease | New England Biolabs | Cat# R0558S |
| FseI endonuclease | New England Biolabs | Cat# R0588S |
| BamHI endonuclease | New England Biolabs | Cat# R3136S |
| NotI endonuclease | New England Biolabs | Cat# R3189S |
| T4 DNA ligase | New England Biolabs | Cat# M0202S |
| DpnI endonuclease | New England Biolabs | Cat# R0176S |
| Phusion high-fidelity DNA polymerase | New England Biolabs | Cat# M0530L |
| RIPA | Beyotime | Cat# P0013B |
| Anti-fade medium | Vectashield | Cat# H1000 |
| dimethylsulfoxide | Sigma-Aldrich | Cat# 276855 |
| **Critical commercial assays** | | |
| RNAprep Pure RNA Isolation Kit | Tiangen | Cat# DP431 |
| Quantitect Reverse Transcription Kit | Vazyme | Cat# R212-01 |
| QIAquick PCR Purification Kit | Vazyme | Cat# DC301-01 |
| qPCR SYBR Green Master Mix | Vazyme | Cat# Q141-02 |
| SP6 mMESSAGE mMACHINE Kit | Thermo Fisher | Cat# AM1340 |
| Pierce ECL Western Blotting Substrate | Thermo Fisher | Cat# 32106 |
| SuperScript III First-Strand Synthesis SuperMix | Thermo Fisher | Cat# 18080-400 |
| Fast Silver Stain Kit | Beyotime | Cat# P0017S |
| **Oligonucleotides and other sequence-based reagents** | | |
| Primer sequences for overexpression analysis | This Study | see Table EV1 |
| Primer sequences for Poly(A) tail assay | This Study | see Table EV1 |
| Primer sequences for qRT-PCR | This Study | see Table EV1 |
| **Recombinant DNA** | | |
| Plasmid: Myc tags in pCS2+ | This Study | N/A |

| Reagent/resource | Reference or source | Identifier or catalog number |
|---|---|---|
| Plasmid: Flag tags in pCS2 | This Study | N/A |
| Plasmid: Myc tags-Btg4 in pCS2+ | This Study | N/A |
| Plasmid: Myc tags-Btg4 mutation in pCS2+ | This Study | N/A |
| Plasmid: Flag tags-Pabpn1l in pCS2 | This Study | N/A |
| Plasmid: Flag tags-Mdh1 in pCS2 | This Study | N/A |
| Plasmid: Flag tags-Mdh1 mutation in pCS2 | This Study | N/A |
| Plasmid: Flag tags-Rpl12 in pCS2 | This Study | N/A |
| Plasmid: Flag tags-Rpl12 mutation in pCS2 | This Study | N/A |
| **Inhibitors** | | |
| CSNK1-IN-1 | MCE | Cat# HY148489 |
| PF-4800567 | MCE | Cat# HY12470 |
| DYRK1-IN-1 | MCE | Cat# HY132308 |
| GRK6-IN-1 | MCE | Cat# HY142812 |
| MP7 | MCE | Cat# HY14440 |
| WEHI-345 | MCE | Cat# HY18937 |
| BIX 02565 | MCE | Cat# HY16104 |
| SRPIN340 | MCE | Cat# HY13949 |
| SBP-3264 | MCE | Cat# HY132969 |
| VRK-IN-1 | MCE | Cat# HY126542 |
| WNK-IN-11 | MCE | Cat# HY112094 |
| hSMG-1 inhibitor 11j | MCE | Cat# HY124719 |
| HI-TOPK-032 | MCE | Cat# HY101550 |
| **Metabolites** | | |
| α-Ketoglutarate | Sigma-Aldrich | Cat# 75890 |
| Deoxycytidine | Sigma-Aldrich | Cat# D3897 |
| 2-Hydroxyglutarate | MCE | Cat# HY-113038B |
| 3-Hydroxypyridine | Sigma-Aldrich | Cat# H57009 |
| 3-Methylindole | Sigma-Aldrich | Cat# M51458 |
| 3-Methylhistidine | Sigma-Aldrich | Cat# M9005 |
| Adenine | Aladdin | Cat# A108804 |
| Adenosine | Adamas | Cat# 67712A |
| Capric acid | Adamas | Cat# 49153A |
| Citric acid | Aladdin | Cat# C108872 |
| Creatinine | Sigma-Aldrich | Cat# C4255 |
| Cystathionine | Sigma-Aldrich | Cat# C7505 |
| 5-Hydroxylysine hydrochloride | Sigma-Aldrich | Cat# H0377 |
| Deoxyinosine | Aladdin | Cat# D119465 |
| Glucosamine 6-phosphate | Sigma-Aldrich | Cat# G5509 |

| Reagent/resource | Reference or source | Identifier or catalog number |
|---|---|---|
| Dimethylbenzimidazole | Sigma-Aldrich | Cat# D147206 |
| Isocitric acid trisodium salt hydrate | Sigma-Aldrich | Cat# I1252 |
| Dodecanedioic acid | Adamas | Cat# 77196A |
| Dodecanoic acid | Adamas | Cat# 23651A |
| D-Glucaric acid potassium salt | Sigma-Aldrich | Cat# S4140 |
| Fumaric acid | Adamas | Cat# 14061B |
| Glucose 6-phosphate | TRC | Cat# G595338 |
| Glucose | MCE | Cat# HY-121965 |
| Glutamine | Sigma-Aldrich | Cat# G8540 |
| Glutaric acid | Sigma-Aldrich | Cat# G3407 |
| Glycerophosphocholine | Adamas | Cat# 44657A |
| Glycine | Adamas | Cat# 65953A |
| Glycolic acid | Adamas | Cat# 83663A |
| Guanine | Aladdin | Cat# G104274 |
| Hexadecanedioic acid | Adamas | Cat# 60631A |
| 4-Hydroxyproline | Sigma-Aldrich | Cat# H54409 |
| Hypotaurine | Sigma-Aldrich | Cat# H1384 |
| Indoleacetic acid | Aladdin | Cat# I101072 |
| Inosinic acid | Sigma-Aldrich | Cat# I2879 |
| L-Carnitine | Adamas | Cat# 64024A |
| L-Cystine | Aladdin | Cat# C108225 |
| L-Dihydroorotic acid | Sigma-Aldrich | Cat# D7128 |
| L-Dopa | Adamas | Cat# 68867A |
| L-Histidine | Aladdin | Cat# H108260 |
| L-Leucine | Adamas | Cat# 70880A |
| L-Lysine | Adamas | Cat# 66313B |
| L-Malic acid | Adamas | Cat# 91942A |
| L-Proline | Aladdin | Cat# P108709 |
| L-Serine | Adamas | Cat# 65987B |
| L-Tryptophan | Adamas | Cat# 79767B |
| Methionine | Aladdin | Cat# M101130 |
| N-alpha-acetyllysine | Sigma-Aldrich | Cat# A2010 |
| N-acetyl-L-alanine | Aladdin | Cat# A105813 |
| N-acetylneuraminic acid | Aladdin | Cat# A100555 |
| Niacinamide | Adamas | Cat# 92273A |
| Nicotinic acid | Aladdin | Cat# N103652 |
| oxidized glutathione | Aladdin | Cat# G105428 |
| Pantothenol | Sigma-Aldrich | Cat# 76200 |
| Phosphoethanolamine | Sigma-Aldrich | Cat# P0503 |
| Pipecolic acid | Aladdin | Cat# S136371 |
| Pyridoxine | Aladdin | Cat# P139145 |
| Pyroglutamic acid | Adamas | Cat# 92246A |

| Reagent/resource | Reference or source | Identifier or catalog number |
|---|---|---|
| Pyruvate | Sigma-Aldrich | Cat# 107360 |
| Quinic acid | Sigma-Aldrich | Cat# 138622 |
| Glutathione | Aladdin | Cat# G105426 |
| Rhamnose | Adamas | Cat# 51662A |
| Riboflavin | Adamas | Cat# 85641A |
| S-adenosylmethionine p-toluenesulfonate salt | Sigma-Aldrich | Cat# A2408 |
| Sebacic acid | Sigma-Aldrich | Cat# 283258 |
| Sorbitol | Adamas | Cat# 60805A |
| Succinic acid | Adamas | Cat# 14056B |
| Taurine | Adamas | Cat# 13304A |
| Tetradecanedioic acid | Sigma-Aldrich | Cat# D221201 |
| Tryptamine | Aladdin | Cat# T101154 |
| Tryptophanol | Sigma-Aldrich | Cat# 469971 |
| Uracil | Sigma-Aldrich | Cat# U0750 |
| Uridine | Aladdin | Cat# U108810 |
| Xanthosine dihydrate | Sigma-Aldrich | Cat# X0750 |
| 1,11-Undecanedicarboxylic acid | Aladdin | Cat# T283675 |
| 3-Indolebutyric acid | Sigma-Aldrich | Cat# 57310 |
| 5-Methylcytosine | Aladdin | Cat# M133493 |
| 5-Methyldeoxycytidine | Aladdin | Cat# D122897 |
| Adenosine monophosphate | Aladdin | Cat# A136967 |
| ADP | Aladdin | Cat# A119474 |
| Chenodeoxycholic acid | Aladdin | Cat# C104902 |
| Cinnamylic acid sodium salt | Aladdin | Cat# C170870 |
| Creatine | Aladdin | Cat# C105933 |
| Cysteic acid | Aladdin | Cat# S161224 |
| Cytidine monophosphate | Sigma-Aldrich | Cat# C1131 |
| dCMP | Sigma-Aldrich | Cat# D7750 |
| Dimethylglycine | Sigma-Aldrich | Cat# D1156 |
| Epinephrine | Aladdin | Cat# E113174 |
| Glycylproline | Aladdin | Cat# G121422 |
| Guanidoacetic acid | Sigma-Aldrich | Cat# G11608 |
| Guanosine | Sigma-Aldrich | Cat# G6752 |
| Hesperidin | Sigma-Aldrich | Cat# H5254 |
| Indole-3-carbinol | Sigma-Aldrich | Cat# I7256 |
| Inosine | Adamas | Cat# 67731A |
| Kynurenic acid | Sigma-Aldrich | Cat# K3375 |
| L-Threonine | Sigma-Aldrich | Cat# T8625 |
| L-Valine | Sigma-Aldrich | Cat# V0500 |
| Maltotriose | Aladdin | Cat# M106947 |
| Mannitol | Sigma-Aldrich | Cat# M4125 |

| Reagent/resource | Reference or source | Identifier or catalog number |
|---|---|---|
| N-acetyl-D-glucosamine | Sigma-Aldrich | Cat# A8625 |
| N-acetylglutamine | Aladdin | Cat# A117200 |
| N-acetyllactosamine | Aladdin | Cat# S115550 |
| NADP | Aladdin | Cat# N303921 |
| N-glycolylneuraminic acid | Aladdin | Cat# G115995 |
| Phenylacetylglycine | Aladdin | Cat# N135929 |
| Pterin | Sigma-Aldrich | Cat# P1132 |
| Ribitol | Sigma-Aldrich | Cat# A5502 |
| Senecioic acid | Sigma-Aldrich | Cat# D138606 |
| Spermidine | Sigma-Aldrich | Cat# S2626 |
| Symmetric dimethylarginine | Aladdin | Cat# S413343 |
| Undecanedioic acid | Sigma-Aldrich | Cat# 177962 |
| Vitamin D3 | Aladdin | Cat# C105354 |
| Xanthine | Sigma-Aldrich | Cat# X7375 |
| Trehalose | Aladdin | Cat# D110019 |
| D-Ornithine monohydrochloride | Sigma-Aldrich | Cat# O5250 |
| Thymidine 5′-monophosphate disodium salt hydrate | Sigma-Aldrich | Cat# T7004 |
| 4-Acetamidobutyric acid | Aladdin | Cat# A169398 |
| Erythritol | Sigma-Aldrich | Cat# E7500 |
| 7-Ketocholesterol | Aladdin | Cat# H130174 |
| N-palmitoylsphingosine | Sigma-Aldrich | Cat# 43799 |
| Gamma-butyrolactone | Sigma-Aldrich | Cat# B103608 |
| Glycerol | Sigma-Aldrich | Cat# G9012 |
| Citrulline | Sigma-Aldrich | Cat# C7629 |
| Gluconolactone | Adamas | Cat# 89621A |
| DL-2-aminooctanoic acid | Sigma-Aldrich | Cat# 217700 |
| Indoxyl sulfate potassium salt | Sigma-Aldrich | Cat# I3875 |
| Uridine 5′-monophosphate | Sigma-Aldrich | Cat# U1752 |
| Trigonelline | Aladdin | Cat# T345622 |
| Cotinine | Aladdin | Cat# S303802 |
| L-Arginine | Sigma-Aldrich | Cat# 11009 |
| L-Asparagine | Sigma-Aldrich | Cat# A0884 |
| L-Aspartic acid | Sigma-Aldrich | Cat# A9256 |
| L-Glutamic acid | Sigma-Aldrich | Cat# G1251 |
| Raffinose | Aladdin | Cat# R413236 |
| Naringin | Sigma-Aldrich | Cat# 71162 |
| Aminoadipic acid | Aladdin | Cat# A100535 |
| Diaminopimelic acid | Sigma-Aldrich | Cat# 33240 |
| N-methylnicotinamide | Aladdin | Cat# N159805 |

| Reagent/resource | Reference or source | Identifier or catalog number |
|---|---|---|
| Oxalacetic acid | Sigma-Aldrich | Cat# O4126 |
| Glyoxalic acid monohydrate | Aladdin | Cat# G108309 |
| **Software** | | |
| ImageJ | NIH | https://imagej.nih.gov/ij/ |
| Prism (V8.0) | GraphPad | https://www.graphpad.com/scientific |
| MaxQuant (V1.6.5.0) | Max-Planck-Institute of Biochemistry | https://www.maxquant.org/ |
| Spectronaut™ (V17.0) | Biognosys | https://biognosys.com/software/spectronaut/ |
| GPS5.0 | Wang et al, 2020 | https://gps.biocuckoo.cn/download.php |
| clusterProfiler (V4.6.2) | Wu et al, 2021 | https://bioconductor.org/packages/release/bioc/html/clusterProfiler.html |
| MetaboAnalyst (V6.0) | Xia Lab @ McGill | https://www.metaboanalyst.ca/ |
| Jalview (V2.11.3.2) | Waterhouse et al, 2009 | https://www.jalview.org/ |
| DESeq2 (V1.40.1) | Love et al, 2014 | https://bioconductor.org/packages/release/bioc/html/DESeq2.html |
| ZEN | ZEISS | https://www.zeiss.com/microscopy/int/downloads.html |
| StepOnePlus | Thermo Fisher Scientific | https://www.thermofisher.cn/cn/zh/home.html |

## Mouse

All animal experiments were approved by the Animal Care and Use Committee of Nanjing Medical University and follow the rules and guidelines of the local animal ethical committee (Protocol No. IACUC-2110009). Female Crl:CD1 (ICR) mice were purchased from Charles River Laboratories China Inc. and housed at the Animal Core Facility of Nanjing Medical University. These mice were housed in ventilated cages with a standard 12 h/12 h light/dark cycle at room temperature (22 °C) under controlled humidity (20–30%), with food and water access ad libitum. In the present study, ICR mice were selected as the animal model because: (i) as an outbred stock, they have more genetic variation as compared to inbred strains, much like humans in a population; (ii) upon induction of superovulation, more oocytes can be retrieved from ICR mice relative to other strains. All mouse experiments were performed without randomization and blinding.

## Oocyte collection and culture

Six-week-old female mice underwent superovulation through the administration of 5 units of pregnant mare's serum gonadotropin (PMSG), followed by 5 units of human chorionic gonadotropin (hCG) 48 h after PMSG priming. Mice were humanely euthanized

by cervical dislocation at 0, 3, or 12 h post-hCG injection. For the collection of GV (10,000 oocytes from 272 mice) and GVBD (10,000 oocytes from 311 mice) oocytes, cumulus-oocyte complexes (COCs) were obtained by manually rupturing antral ovarian follicles, with cumulus cells removed by repeated pipetting. When collecting GVBD oocytes, those that did not undergo germinal vesicle breakdown were discarded. MII oocytes (10,000 oocytes from 367 mice) were collected by isolating COCs from oviduct ampullae, and cumulus masses were removed in a medium containing 0.5 mg/ml hyaluronidase at 37 °C. For in vitro maturation, fully grown GV oocytes were cultured in M16 medium under mineral oil at 37 °C in a 5% $CO_2$ incubator.

## Protein sample preparation, digestion, TMT labeling, fractionation and phosphopeptide enrichment

For TMT-based proteomic analysis, GV, GVBD and MII oocytes were analyzed as we previously described (Li et al, 2020). In brief, for each replicate, 2000 oocytes collected from each of the GV, GVBD, and MII stages were lysed in urea lysis buffer (8 M urea, 75 mM NaCl, 50 mM Tris, pH 8.2, 1% mixture, 1 mM NaF, 1 mM β-glycerophosphate, 1 mM sodium orthovanadate, and 10 mM sodium pyrophosphate) and subjected to centrifugation at 40,000× $g$ for 1 h. Cysteines were reduced with 5 mM DTT at 56 °C for 25 min and alkylated in 14 mM iodoacetamide (IAA) solution for 30 min in the dark at room temperature. Unreacted IAA was quenched by DTT for 15 min. Then 25 mM Tris, pH 8.2 was added to dilute the urea. The proteins were digested overnight at 37 °C in a solution of 5 ng/µl trypsin (Promega, Fitchburg, USA), and terminated by trifluoroacetic acid (TFA). After purified by an OASIS HLB 1cc Vac cartridge (Waters), the samples were then subjected to TMTpro 15-plex labeling according to the manufacturer's protocols.

To fractionate peptides for proteomic quantification, the TMT-labeled peptide mixtures were separated using the high-pH reversed-phase (HP-RP) fractionation technology based on the ACQUITY ® UPLC M-class system (Waters) with a BEH C18 Column (300 µm × 150 mm, 1.7 µm; Waters). A 128 min gradient (3% buffer B for 14 min, 3%–8%B for 1 min, 8%–29%B for 71 min, 29%–41% B for 12 min, 41%–100%B for 1 min, 100% buffer B for 8 min, 100%–3%B for 1 min, followed by 20 min at 3% B) was employed with buffer A (20 mM ammonium formate, pH 10) and buffer B (100% ACN). A total of 30 fractions were generated and dried with a SpeedVac concentrator.

To fractionate and enrich phosphopeptides, the TMT-labeled peptide mixture was fractionated using Agilent 1260 system with XBridge BEH C18 column (4.6 × 150 mm, 3.5 µm; Waters) into ten fractions. TMT-labeled phosphopeptides were enriched through Ti4+-IMAC (immobilized metal affinity chromatography, J&K Scientific) (Liu et al, 2022). Briefly, peptides from each fraction were dissolved in loading buffer (80% acetonitrile, 6% trifluoroacetic acid (TFA)), incubated with IMAC beads, washed with washing buffer I (50% acetonitrile, 200 mM NaCl, 6% TFA) and II (30% acetonitrile, 0.1% TFA), and eluted with elution buffer (10% NH4OH). The phosphopeptide eluates were dried and desalted using C18 StageTips (Thermo Fisher Scientific).

To construct a spectra library for DIA analysis, 1000 GV oocytes, 1000 MII oocytes and 1000 zygotes were collected for protein extraction, reduction, alkylation, trypsin digestion, and desalting by C18 columns (Waters) as described above. The purified peptides were separated into 12 fractions based on HP-RP fractionation technology using a BEH C18 Column (300 µm × 150 mm, 1.7 µm; Waters), as mentioned above.

For DIA-based protein quantification, 70 RPL12 mutant MII oocytes in each of the five replicates and 70 wild-type control MII oocytes in each of the six replicates were acquired, and lysed and digested in solution of 0.1% n-Dodecyl β-D-maltoside (DDM), 10 mM Tris(2-carboxyethyl)phosphine (TCEP), 25 mM cysteamine (CAA), 130 mM triethylammonium bicarbonate (TEAB), and 10 ng/µL trypsin at 37 °C for 5 h. Formic acid (FA) was introduced to adjust the solution to a final pH of 2-3. The peptide digests were desalted by C18 StageTips and dried in a SpeedVac.

## LC–MS/MS

For TMT-based quantification, TMT-labeled peptides or enriched TMT-labeled phosphopeptides were analyzed using an Orbitrap Fusion Lumos mass spectrometer (ThermoFisher Scientific) coupled to a Proxeon Easy-nLC 1200 system. Peptides were separated on an analytical column (75 µm × 150 mm, 1.7 µm, CoAnn Technologies) using a 95 min gradient (3% to 5% buffer B for 5 s, 5% to 15% buffer B for 40 min, 15% to 28% buffer B for 34 min and 50 sec, 28% to 38% buffer B for 12 min, 38% to 100% buffer B for 5 sec, 100% buffer B for 8 min) at 300 nl/min. The parameter settings for MS could be found in previously published paper (Li et al, 2022). Briefly, data were acquired with an MS1 scan for a $m/z$ range 350–1500 with a resolution of 60,000 followed by data dependent HCD MS2 spectra in the Orbitrap with a resolution of 50,000 and HCD collision energy of 36%. For DIA library construction, data were acquired with an MS1 scan for a $m/z$ range 350–1500 with a resolution of 60,000 followed by data dependent HCD MS2 spectra with a resolution of 15,000 and HCD collision energy of 30%.

For DIA-based quantification, samples were separated on an analytical column with the same parameters as above for TMT-based analysis. The parameters for Orbitrap Fusion Lumos mass spectrometer analysis were set as follows. The MS1 spectra were collected in Orbitrap at a scan range of 350–1500 $m/z$, a resolution of 120,000 and an AGC target of 250% as well as maximum injection time of 50 ms. Precursor ions were fragmented with HCD levels of 30%, an AGC target of 100% and dynamic maximum injection time. The fragments were scanned in an Orbitrap at resolution of 30,000, with isolation window of $m/z = 50$ for the first two windows and $m/z = 15$ for the remaining.

## Protein identification and quantification

For TMT-based quantification, the MaxQuant software (v1.6.5.0) was used to search the raw files against UniProt mouse proteome database. The FDR (false discovery rate) of identified peptides and proteins was set to 1%. Searches were performed using Trypsin/P enzyme specificity while allowing up to two missed cleavages. Carbamidomethylation of cysteine residue was set as fixed modifications. Variable modifications included oxidation of methionine, acetylation of protein N-termini, and phosphorylation of threonine, serine and tyrosine. For TMT settings, the protein quantification values were calculated using the reporter ion MS2 method of isobaric labels in MaxQuant. Phosphorylation sites,

peptides and proteins with FDR ≤1% were considered confident. Phosphorylation sites with localization probability above 0.75 and proteins with at least one unique peptide were subjected to quantitative analysis (Li et al, 2019).

For phosphorylation quantification, quantification values of phosphorylation sites with the highest priority (Class I > Class II > Class III) were used. The expression level of each protein was normalized by dividing the mean expression level of the corresponding protein across samples, and the expression level of each phosphorylation site was normalized in the same way. The normalized level of each phosphorylation site was then calibrated by dividing against the normalized level of the corresponding protein. If the protein level was zero in any of the samples, it was replaced with the lowest value in the protein dataset to avoid the invalid value by dividing zero. If the corresponding protein could not be quantified by proteomics analysis, the normalized level of the phosphorylation site was used directly for statistical analysis (Fan et al, 2020). The differential analysis of protein and phosphorylation sites within each group was performed using Student's $t$ test followed by Benjamini–Hochberg (BH) $P$ value adjustment ($P$.adjust). Only a protein or a phosphorylation site with $P$.adjust <0.05 and fold change (FC) > 1.5 were significant.

For DIA-based quantification, the raw files were searched with Spectronaut™ software (v17.0) in library-based search mode using a hybrid library by further merging the directDIA library (the UniProt mouse proteome database) and DIA library built from GV oocytes, MII oocytes and zygotes as described above. Peptides with lengths of 7 to 52 amino acids were considered for the search, and FDR ≤ 1% were considered confident on both precursor and protein group levels. The differential analysis of protein was performed using Student's $t$ test ($P$ value). Only a protein with $P$ value < 0.05 and fold change >1.2 were significant.

## Bioinformatics analysis

WebLogo 3.7.11 was used to generate frequency plots of amino acids surrounding modified sites base on 13-mer amino acid sequences centered on each phosphorylation sites (Crooks et al, 2004). GPS5.0 (Wang et al, 2020) was used to annotate the regulatory relationships between kinases and phosphorylated sites, which was then visualized by Cytoscape 3.8.2 (Saito et al, 2012). Kinase enrichment analysis of the significant upregulated phosphorylation site against the rest of the phosphorylation site was performed by Fisher's exact test (Li et al, 2022). The gene ontology (GO) and Kyoto Encyclopedia of Genes and Genomes (KEGG) pathway enrichment analyses were performed using clusterProfiler 4.6.2 package (Wu et al, 2021).

## Metabolomics

In total, 30 MII oocytes were harvested separately per sample, 7 (control group) and 9 (mutation group) replicates for each group. Samples were immediately flash-frozen in liquid nitrogen, and then stored at −80 °C. For metabolite extraction, samples were resuspended and sonicated in 60 μL of 80% methanol/water (vol/vol) using a non-contact ultrasonic crusher (Diagenode, Belgium). After cooling on ice for 10 min, samples were spun at 8000× $g$ for 10 min at 4 °C. Supernatant was dried and stored at −80 °C until instrumental analysis. The metabolomics data were collected using a standard metabolic profiling method we previously described (Li et al, 2020). Briefly, instrumental analysis was performed on an UPLC Ultimate 3000 system (Dionex, Germering, Germany) coupled to a Q-Exactive mass spectrometer (Thermo Fisher Scientific, Bremen, Germany). The chromatographic separation was conducted using Hypersil GOLD C18 column (100 mm × 2.1 mm, 1.9 μm) (Thermo Fisher Scientific) at 40 °C. The mobile phase consists of phase A (0.1% formic acid in ultra-pure water) and B (0.1% formic acid in pure ACN) with a flow rate of 0.4 mL/min. After the initial 3-min elution of 99% (vol/vol) phase A, the phase A gradually decreased to 1% (vol/vol) at t = 10 min. The phase A was maintained at 1% (vol/vol) for 5 min ($t$ = 15 min), and then immediately increased to 99% (vol/vol) for 2 min (t = 17 min). The mass spectrometer was performed in a full-scan mode ranging from 70 $m/z$ to 1050 $m/z$ with a 70,000 resolution in both positive and negative modes. Each sample was injected for analysis in a randomized fashion to avoid complications related to the injection order. Raw data were submitted to TraceFinder (v5.1). The metabolite identification was conducted by the comparison of accurate mass and retention time with the commercial standard compounds using the author-constructed library. All statistical analyses were performed using "R" (v4.1.0). SIMCA-P software (v14.1; Umetrics AB, Umea, Sweden) was used for orthogonal partial least squares-discriminant analysis (OPLS-DA). Calculate the Z-Score for each data point. If the Z-Score is greater than a certain threshold (±3), the data point is considered an outlier. The variable importance in projection (VIP) value > 1.00 and $P$ value < 0.1 with a 1.2-fold change of each metabolite were used as the combined cut-offs of the statistical significance. This combination criteria allow to find more biologically meaningful sets of metabolites than $P$ value alone (Han et al, 2022, McCarthy and Smyth, 2009). The pathway analysis was conducted with the MetaboAnalyst (v6.0).

## RNA-seq

The analysis of $Btg4^{wt}$ and $Btg4^{tm}$ oocyte transcriptomes was conducted using Smart-seq2, as detailed in our prior work (Han et al, 2022). Initial steps involved the reversal of single-strand RNA molecules using an oligo (dT) primer. The resulting amplified cDNA was fragmented into ~300 bp fragments and subsequently prepared using the TruePrep DNA Library Prep Kit by Vazyme. In the case of RNA-seq data, trim_galore (v0.6.10) was employed to remove adaptors and low-quality bases. Sequences were then aligned to the mouse reference genome (GRCm38) using HISAT2 (v2.2.0) with gencode annotation (vM25). Read quantification was performed with FeatureCounts (v2.0.6) using parameters: -t exon -g gene_id -Q 10 --primary -s 0 -p. Differential gene expression analysis utilized DESeq2 (v1.40.1), following previously described procedures (Love et al, 2014).

## Identification of conserved phosphosites

For the analysis of conserved phosphosites in oocytes across multiple species, we utilized four oocyte phosphoproteome datasets containing motif peptides corresponding to 13-mer amino acid sequences, with six amino acid residues flanking on either side of the phosphosite. These datasets were centered on each phosphorylation residue and included data from mouse (from this study), *Sea star* (Swartz et al, 2021), *Drosophila* (Zhang et al, 2019), and *Xenopus* (Presler et al, 2017). Phosphorylated peptides from the other three species were compared with those from mice, and those

exhibiting a sequence similarity greater than 69% (at least 9 out of 13 residues) were included in the subsequent analysis. At least three species' phosphorylated peptide sequences, including mice, with a similarity exceeding 69%, were included for further analysis. The functions of the phosphorylated peptides obtained from each species were individually analyzed to explore whether protein functions are evolutionarily conserved across species, performing similar or consistent biological functions. Jalview (v2.11.3.2) (Waterhouse et al, 2009) was used to generate multiple alignment plot of amino acids surrounding conserved sites base on 13-mer amino acid sequences centered on each phosphorylation sites.

## Overexpression analysis

The procedure for in vitro cRNA synthesis was performed as detailed in our previous work (Li et al, 2020). In brief, total RNA extracted from 100 oocytes using the RNAprep Pure RNA Isolation Kit (Tiangen, China) served as the starting material for cDNA synthesis, accomplished with the HiScript II 1st Strand cDNA Synthesis Kit (Vazyme, China). Following full-length PCR amplification, the products were cloned into the vector. Capped cRNAs were synthesized through in vitro transcription using the SP6 mMESSAGE mMACHINE kit (Ambion, CA, USA). For overexpression experiments, ~5–10 pl of cRNA at a concentration of 500 ng/ml was microinjected into fully grown oocytes. The Myc-pCS2+ vector was employed for BTG4 overexpression, while for the overexpression of other proteins, pCS2Flag was utilized. The phosphorylation mutation experiment involves mutating the phosphorylated serine/threonine to alanine, and constructing the mutation into the corresponding vector. The vector is then transcribed in vitro to mRNA, which is injected into the cytoplasm of oocytes using microinjection technique. The accuracy of the mutated sequence is verified through sanger sequencing, and protein expression is confirmed by western blot. The primer sequences can be found in Table EV1.

## Treatment of oocytes with inhibitors

All the inhibitors in this study were purchased from MedChemExpress (China). Inhibitors solutions were prepared in dimethyl sulfoxide (DMSO), and then diluted to yield a final concentration in maturation medium as needed. GV oocytes were in vitro cultured in M16 medium containing different doses of inhibitor for further analysis. Correspondingly, 0.1% DMSO was included as a control. The relevant phenotypes were examined at the indicated time points.

## Immunoprecipitation

Lysis of 500 oocytes was performed using IP buffer (50 mM Tris-HCl pH 7.4, 1 mM EDTA, 1% NP40, 150 mM NaCl) supplemented with protease inhibitors and PMSF, conducted on ice for 30 min. A total of 5 μg antibodies were interacted with protein A/G beads (Yeasen Biotech Co., Shanghai, China) in 250 μl Lysis buffer on a rotator at 4 °C for 4 h. Concurrently, cell or oocyte lysates underwent pre-cleaning with 30 μl protein A/G beads, rotating for 4 h at 4 °C. Subsequently, the protein A/G-coupled antibody was introduced to pre-cleaned lysates and incubated at 4 °C on a rotating wheel overnight. Following centrifugation at 2000× *g* for

10 min, supernatants were discarded, and the beads were thrice washed with IP buffer. The immune complexes were then denatured at 95 °C for 5 min with protein loading buffer. Finally, the immunoprecipitated complexes underwent western blotting using corresponding antibodies.

## Western blotting

Oocytes were lysed in RIPA buffer containing β-mercaptoethanol for 5 min on ice. The lysate was then supplemented with loading buffer and subjected to heat treatment at 95 °C for 5 min. The denatured proteins were loaded onto a 12% SDS-PAGE gel and subsequently transferred to PVDF membranes (Life Technologies). After blocking the membranes in PBST with 5% low-fat dry milk for 1 h at room temperature, they were incubated overnight at 4 °C in a diluted primary antibody solution. Following a minimum of three washes, the membranes were then incubated with a secondary antibody (HRP Goat anti-Rabbit/Mouse IgG, Proteintech, 1:2000) at room temperature for 1 h. Protein bands were visualized using Pierce ECL western blotting substrate (Thermo) according to the manufacturer's instructions.

## Silver staining

Fast silver stain kit was purchased from Beyotime Biotech (China). For electrophoresis, 100 oocytes at GV, GVBD, and MII stages were utilized. After electrophoresis, transfer the gel into approximately 100 ml of fixing solution (50 ml ethanol, 10 ml acetic acid, and 40 ml distilled water) and shake at room temperature on a shaker for 20 min. Discard the fixing solution, add 100 ml of 30% ethanol, and shake at room temperature on a shaker for 10 min. Discard the 30% ethanol, add 200 ml of Milli-Q grade distilled water or double-distilled water, and shake at room temperature on a shaker for 10 min. Discard the water, add 100 ml of silver staining sensitizing solution (1X), and shake at room temperature on a shaker for 2 min at a speed of 60–70 rpm. Discard the original solution, add 200 ml of Milli-Q grade distilled water or double-distilled water, and shake at room temperature on a shaker for 1 min. Repeat the washing step once. Discard the water, add 100 ml of silver staining solution (1×), and shake at room temperature on a shaker for 10 min. Discard the original solution, add 100 ml of Milli-Q grade distilled water or double-distilled water, and shake at room temperature on a shaker for 1 min. Discard the water, add 100 ml of silver staining developer, and shake at room temperature on a shaker for 5 min until the desired protein bands appear. Discard the silver staining developer, add 100 ml of silver staining stop solution (1×), and shake at room temperature on a shaker for 10 min. Discard the silver staining stop solution, add 100 ml of distilled water, and shake at room temperature on a shaker for 3 min.

## Immunofluorescence

Oocytes were fixed with 4% paraformaldehyde for 30 min and permeabilized with 0.5% Triton X-100 for 20 min. After blocking in 1% BSA-supplemented PBS for 1 h, the samples underwent an overnight incubation at 4 °C with FITC-conjugated anti-tubulin antibody. Chromosomes were stained with either propidium iodide for a duration of 15 min. Following several washes, the samples were mounted on an anti-fade medium (Vectashield, Burlingame,

CA, USA) and observed using a Laser Scanning Confocal Microscope (LSM 710, Zeiss, Germany).

## Quantitative real-time PCR

mRNA levels were assessed through quantitative real-time PCR (qRT-PCR), following the protocol described in our prior work (Sun et al, 2023). RNAprep Pure RNA Isolation Kit (Tiangen, China) was utilized for total RNA extraction from 50 oocytes, and first-strand cDNA was synthesized using a cDNA Synthesis Kit (Vazyme, China). Real-time PCR was conducted with SYBR Green on an ABI StepOnePlus Real-time PCR system (Applied Biosystems, CA, USA). *Gapdh* served as an internal control, and the experiments were conducted in triplicate at a minimum. The primer sequences can be found in Table EV1.

## Poly (A) tail assay

The Poly(A) Tail (PAT) assay was conducted with slight modifications based on the published protocol (Yu et al, 2016). In brief, RNAprep Pure RNA Isolation Kit (Tiangen, China) was employed for total RNA extraction from 200 oocytes at the designated time points. T4 RNA ligase anchored primer P1 (5′-P-GGTCACCTTGATCTGAAGC-NH2-3′) to RNA, and reverse transcription was carried out using SuperScript III First-Strand Synthesis SuperMix (Life Technologies, Invitrogen TM, Cat#: 18080-400) with the P1-antisense primer P2 (5′-GCTTCAGAT-CAAGGTGACCTTTTT-3′). The resulting products were amplified by PCR with gene-specific primers and primer P2. PCR conditions were as follows: 30 s at 94 ℃, 30 s at 50 ℃, and 50 s at 72 ℃ for 35 cycles, and the products were visualized on a 2.5% agarose gel. Primer sequences can be found in Table EV1.

## Statistical analysis

The experiments were independently replicated three times, yielding consistent results. Unless specified otherwise, the data presented in the figures represent a single representative experiment. GraphPad Prism (V8.0) for Windows was utilized for the analyses. Statistical comparisons were conducted using two-tailed Student's $t$ test, and Welch's $t$ test when appropriate. The data are expressed as the mean value ± SD. Statistically significant values of $P < 0.05$, $P < 0.01$, and $P < 0.001$ are indicated by asterisks (*), (**), and (***), respectively. All tests and $P$ values are provided in the corresponding legends and/or figures.

# Data availability

The data that support the findings of this study are available from the corresponding authors upon request. All mass spectrometry data and spectral identifications have been deposited in the ProteomeXchange Consortium via the PRIDE (Vizcaino et al, 2014) partner repository (https://www.ebi.ac.uk/pride/) with the data set identifier PXD050645 and PXD050652. Username: reviewer_pxd050645@ebi.ac.uk, Password: F9KIKQZS; Username: reviewer_pxd050652@ebi.ac.uk, Password: 6W011Msc. After logging into the PRIDE archive website with the provided account and password, the raw data of this study can be accessed. The transcriptome data have been deposited to GEO with the accession number GSE261544. (https://www.ncbi.nlm.nih.gov/geo/query/acc.cgi?acc=GSE261544) (enter token: uzknouoebvwrlqf).

The source data of this paper are collected in the following database record: biostudies:S-SCDT-10_1038-S44318-024-00222-1.

# Peer review information

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

## Acknowledgements

This work was supported by the National Natural Science Foundation of China (No. 81925014, 82221005 to QW and 82101736 to HS), the National Key Research and Development Program of China (No. 2021YFC2700400 to QW) and the Natural Science Foundation of Jiangsu Province (BK20190615 to LH).

## Author contributions

**Hongzheng Sun**: Data curation; Software; Formal analysis; Validation; Investigation; Visualization; Funding acquisition; Writing—original draft.

**Longsen Han**: Funding acquisition; Validation; Investigation. **Yueshuai Guo**: Software; Methodology. **Huiqing An**: Formal analysis; Investigation; Methodology. **Bing Wang**: Visualization; Methodology. **Xiangzheng Zhang**: Visualization; Methodology. **Jiashuo Li**: Software; Validation. **Yingtong Jiang**: Investigation; Methodology. **Yue Wang**: Methodology. **Guangyi Sun**: Supervision; Investigation; Visualization. **Shuai Zhu**: Validation; Investigation; Visualization. **Shoubin Tang**: Validation; Investigation. **Juan Ge**: Formal analysis; Investigation. **Minjian Chen**: Visualization; Methodology. **Xuejiang Guo**: Software; Investigation; Methodology. **Qiang Wang**: Conceptualization; Resources; Data curation; Supervision; Funding acquisition; Project administration; Writing—review and editing.

Source data underlying figure panels in this paper may have individual authorship assigned. Where available, figure panel/source data authorship is listed in the following database record: biostudies:S-SCDT-10_1038-S44318-024-00222-1.

## Disclosure and competing interests statement

The authors declare no competing interests.

# Expanded View Figures

**Figure EV1. Quality control for the phosphoproteomics and proteomics data. Related to Fig. 1.**

(A) Workflow of Proteomics and Phosphoproteomics. (B) Principal component analysis depicting the clustering of five proteome replicates from GV, GVBD, and MII oocytes. (C) Heatmap illustrating the Pearson's correlation among the 15 proteome replicates obtained from GV, GVBD, and MII oocytes. (D) Principal component analysis depicting the clustering of five phosphoproteome replicates from GV, GVBD, and MII oocytes. (E) Heatmap illustrating the Pearson's correlation among the 15 phosphoproteome replicates obtained from GV, GVBD, and MII oocytes. (F) Coefficient of variance boxplot of proteome for each stage sample derived from GV, GVBD, and MII oocytes. Box plots: centerlines show the medians; box limits indicate the 25th and 75th percentiles; whiskers extend to the minimum and maximum. $n = 6700$. (G) Coefficient of variance boxplot of phosphoproteome for each stage sample derived from GV, GVBD, and MII oocytes. Box plots: centerlines show the medians; box limits indicate the 25th and 75th percentiles; whiskers extend to the minimum and maximum. $n = 8090$. (H, I) Total phosphopeptide (G) and peptide (H) intensities ranked ascending to illustrate the dynamic range of the dataset. (J) Bar chart showing the number of identified peptides (up), proteins (middle), and quantified proteins (bottom). (K) The overlap of proteins and phosphoproteins.

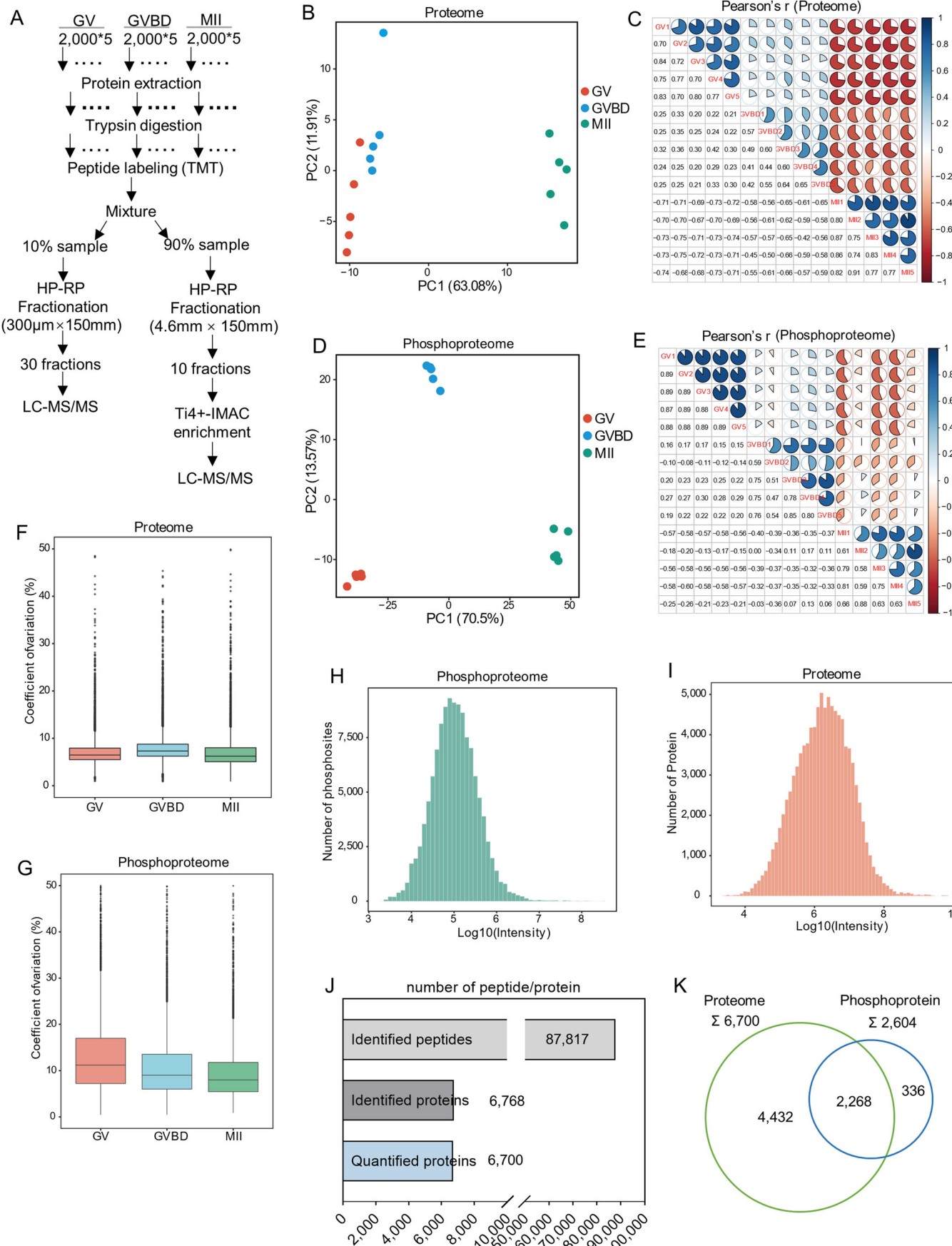

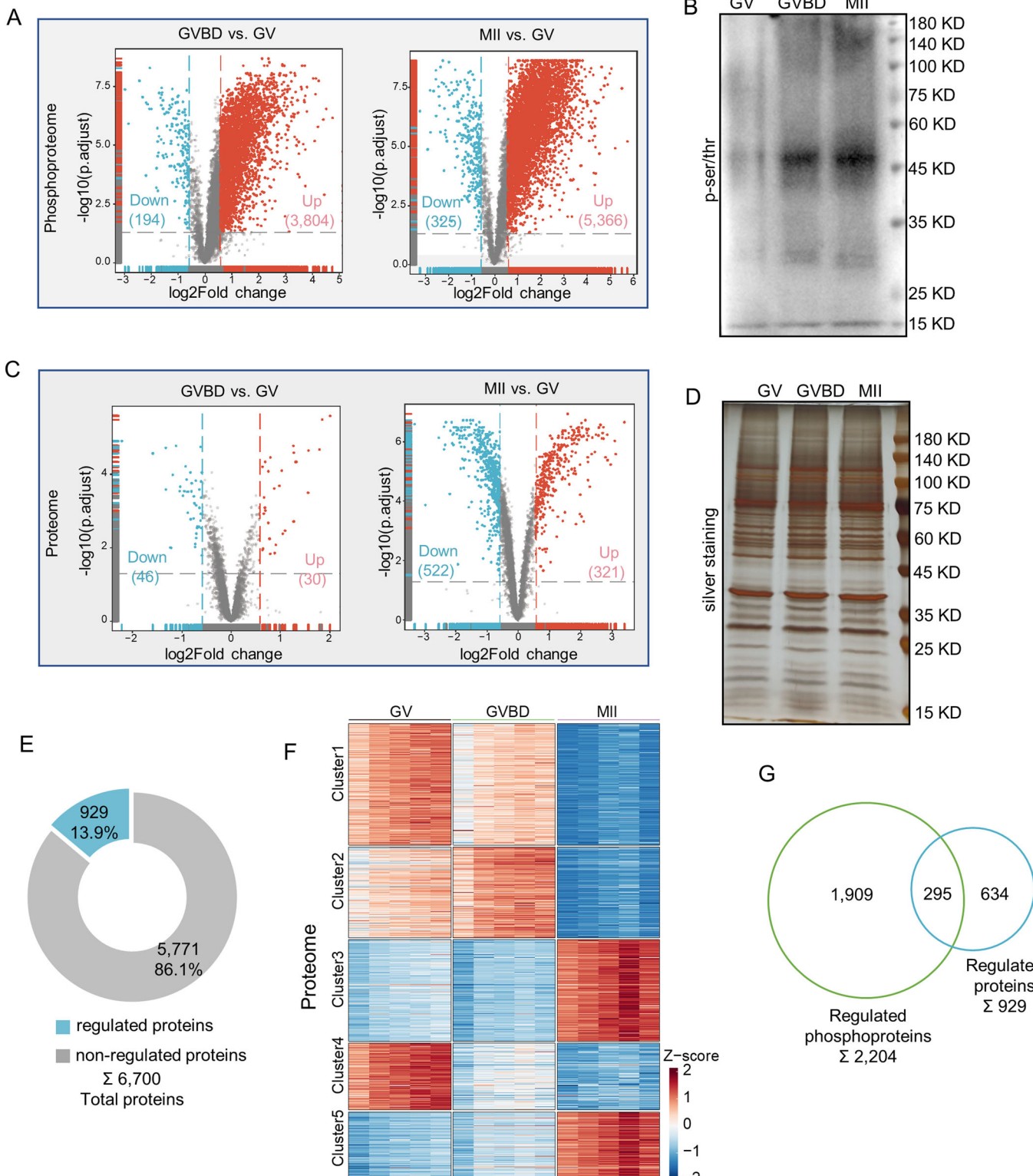

**Figure EV2. Features of dynamic phosphoproteome and proteome. Related to Fig. 2.**

(A) Volcano plots showing the differentially phosphorylated proteins between GVBD and GV (left), or between MII and GV (right). Student's *t* test followed by Benjamini–Hochberg (BH) *P* value adjustment (*P*.adjust) was used for the statistical analysis. (B) Representative immunoblots of oocytes at GV, GVBD, and MII stages probed with pan-Ser/Thr antibody. (C) Volcano plots showing the differentially expressed proteins between GVBD and GV (left), or between MII and GV (right). Student's *t* test followed by Benjamini–Hochberg (BH) *P*-value adjustment (*p*.adjust) was used for the statistical analysis. (D) Representative silver staining of oocytes at GV, GVBD, and MII stages for protein expression detection. (E) Pie chart showing the number and percentage of regulated proteins. (F) Heatmap illustrating the dynamic changes in regulated proteins during oocyte maturation. (G) The overlap of regulated proteins and regulated phosphoproteins.

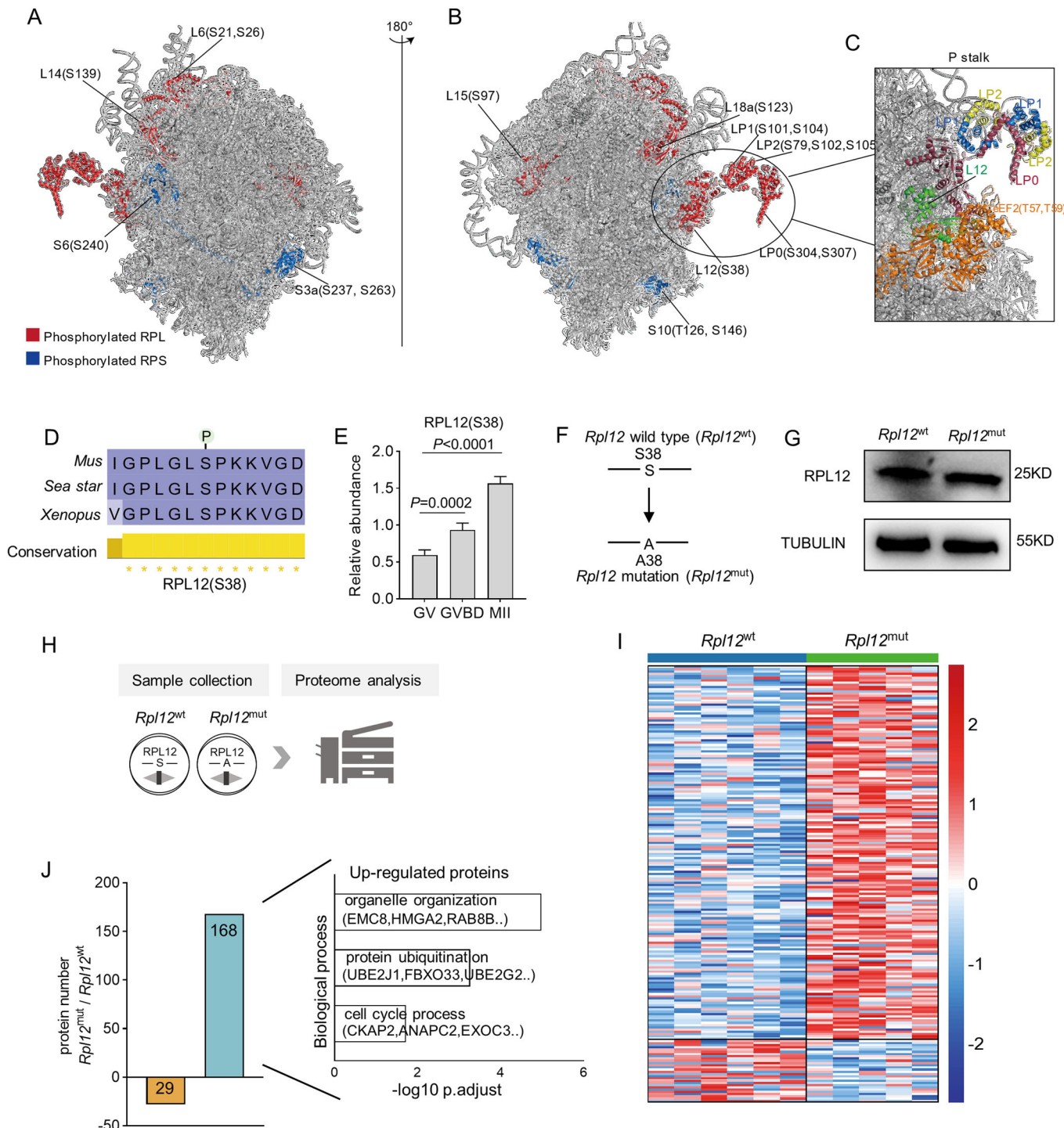

**Figure EV3. RPL12-Ser38 phosphorylation participates in maintaining translational homeostasis in oocytes. Related to Fig. 5.**

(**A, B**) Quantified phosphorylation sites of ribosome proteins (RPs) mapped to the ribosome structure (PDB: 4V6X). Phosphorylated RPSs and RPLs are shown in blue and red, respectively. RPL: Large ribosomal subunit proteins; RPS: Small ribosomal subunit proteins. (**C**) An enlarged view of the P stalk showing that RPL12-S38 is proximal to the ribosomal GTPase EEF2, RPLP0, RPLP1, and RPLP2. (**D**) Alignment and conservation analyses of RPL12 sequences flanking conserved phosphosites. (**E**) Bar chart showing the phosphorylation level of RPL12-S38 during oocyte maturation. The *p* value is labeled in the figure. Data are expressed as mean percentage ±SD from five independent replicates. Two-tailed Student's *t* test was used for statistical analysis, comparing to GV oocytes. (**F**) Schematic representation of the design for RPL12 phosphomutant. (**G**) Immunoblotting showing the overexpression of exogenous RPL12 (RPL12^wt and RPL12^mut) protein in oocytes. (100 oocytes per lane). (**H**). Diagram showing the sample collection for proteomic analysis. (**I**). Heatmap showing the differentially expressed proteins between *Rpl12*^wt and *Rpl12*^mut oocytes. (**J**). Bar chart showing the differentially expressed proteins between *Rpl12*^wt and *Rpl12*^mut oocytes. The number of upregulated (up-) and downregulated (down-) proteins in *Rpl12*^mut oocytes (left). Representative biological processes enriched for upregulated proteins in *Rpl12*^mut oocytes (right). Benjamini–Hochberg (BH) corrected *p*-value adjustment (*p*.adjust) was used for the enrichment analyses.

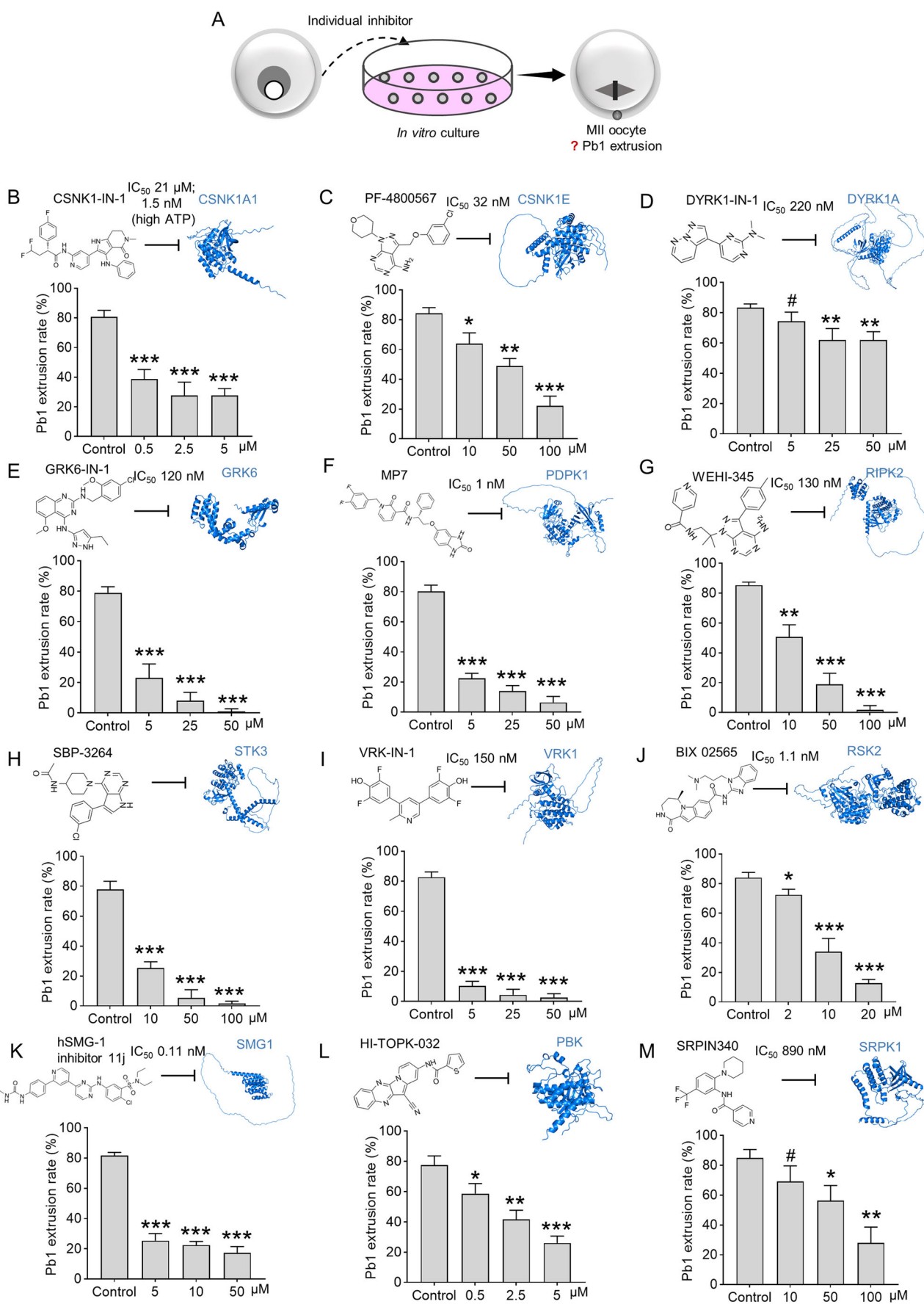

**Figure EV4.  Effects of different kinase inhibitors on oocyte maturation. Related to Fig. 6.**

(A) Schematic presentation of the inhibitor treatment experiment. (B–M) Quantitative analysis of the Pb1 extrusion rate in oocytes treated with different inhibitors. Data are expressed as mean percentage ±SD from three independent replicates in which at least 100 oocytes were analyzed for each group. Two-tailed Student's *t* test was used for statistical analysis, comparing to control group (DMSO treatment). $^{\#}P > 0.05$, $^{*}P < 0.05$, $^{**}P < 0.01$, $^{***}P < 0.001$.

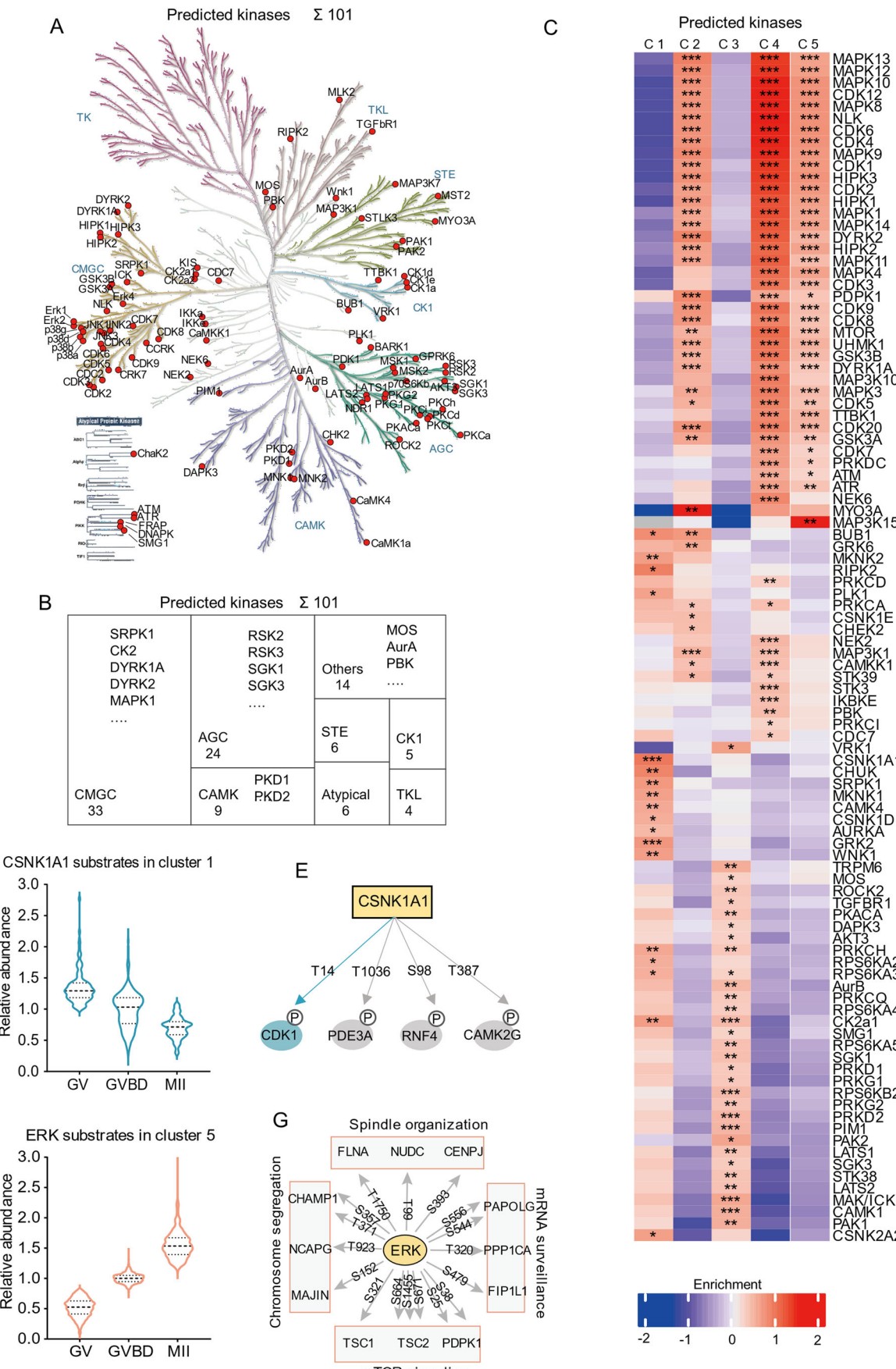

◄ **Figure EV5. Enrichment analysis of the predicted kinases in oocytes. Related to Fig. 6.**

(A) Kinase tree showing the predicted kinases annotated to the major kinase families. (B) The number of predicted kinases in each kinase family. (C) Heatmap showing the predicted kinases enriched in each cluster. $*P < 0.05$, $**P < 0.01$, $***P < 0.001$. Benjamini–Hochberg (BH) corrected $P$ value adjustment ($P$.adjust) was used for the enrichment analyses. (D) Phosphorylation levels of the predicted CSNK1A substrates in cluster 1 during oocyte maturation. $n = 191$. (E). Networks showing the representative phosphosites in annotated substrates (KSPN) of CSNK1A1 kinase. Proteins are shown as nodes and phospho-residue is indicated by the number. (F) Phosphorylation levels of the predicted ERK substrates in cluster 5 during oocyte maturation. $n = 672$. (G) Networks showing the representative phosphosites in annotated substrates (KSPN) of ERK kinases. Proteins are shown as nodes and phospho-residue is indicated by the number.

