## [Peer Review File · The EMBO Journal]

The global phosphorylation landscape of mouse oocytes during meiotic maturation

Hongzheng Sun, Longsen Han, Yueshuai Guo, Huiqing An, Bing Wang, Xiangzheng Zhang, Jiashuo Li, Yingtong Jiang, Yue Wang, Guangyi Sun, Shuai Zhu, Shoubin Tang, Juan Ge, Minjian Chen, Xuejiang Guo, and Qiang Wang

Corresponding author(s): Qiang Wang (qwang2012@njmu.edu.cn), Xuejiang Guo (guo_xuejiang@njmu.edu.cn)

Review Timeline:

Submission Date:	22nd Apr 24
Editorial Decision:	23rd May 24
Revision Received:	5th Jul 24
Editorial Decision:	5th Aug 24
Revision Received:	10th Aug 24
Accepted:	13th Aug 24

Editors: Hartmut Vodermaier and Ieva Gailite

Transaction Report:

Prof. Qiang Wang
Nanjing Medical University
State Key Laboratory of Reproductive Medicine and Offspring Health
Nanjing, Jiangsu
China

23rd May 2024

Re: EMBOJ-2024-117657
The global phosphorylation landscape of mouse oocytes during meiotic maturation

Dear Qiang,

Thank you again for submitting your phosphoproteomic analysis of mouse oocytes for consideration as a Resource Article in The EMBO Journal. I sent it to four referees with expertise in proteomics (referees 1 and 2) and mammalian oocyte maturation (referees 3 and 4), who have now provided the reports copied below. All referees appreciate the comprehensiveness and the potential value of the presented dataset. However, they also raise various presentational and experimental concerns that would need to be clarified prior to publication.

Should you be able to adequately and comprehensively address these concerns, we would be interested in pursuing a revised version further for EMBO Journal publication. Key aspects will be clarifying more experimental details, oocyte synchronization, phosphosite normalization to protein levels, and inhibitor concentrations. On the other hand, obtaining further or replicate datasets that would require additional large quantities of oocytes would not appear to be justified for the purpose of this revision.

Since it is our policy to allow only a single round of major revision, it will however be important to carefully respond to all points at the time of resubmission. Should this require more time than our default three-months revision period, we would be happy to offer an extension, during which our 'scooping protection' (meaning that competing work appearing elsewhere in the meantime will not affect our considerations of your study) would of course remain valid. Also, please do not hesitate to contact us in case you should want to discuss any specific points related to the referee reports and your revision.

Further information on preparing and uploading a revised manuscript can be found below and in our Guide to Authors. Thank you again for the opportunity to consider this work for The EMBO Journal, and I look forward to your revision.

With kind regards,

Hartmut

3) Revised manuscript text (including main tables, and figure legends for main and EV figures) has to be submitted as editable

text file (e.g., .docx format). We encourage highlighting of changes (e.g., via text color) for the referees' reference.

9) Digital image enhancement is acceptable practice, as long as it accurately represents the original data and conforms to community standards. If a figure has been subjected to significant electronic manipulation, this must be clearly noted in the figure legend and/or the 'Materials and Methods' section. The editors reserve the right to request original versions of figures and the original images that were used to assemble the figure. Finally, we generally encourage uploading of numerical as well as gel/blot image source data; for details see: embopress.org/page/journal/14602075/authorguide#sourcedata

At EMBO Press, we ask authors to provide source data for the main manuscript figures. Our source data coordinator will contact you to discuss which figure panels we would need source data for and will also provide you with helpful tips on how to upload and organize the files.

Further information is available in our Guide For Authors:

In the interest of ensuring the conceptual advance provided by the work, we recommend submitting a revision within 3 months (21st Aug 2024). Please discuss the revision progress ahead of this time with the editor if you require more time to complete the revisions. Use the link below to submit your revision:

Link Not Available

Referee #1:

The manuscript by Sun et al. provides a comprehensive phosphoproteomic analysis of mouse oocyte maturation by monitoring three stages. For this analysis, 30,000 oocytes were arrested at GV, GVBD, and MII stages, with 10,000 oocytes per stage divided into five replicates. Through phosphopeptide enrichment and TMT labeling, about 8000 phosphopeptides were quantified across different meiotic stages. The authors identified multiple novel sites and examined corresponding kinases and phosphatases. In addition, by using small molecule inhibitors, the role of 13 previously uncharacterized kinases in oocyte maturation was explored. Specific roles of three different phosphoproteins in oocytes were also explored using site-directed mutagenesis. Each subsequent experiment on local regulation provides descriptive results but lacks detailed molecular analysis and some controls. In general, this manuscript is rich in data and will be a valuable resource for the scientific community. However, before publication, the authors should address the following concerns:

Major Concerns:

1. One of the main purposes of the manuscript is to differentiate the phosphoproteomic profiling of three key stages of oocyte meiosis. However, the manuscript lacks information regarding the level of synchronization achieved at specific stages. Western blotting and microscopy should be performed to assess each stage and the effectiveness of arresting at a given stage, as well as the homogeneity of the populations. Simply showing the ratios of four known phosphosites is not enough to assess the pooled populations.

2. For defining each stage selective phosphorylation sites, each phosphosite needs to be normalized to its protein level. Otherwise, it is not clear if the regulation is at phosphorylation or protein level. Are the values in Figure 2 normalized to their protein level? Those values should also be reported in the Supplementary Table.
3. The experimental setup is not well explained at the beginning of the result section, some information such as the number of mouse/oocytes/replicates are in the discussion but not in the result part. Those details should be included in the result part or Figure 1A.
4. In myc-BTG4 wt vs myc-BTG4 tm expressing cells (Figure S5C), it appears that myc-BTG4 tm expressing cells have lower levels of endogenous BTG4 expression. Is it possible that the triple mutant is acting as dominant negative, leading to a depletion of endogenous BTG4 and causing the observed phenotype? This may be an alternative explanation to the proposed model. Additionally, the western blot in Figure S5 may not be sufficient to conclude that BTG4 tm disrupts the endogenous interaction with PABNL1, as there still appears to be an interaction that is not disrupted. Also the interactions are shown by exogenously expressed proteins, not the endogenous ones. This experiment requires quantification over replicates. Furthermore, it would be helpful to address the potential kinase and its presence among the identified potential kinases at oocyte maturation.
5. In Fig. S9E, is the phosphorylation normalized to the protein level? Is there a prediction for the responsible kinase? In this experiment, WT and phosphomutant RPL12 proteins are overexpressed in oocytes and the protein profile is monitored. How many proteins are identified here, and how are their distribution? Having disproportional up vs down-regulation may not be solely due to enhanced translation. There could be different explanations for this. More controlled experiments need to be done. In addition, "phosphorylation of ribosomal protein RPL12-S38 is critical for maintaining translational homeostasis in mammalian oocytes (Fig. S9K)" is very strong conclusion for this. All experiments are on the WT background the phenotype of only RPL12 phosphomutant expression cells is not examined and the experiment performed here does not address translational homeostasis.
6. The manuscript is focused on the regulated kinases, meaning the phosphosites of kinases are dynamic in oocytes. However, it would be more interesting to find out the responsible kinases that are active at particular stages and phosphorylate multiple substrates. Is it possible to distinguish kinase activities at different stages? By using inhibitors, multiple kinases are inhibited and all of them affect the Polar body1 extrusion rate. However, in multiple cases, the concentration of inhibitors is very high (around 10 uM-100uM). This brings the off-target effect issue. What other potential kinases are inhibited at this rate, and how specific are the treatments to a particular kinase?
7. The authors state multiple times that the rate of Tyrosine phosphorylations is significantly lower than in other tissues suggesting different activities of Tyrosine kinase in somatic versus germ cells. However, the difference could be solely due to technical reasons of having a low amount of protein starting material or applying different enrichment techniques. A similar amount of somatic and germ cells should be analyzed in parallel by either performing western blotting with a pan phospho-tyrosine antibody or the same phospho-enrichment protocol to conclude.
8. Similar to kinases, potential functional phosphatases are also suggested based on their phosphorylation dynamics, however, this may not be the case.
9. In the discussion section, it is mentioned that 200 µg of protein starting material is required for 10,000 phosphosites (Humphrey, 2018). However, due in part to upgraded mass spectrometry and acquisition modes, better higher number of phosphopeptides can be determined with much less starting material (ie. Bortel et. al, 2024, around 10ug, 15k sites). It may be better to be more specific when defining the limit of detection, as this has been changing due to recent advancements.

Minor Concerns:

1. Supplementary Table 1 should be in xlsx format. The current version of this table is not readable. It is not clear why some tables are in CVS format
2. Phosphopeptide enrichment should be included in the method subtitles
3. In Figure 2F right part of the graph needs a revision, those graphs are not clear

4. Why the phosphopeptide numbers are lower than phosphosites? Does it mean that there are not many multiply phosphorylated peptides.
5. Why Y97 is standing in Figure 3E, what are its measurements?
6. Figure 3K should be revised to a more informative representation, only the number of phosphosites without their sequence, structure, or cell stage ratios are not useful.
7. Across the manuscript there are overstatements, like
Line 263: On the other hand, we also discovered some fully conserved phosphosites across four species, and which have never been reported in oocytes (i.e., HDAC2-S422/S424, NUP98-S888, PDCD4-S457) (Fig. 5C-5E), implying the existence of evolutionarily conserved phosphorylation mechanisms controlling critical events in oocytes. Unless some functional assays are done it is not possible to predict that they are controlling critical events. Those comments should be revised.
8. There are typos like "Phosphorproteome" in the legends or figures.
9. In S9 K a single phospho-site mutation, does not make a protein phosphodead. It should be corrected.

Referee #2:

Sun et al perform proteomic and phosphoproteomic analyses of isolated mouse oocytes at the germinal vesicle (GV) stage, upon the breakdown of the GV upon meiotic resumption, and in meiosis II. This is an interesting analysis of meiosis by proteomic and phosphoproteomics with extensive follow-up analysis of regulated phosphorylation sites.

Phosphorylation is an essential regulatory mechanism governing meiotic progression. Previous studies have investigated protein abundance and phosphorylation changes in organisms producing large numbers of oocytes. However, similar studies in mice are hindered by the limited number of oocytes per animal at the time of birth.

The authors overcome this by using a very large number of oocytes. The authors isolate 30,000 oocytes. It would be helpful for the reader to have a sense of how many oocytes are isolated from each mouse and how many mice were used for each replicate. Images of the oocytes to demonstrate synchronicity should be included. The main text should also state how much protein was isolated for each replicate. Did the authors consider including an isobaric carrier? The description of the proteomic and phosphoproteomic methods needs clarification. The authors used 2,000 oocytes per replicate, which they lysed, digested, labeled, and HP-RP fractionated into 30 fractions. Next, the authors discuss the phosphoproteomic analysis and state that labeled peptides were fractions into fractions before enrichment. How was this mixture generated? Also, from 2,000 oocytes/replicate? The workflow in Figure 1A is inconsistent with the method description and should be corrected. Also, the number of replicates should be indicated and the labeling scheme. A more detailed workflow as supplemental data would be helpful.

The authors normalize phospho-site to protein abundances. However, the site was still included in further analyses if the protein was not detected or quantified. Please clarify. If yes, how are corrected and non-corrected sites distinguished in subsequent analyses? How many sites were corrected versus not?

A heatmap depicting proteome changes should be included as supplemental data. The accounting of phosphoproteins and proteins is confusing. Most studies find that the proteome is stable during meiotic progression. How do the authors explain this difference? Are specific proteins stable in some organisms but not in mouse?

The authors state in line 153: "These findings unveil that phosphorylation modification is a new layer of molecular control for oocyte maturation, complementing the proteome landscape we discovered previously". However, the highly dynamic changes in the phosphoproteome during meiosis are consistent with studies in other organisms. This should be rephrased.

About the comparison with the Gygi dataset, where the 1,842 phosphorylation sites not detected in the Gygi dataset on proteins detected in their dataset indicate oocyte-specific regulation or where the proteins not detected? This should be clarified.

BTG4 phosphorylation sites. Are these sites individually detected or as double and triple phosphorylation sites? Figure 4A makes it look like individual sites. Are these sites localized on the phosphopeptide? The authors generated a BTG4 3A mutant and did RNA-seq and PAT. An additional description of how the constructs were expressed and how their expression was confirmed is needed in the main text. How many oocytes were used per experiment?

Same lack of information for MDH1 experimental description.

In the kinase analyses, the authors interpret the changes in kinase phosphorylation with altered kinase activity. How is this statement supported? Either structural modeling or activity analyses should be performed or cited for support.

Phosphoproteome is frequently misspelled in Figure S1.

Referee #3:

With this report, Dr. Sun and collaborators have investigated the changes in protein phosphorylation taking place during mouse oocyte maturation at the genome-wide level. Using high resolution mass spectrometry-based phosphoproteomics, they identify more than 8000 phosphosites, most of which fluctuate during oocyte maturation. The authors have identified several new and never described before phosphorylation sites in proteins critical for oocyte maturation and have explored their function. They use dominant negative experiments with phosphorylation-mutant proteins to investigate the functional significance of these phosphorylation sites. In addition, they have used a pharmacological approach to define the function of novel kinases identified in their phosphoproteomes.

The study provides a comprehensive database of phosphorylations during oocyte maturation. Undoubtedly, this dataset is an important contribution to the field of meiotic maturation. However, the enthusiasm for the study is tempered by several oversights, by the lack of robust controls in the experiments using phospho-mutant proteins, and the poorly controlled use of inhibitors to investigate the function in oocyte maturation of the novel kinases they have identified.

1. The authors state several times that the pivotal points in oocyte maturation are GV, GVBD and MII. Certainly, GVBD may be a useful transition to study because changes in phosphorylation occur with minimal changes in protein levels. GVBD does not require protein synthesis in the mouse. However, one may point that metaphase one maybe equally important in oocyte maturation. Critical changes in phosphoproteome are likely necessary for chromosome condensation and trafficking and for spindle assembly and function, changes not detected or obscured by doing measurements only in MII. The authors may want to better discuss the rationale for their choices of maturation times. Moreover, are 3 hrs after hCG injection sufficient to produce 100% GVBD? Have the authors discarded oocytes that have not gone GVBD? This should be specified in the methods.
2. Page 5, line 129. The paper cited to document Mapk and CDk1 phosphorylations may not be correct.
3. Page 5/6 line 144-151. Here the authors state that the phosphoproteome changes are much larger than the changes in proteome during oocyte maturation. Inspection of the proteome data shows significant changes for about 900 proteins. This number is considerably lower than the number predicted by the changes in translation of mRNAs during oocyte maturation (see PMID: 35697785; PMID: 31970406). More in depth inspection of the levels of candidate proteins well know to change during oocyte maturation show some discrepancies. In the included proteome, CcnB1 increases 2.1 fold, whereas WB of numerous publications show up to 10 fold changes in protein level, changes consistent with the large increase in CcnB1 mRNA translation; Cdc20 levels in the proteome increase 2.4fold, whereas 10fold increase have been commonly detected by WB; Tex19 levels do not change from the proteome data, whereas large fold changes in protein levels and mRNA translation are reported in the literature. Even proteins known to undergo major decrease during maturation including Zar1, Zp1, Zp2, Zp3, show more moderate decline in the proteome or called unchanged. Surprisingly, their protein level changes also are considerably lower than in the proteome published in 2023 by the authors (Fig.S1 of PMID: 36496143). This raises the issue of whether the normalization applied to the raw data may have affected the calculations. Have the authors tried to apply different normalization strategies on their raw data? How do they explain the discrepancies with their previous report? Since the raw phosphoproteome data are corrected for the protein abundance, would the data be different if they used their previous proteome for normalization? Perhaps, the normalization strategies used here should be more thoroughly justified. Given these uncertainties, it is not clear whether the statement that phosphoproteome changes are much larger than proteome is correct. The statement should be tempered down.
4. Page 7 Line 209. YBX2/MSY2 phosphorylations and their functions have been described by R. Schultz and collaborators (PMID: 18606161).
5. Page 8. In the experiments where the authors express wild type and mutant BTG4, they see a considerable decrease of the endogenous BTG4 (supplementary Fig.S5). Is this a consistent finding? If yes, the authors should exclude that the mRNA stabilization phenotype they see is not due to the decrease in the endogenous Btg4. More in general, all the experiments overexpressing a phospho-mutant protein should be better controlled by assessing the effects on the endogenous protein levels.
6. Page 8 line 217. The authors state that the BTG4 phosphorylation sites are conserved across species. However, in human and chimpanzee there is a cysteine flanking the sequence. Would disulfide bridges hinder phosphorylation at these sites? Please, discuss. Moreover, it is not mentioned whether these phosphorylated sites are found also in the *Xenopus* oocyte phosphoproteome the authors have analyzed.
7. Page 10, line 301. The statement: "whether and how ribosomal protein phosphorylation influence oocyte maturation remain to be explored" should be toned down. RPS6 phosphorylation and S6kinase activity in the oocyte have been reported.
8. Page 11, line 350 Kinase inhibitor studies. A quick search on the web indicates that the reported CSNK1-IN-1 IC50 is 21 uM. However, the authors see a maximum effect already at 0.5 uM (Fig S11). An IC50 of 32 nM is reported for PF-4800567. However, the authors see effects only in the uM range. These raises concern of non-specific effects affecting the measurements. It is also of concern that they use oil in the oocyte incubation; this condition may have a major effect on hydrophobic compounds. The IC50 for all these drugs should be included in Fig.S11 and Fig.6 G. The concentration used to monitor polar body extrusion should be included in the figure or in the figure legend.
9. Page 14, 444. The authors state that there is the comprehensive description of the dynamics of phosphorylation in mammalian oocytes. However, a comparison of phosphorylation in GV and MI has been reported for mouse oocytes (PMID: 36264786). This publication should be mentioned here.

Referee #4:

In the current study, Qiang Wang and colleagues present a detailed quantitative phosphoproteomic analysis of oocytes across three crucial maturation stages. Additionally, they conduct a functional examination of specific phosphosites, chosen due to their specificity and conservation. The identification of kinases also expands our understanding of the mechanism underlying oocyte maturation. While the article is generally well-written and the figures are of high quality, some of the legends could benefit from further elaboration.

Overall, this study offers valuable insights into the phosphorylation landscape of mouse oocytes. The findings are intriguing, and the experiments are well-designed. However, there are a few issues that need to be addressed to enhance the strength of this manuscript.

1. This study utilized a large number of mouse oocytes, resulting in good reproducibility of the data. Therefore, three replicates should be sufficient to meet statistical requirements. The rationale behind the decision to conduct phosphoproteomics profiling on 5 replicates rather than 3 should be clarified.
2. The author conducted phosphoproteomic studies on oocytes during the GV, GVBD, and MII stages. Why were oocytes at the MI stage not collected?
3. In line 200, the author mentions numerous phosphorylation sites located on proteins specifically expressed in oocytes. What evidence does the author have to define these oocyte-specific proteins?
4. In the kinase assay section, what criteria guided the selection of these specific 13 kinases for inhibitor assays? Furthermore, how was the potential toxicity of the inhibitors on oocyte maturation mitigated?
5. In metabolomic studies, the reproducibility is not as robust as in proteomic studies. How to interpret and address this issue.
6. immunofluorescence: how many oocytes were pooled, and from how many mice?
7. What's the rationale behind selecting 70% as the criterion for conservation analysis?

Dear Qiang,

Thank you again for submitting your phosphoproteomic analysis of mouse oocytes for consideration as a Resource Article in The EMBO Journal. I sent it to four referees with expertise in proteomics (referees 1 and 2) and mammalian oocyte maturation (referees 3 and 4), who have now provided the reports copied below. All referees appreciate the comprehensiveness and the potential value of the presented dataset. However, they also raise various presentational and experimental concerns that would need to be clarified prior to publication.

Should you be able to adequately and comprehensively address these concerns, we would be interested in pursuing a revised version further for EMBO Journal publication. Key aspects will be clarifying more experimental details, oocyte synchronization, phosphosite normalization to protein levels, and inhibitor concentrations. On the other hand, obtaining further or replicate datasets that would require additional large quantities of oocytes would not appear to be justified for the purpose of this revision. Since it is our policy to allow only a single round of major revision, it will however be important to carefully respond to all points at the time of resubmission. Should this require more time than our default three-months revision period, we would be happy to offer an extension, during which our 'scooping protection' (meaning that competing work appearing elsewhere in the meantime will not affect our considerations of your study) would of course remain valid. Also, please do not hesitate to contact us in case you should want to discuss any specific points related to the referee reports and your revision.

Further information on preparing and uploading a revised manuscript can be found below and in our Guide to Authors. Thank you again for the opportunity to consider this work for The EMBO Journal, and I look forward to your revision.

With kind regards,

Hartmut

1) Every manuscript requires a Data Availability section (even if only stating that no deposited datasets are included). Primary datasets or computer code produced in the current study have to be deposited in appropriate public repositories prior to resubmission, and reviewer access details provided in case that public access is not yet allowed.

Further

information:

embopress.org/page/journal/14602075/authorguide#dataavailability

9) Digital image enhancement is acceptable practice, as long as it accurately represents the original data and conforms to community standards. If a figure has been subjected

to significant electronic manipulation, this must be clearly noted in the figure legend and/or the 'Materials and Methods' section. The editors reserve the right to request original versions of figures and the original images that were used to assemble the figure. Finally, we generally encourage uploading of numerical as well as gel/blot image source data; for details see: embopress.org/page/journal/14602075/authorguide#sourcedata

At EMBO Press, we ask authors to provide source data for the main manuscript figures. Our source data coordinator will contact you to discuss which figure panels we would need source data for and will also provide you with helpful tips on how to upload and organize the files.

Further information is available in our Guide For Authors:

In the interest of ensuring the conceptual advance provided by the work, we recommend submitting a revision within 3 months (21st Aug 2024). Please discuss the revision progress ahead of this time with the editor if you require more time to complete the revisions. Use the link below to submit your revision:

<https://emboj.msubmit.net/cgi-bin/main.plex?el=A6li6BCJE4A5BjGb2I7A9ftdRY9aMdXzPkAvouJd03vO6AY>

Referee #1:

The manuscript by Sun et al. provides a comprehensive phosphoproteomic analysis of mouse oocyte maturation by monitoring three stages. For this analysis, 30,000 oocytes were arrested at GV, GVBD, and MII stages, with 10,000 oocytes per stage divided into five replicates. Through phosphopeptide enrichment and TMT labeling, about 8000 phosphopeptides were quantified across different meiotic stages. The authors identified multiple novel sites and examined corresponding kinases and phosphatases. In addition, by using small molecule inhibitors, the role of 13 previously uncharacterized kinases in oocyte maturation was explored. Specific roles of three different phosphoproteins in oocytes were also explored using site-directed mutagenesis. Each subsequent experiment on local regulation provides descriptive

results but lacks detailed molecular analysis and some controls. In general, this manuscript is rich in data and will be a valuable resource for the scientific community. However, before publication, the authors should address the following concerns:

Response: Thank you for the supportive comments.

Major Concerns:

1. One of the main purposes of the manuscript is to differentiate the phosphoproteomic profiling of three key stages of oocyte meiosis. However, the manuscript lacks information regarding the level of synchronization achieved at specific stages. Western blotting and microscopy should be performed to assess each stage and the effectiveness of arresting at a given stage, as well as the homogeneity of the populations. Simply showing the ratios of four known phosphosites is not enough to assess the pooled populations.

Response: Thank you for the comment. Three stages of oocytes can be directly distinguished based on their morphology, as shown in Author Response Figure 1. GV oocytes exhibit the germinal vesicle (arrow), while GVBD oocytes lack this structure after germinal vesicle breakdown; Typical MII oocytes display the first polar body (arrowhead). Since these oocytes were picked one by one under microscope at specific time points, we are pretty sure the homogeneity of the populations at each stage. Additionally, four known phosphorylation sites (Fig. 1C and 1D) were presented to demonstrate the reliability of data quality and good intra-group reproducibility.

Author Response Figure 1

2. For defining each stage selective phosphorylation sites, each phosphosite needs to be normalized to its protein level. Otherwise, it is not clear if the regulation is at phosphorylation or protein level. Are the values in Figure 2 normalized to their protein level? Those values should also be reported in the Supplementary Table.

Response: We appreciate the reviewer's valuable comments. The phosphosite differences presented in Fig. 2 have indeed been normalized to their respective protein levels using our proteomics data. We have included these normalized values (phosphosite value, protein value and normalization value) in the Dataset EV1.

3. *The experimental setup is not well explained at the beginning of the result section, some information such as the number of mouse/oocytes/replicates are in the discussion but not in the result part. Those details should be included in the result part or Figure 1A.*

Response: Thank you for the suggestion. We have included the number of oocytes and replicates used for each group in new Fig.1A, and described in the new version of the manuscript (line 108-113). Additionally, the number of mice used in the study is detailed in the Materials and Methods section (lines 575-580).

“Here, we isolated a substantial quantity of mouse oocytes at three pivotal time points during meiotic maturation (arrested GV stage, meiotic resumption GVBD stage, and ovulated MII oocyte; 10,000 oocytes for each stage with 5 replicates), and obtained the intracellular phosphoproteome utilizing multiplex tandem mass tag (TMT) labelling coupled with liquid chromatography–mass spectrometry (LC–MS) (Fig. 1A).” (line 108-113)

4. *In myc-BTG4 wt vs myc-BTG4 tm expressing cells (Figure S5C), it appears that myc-BTG4 tm expressing cells have lower levels of endogenous BTG4 expression. Is it possible that the triple mutant is acting as dominant negative, leading to a depletion of endogenous BTG4 and causing the observed phenotype? This may be an alternative explanation to the proposed model. Additionally, the western blot in Figure S5 may not be sufficient to conclude that BTG4 tm disrupts the endogenous interaction with PABNL1, as there still appears to be an interaction that is not disrupted. Also, the interactions are shown by exogenously expressed proteins, not the endogenous ones. This experiment requires quantification over replicates. Furthermore, it would be helpful to address the potential kinase and its presence among the identified potential kinases at oocyte maturation.*

Response: We totally agree with the reviewer’s comments. Based on our data (Appendix Figure S3C), it seemed that both exogenous myc-BTG4 wt and myc-BTG4 tm potentially reduce the accumulation of endogenous BTG4. However, the exogenous myc-BTG4 wt still mediates the degradation of maternal transcripts and interacts with PABPN1L (Fig. 4G-4L and Appendix Figure S3F). In contrast, the exogenous myc-BTG4 tm was able to block this process (Fig. 4G-4L and Appendix Figure S3F). Thus, these observations promoted us to believe that the reduction of endogenous BTG4 in these two groups of oocytes has little impact on the function of exogenous BTG4. According to the reviewer's suggestion, we have toned down the conclusion in line 243 as follows: “Co-immunoprecipitation assay further corroborated that phospho-deficient mutant of BTG4 attenuates its interaction with PABPN1L.” We have included the quantification analysis of the experimental replicates, as shown in new Appendix Figure S3F.

It is important to identify the potential kinases of BTG4 in oocytes, as the reviewer indicated. In the present study, we just focus the function of phosphorylation sites on BTG4; an ongoing project in the lab is to search for the relevant kinases. Thank you again for this great suggestion!

5. In Fig. S9E, is the phosphorylation normalized to the protein level? Is there a prediction for the responsible kinase? In this experiment, WT and phosphomutant RPL12 proteins are overexpressed in oocytes and the protein profile is monitored. How many proteins are identified here, and how are their distribution? Having disproportional up vs down-regulation may not be solely due to enhanced translation. There could be different explanations for this. More controlled experiments need to be done. In addition, "phosphorylation of ribosomal protein RPL12-S38 is critical for maintaining translational homeostasis in mammalian oocytes (Fig. S9K)" is very strong conclusion for this. All experiments are on the WT background the phenotype of only RPL12 phosphomutant expression cells is not examined and the experiment performed here does not address translational homeostasis.

Response: We thank the reviewer for the insightful comments. The phosphorylation levels presented in Fig. S9E (Fig. EV3E) were indeed normalized to their corresponding protein levels. We did not conduct the kinase-related experiments in this section. Here, we try to predict the potential kinases for S38 site on RPL12 using PhosphoSitePlus and GPS5, and CDK1 and CDK3 may be the responsible kinases (Author Response Figure 2). An ongoing project in the lab is to identify the kinases responsible for the phosphorylation of RPL12-S38 in mammalian oocytes.

We identified a total of 4,540 proteins, with 197 showing differential accumulation (168 up and 29 down) between Rpl12-mut and Rpl12-wt. The relevant information has been included in Dataset EV17 and described in the revised manuscript (line 313).

According to the reviewer's comments, we have toned down our conclusion in the revised manuscript (line 317-320). "Together, these observations suggest that ribosomal protein RPL12 (S38) phosphorylation does not globally impact translation during meiosis, but may regulate the expression of specific subsets of mRNAs." Correspondingly, we have removed the original Fig. S9K (Figure EV3), as it seems to be overinterpreting the data and might mislead the readers.

Author Response Figure 2

6. *The manuscript is focused on the regulated kinases, meaning the phosphosites of kinases are dynamic in oocytes. However, it would be more interesting to find out the responsible kinases that are active at particular stages and phosphorylate multiple substrates. Is it possible to distinguish kinase activities at different stages?*

By using inhibitors, multiple kinases are inhibited and all of them affect the Polar body1 extrusion rate. However, in multiple cases, the concentration of inhibitors is very high (around 10 μ M-100 μ M). This brings the off-target effect issue. What other potential kinases are inhibited at this rate, and how specific are the treatments to a particular kinase?

Response: We are appreciative of this comment. In the current study, we first depicted the dynamic changes in phosphorylation sites during maturation based on different clusters (Figure 2). Starting from this clue, we further analyzed the kinases potentially responsible for these dynamic phosphosites (Figure EV5). For example, we discovered not only the stage-specific kinases (Cluster 1 corresponds to GV kinases, Cluster 2 corresponds to GVBD kinases, and Cluster 3 corresponds to MII kinases), but also other kinases enriched in two stages (Clusters 4 and 5). On the other hand, kinases identification based on clusters can reflect the dynamic changes in their activity.

Mouse oocytes have unique cellular structures and compositions that may affect the penetration and efficacy of inhibitors. Higher concentrations are often necessary to achieve effective inhibition within these cells (Su, Denegre et al., 2003, Swain, Ding et al., 2008, Wei, Greaney et al., 2018, Zhou & Homer, 2022). For instance, 15 μ M

U0126 (MAPK inhibitor; IC50 72 nM) is required to sufficiently inhibit MAPK activity in mouse oocytes (Miyagaki, Kanemori et al., 2014, Tong, Fan et al., 2003). In our study, different concentrations were tested for each inhibitor before use. The better approach to determine the functions of these kinases is to use the genetic mouse model (i.e. knockout or knockin mice), which is a significant amount of work beyond the scope of current study. In future research, we will construct mouse model with specific point-mutations to elucidate the regulatory mechanisms of these kinases in meiosis.

As the reviewer suggested, we have included a description about the limitations of this study in the Discussion section of revised manuscript (line 548-553). For example:

“Various inhibitors (e.g., CSNK1-IN-1 and WNK-IN-1) were utilized to evaluate the effects of kinase inhibition on oocyte maturation in current study. Due to the potential off-target effects of these compounds, a cleaner way, such as knockdown or knockout targeting the specific kinase, is crucial for clarifying this question.”

7. The authors state multiple times that the rate of Tyrosine phosphorylations is significantly lower than in other tissues suggesting different activities of Tyrosine kinase in somatic versus germ cells. However, the difference could be solely due to technical reasons of having a low amount of protein starting material or applying different enrichment techniques. A similar amount of somatic and germ cells should be analyzed in parallel by either performing western blotting with a pan phospho-tyrosine antibody or the same phospho-enrichment protocol to conclude.

Response: Thank you for the insightful comments. Our collaborator, Xuejiang Guo et al., established the phosphoproteome of mouse spermatogonia using the same mass spectrometry platform (Li, Chen et al., 2022). They profiled the large-scale phosphoproteome of primary spermatocytes (which are easily accessible and abundant in mice), and found that the proportion of tyrosine phosphosites was 0.19%. In a recent study, researchers detected a proportion of 0.57% tyrosine phosphosites using 1,500 mouse oocytes (Cheng, Altmeppen et al., 2022). These observations together indicate that the sample input amount appears not substantially affect the proportion of tyrosine phosphosites; and the number of tyrosine phosphosites in mouse germ cells (spermatocytes or oocytes) is likely lower than somatic cells. According to the reviewer’s suggestion, the related conclusion has been toned down in the revised manuscript (Line 124-126).

“The phosphotyrosine abundance was relatively diminished in oocytes compared to mouse organs (0.5% vs. 2.5%) (Giansanti, Samaras et al., 2022, Huttlin, Jedrychowski et al., 2010), indicating that the proportion of tyrosine phosphorylation in germ cells

may be lower than that in somatic cells.”

8. *Similar to kinases, potential functional phosphatases are also suggested based on their phosphorylation dynamics, however, this may not be the case.*

Response: Thank you for the suggestion. We did not conduct functional validation for phosphatases for the following reasons: (1). Compared to kinases, phosphatases act on a wide range of substrates that are not very specific; and therefore, they may modulate various biological events. (2). The phosphoproteomics revealed the majority of proteins with elevated phosphorylation levels (Fig. 2E), indicating the primary involvement of kinases in this process. (3). There are fewer commercially available inhibitors for phosphatases (i.e. MTMR2/3/4/7, DUSP7 and PTPN13/21/23). (4). We are currently conducting the large-scale phosphosite mutation studies to investigate the functions of phosphatases and kinases in oocytes.

9. *In the discussion section, it is mentioned that 200 µg of protein starting material is required for 10,000 phosphosites (Humphrey, 2018). However, due in part to upgraded mass spectrometry and acquisition modes, better higher number of phosphopeptides can be determined with much less starting material (ie. Bortel et. al, 2024, around 10ug, 15k sites). It may be better to be more specific when defining the limit of detection, as this has been changing due to recent advancements.*

Response: Thank you for the suggestion. The sample input amount was determined based on the literature published in 2018 (Humphrey, 2018) when we start the project. In the revised manuscript, a brief description has been included in the Discussion section as follows: “With the advancements in mass spectrometry and acquisition modes (Bortel, Piga et al., 2024), employing minute samples or even single-cell phosphoproteomics could substantially reduce the number of oocytes required for similar investigations.”(line 545-548).

Minor Concerns:

1. *Supplementary Table 1 should be in xlsx format. The current version of this table is not readable. It is not clear why some tables are in CVS format*

Response: As suggested, we have now converted supplementary table 1 to an xlsx format. Furthermore, we have carefully reviewed the formatting of all tables and ensured they are appropriately presented.

2. *Phosphopeptide enrichment should be included in the method subtitles*

Response: As suggested, we have included the “Phosphopeptide enrichment” in the method subtitles (line 586).

3. In Figure 2F right part of the graph needs a revision, those graphs are not clear

Response: As suggested, we have revised the right part of the graph (new Fig. 2F).

4. Why the phosphopeptide numbers are lower than phosphosites? Does it mean that there are not many multiply phosphorylated peptides.

Response: From the oocyte phosphoproteome, we identified a total of 8,090 phosphosites, corresponding to 8,725 phosphopeptides. Actually, the phosphopeptide number is larger than phosphosites.

5. Why Y97 is standing in Figure 3E, what are its measurements?

Response: The phosphorylation dynamics of UHRF1-S18 and Y97 are quite consistent, resulting in two nearly overlapping curves (Author Response Figure 3); so, we place Y97 standing in the figure

6. Figure 3K should be revised to a more informative representation, only the number of phosphosites without their sequence, structure, or cell stage ratios are not useful.

Response: Thank you for the suggestion. This figure was designed to highlight the presence of numerous novel phosphosites in SCMC. As the reviewer indicated, we have included the information on the phosphorylation sites, peptide sequences, and temporal changes of the SCMC complex in a supplementary table (Dataset EV7, sheet2).

7. Across the manuscript there are overstatements, like

Line 263: On the other hand, we also discovered some fully conserved phosphosites across four species, and which have never been reported in oocytes (i.e., HDAC2-S422/S424, NUP98-S888, PDCD4-S457) (Fig. 5C-5E), implying the existence of evolutionarily conserved phosphorylation mechanisms controlling critical events in oocytes. Unless some functional assays are done it is not possible to predict that they are controlling critical events. Those comments should be revised.

Response: Thank you for the valuable comment. We have revised the relevant statement in the new version of the manuscript as suggested. For example:

“On the other hand, we also discovered some fully conserved phosphosites across four species, which have never been reported in oocytes (i.e., HDAC2-S422/S424, NUP98-S888, PDCD4-S457) (Fig. 5C-5E). These findings suggest such an evolutionarily conserved phosphorylation mechanism may be involved in the critical events in oocytes.” (line 263-266)

8. *There are typos like "Phosphorproteome" in the legends or figures.*

Response: Thank you for pointing this out. The typos "Phosphorproteome" to "Phosphoproteome" have been corrected in the revised version (new Figure. EV1D, EV1F, EV1G).

9. *In S9 K a single phospho-site mutation, does not make a protein phosphodead. It should be corrected.*

Response: As addressed in major concern 5, Figure S9K has been removed in the new version of the manuscript.

Referee #2:

Sun et al perform proteomic and phosphoproteomic analyses of isolated mouse oocytes at the germinal vesicle (GV) stage, upon the breakdown of the GV upon meiotic resumption, and in meiosis II. This is an interesting analysis of meiosis by proteomic and phosphoproteomics with extensive follow-up analysis of regulated phosphorylation sites.

Phosphorylation is an essential regulatory mechanism governing meiotic progression. Previous studies have investigated protein abundance and phosphorylation changes in organisms producing large numbers of oocytes. However, similar studies in mice are hindered by the limited number of oocytes per animal at the time of birth. The authors overcome this by using a very large number of oocytes.

1. *The authors isolate 30,000 oocytes. It would be helpful for the reader to have a sense of how many oocytes are isolated from each mouse and how many mice were used for each replicate. Images of the oocytes to demonstrate synchronicity should be included. The main text should also state how much protein was isolated for each replicate.*

Response: Thank you for the constructive comments. We have included detailed information on the number of oocytes used for each replicate in new Fig. 1A, and the

number of mice used has been added in the Materials and Methods section (line 572). On average, approximately 32 oocytes can be collected from each mouse. Three stages of oocytes can be directly distinguished based on their morphology. GV oocytes exhibit the germinal vesicle (arrow), while GVBD oocytes lack this structure after germinal vesicle breakdown; typical MII oocytes display the first polar body (arrowhead), see Author Response Figure 4 below. These representative images have been incorporated in new Fig. 1A. Oocytes were picked one by one under microscope at specific time points, we are pretty sure the homogeneity of the populations at each stage. In addition, the protein content was assessed in our preliminary experiments, approximately around 8 ng per mouse oocyte. In the Discussion section, we have mentioned the total protein content used in the present study (240 μ g, line 537).

Author Response Figure 4

2. *Did the authors consider including an isobaric carrier? The description of the proteomic and phosphoproteomic methods needs clarification. The authors used 2,000 oocytes per replicate, which they lysed, digested, labeled, and HP-RP fractionated into 30 fractions. Next, the authors discuss the phosphoproteomic analysis and state that labeled peptides were fractions into fractions before enrichment. How was this mixture generated? Also, from 2,000 oocytes/replicate? The workflow in Figure 1A is inconsistent with the method description and should be corrected. Also, the number of replicates should be indicated and the labeling scheme. A more detailed workflow as supplemental data would be helpful.*

Response: Thank you for the valuable comments. We did not use the isobaric carrier in the present study. The carrier strategy has been applied in ultrasensitive mass spectrometry analysis of very small samples, such as single cells (Cheung, Lee et al., 2021). Here, we used 2,000 oocytes per replicate, with a total of five replicates for each of the GV, GVBD, and MII groups. Each biological replicate underwent protein extraction, trypsin digestion, and TMT labeling. After equal mixing, one-tenth of this mixture was used for quantitative proteomic analysis, while the remaining 90% was used for quantitative phosphoproteomic analysis. We have provided a more detailed workflow (Figure EV1A) in the new version of the manuscript, as the reviewer suggested.

3. The authors normalize phospho-site to protein abundances. However, the site was still included in further analyses if the protein was not detected or quantified. Please clarify. If yes, how are corrected and non-corrected sites distinguished in subsequent analyses? How many sites were corrected versus not?

Response: Thank you for these insightful comments. We normalize the intensity of phosphosites to the corresponding protein abundance when the protein is detected and quantified. If the corresponding protein could not be quantified by proteomics analysis (515 out of 8,090 proteins), the phosphorylation levels of these sites were directly analyzed (Protein Identification and Quantification, line 675), as previously reported (Fan, Cheng et al., 2020).

In this study, approximately 93.6% (7,575/8,090) of the phosphorylation sites ratios could be calibrated by protein levels, allowing differential phosphorylation to be distinguished from altered protein expression (Wu, Dephoure et al., 2011). Due to the scarcity of oocyte samples, the identification of the phosphoproteome is quite challenging. Therefore, this study preserved all phosphorylation site information as much as possible (Figure.1A). Only about 6.4% (515 out of 8,090) of the sites did not have corresponding protein identified.

Overall, these sites have no evident effects on the analysis of phosphorylation date (Author Response Figure 5). Therefore, we did not distinguish the corrected and non-corrected phosphorylation sites in subsequent analyses. However, this information is included in dataset EV1.

4. A heatmap depicting proteome changes should be included as supplemental data. The accounting of phosphoproteins and proteins is confusing. Most studies find that the proteome is stable during meiotic progression. How do the authors explain this difference? Are specific proteins stable in some organisms but not in mouse?

Response: Thank you for the insightful comments. According to the suggestion, we have added a heatmap depicting proteome changes in new Figure EV2F. Regarding

oocyte proteome, we and another group identified a significant number of differentially-expressed proteins during mouse oocyte maturation (DEPs) (Sun, Sun et al., 2023, Wang, Kou et al., 2010). Similarly, Kronja et al. found that the proteome undergoes substantial remodeling during *Drosophila* oocyte development. Approximately 30% of detectable proteins experienced the changes in abundance (1,048 DEPs) (Kronja, Whitfield et al., 2014). In striking contrast, here, we found that the dynamic changes in the phosphoproteome were more pronounced relative to the proteome (6,130 regulated phosphosites on 2,204 proteins vs. 929 DEPs, Figure EV2G).

5. *The authors state in line 153: "These findings unveil that phosphorylation modification is a new layer of molecular control for oocyte maturation, complementing the proteome landscape we discovered previously". However, the highly dynamic changes in the phosphoproteome during meiosis are consistent with studies in other organisms. This should be rephrased.*

Response: Thank you for the suggestion. What we meant to say is that, compared to the proteome, phosphorylation participates in the regulation of mouse oocyte maturation at a different layer. As reviewer suggested, we have rephrased the sentence in the revised manuscript: "These findings unveil that phosphorylation modification serves as an additional layer of molecular control for oocyte maturation, complementing the proteome landscape we discovered previously" (line 155).

6. *About the comparison with the Gygi dataset, where the 1,842 phosphorylation sites not detected in the Gygi dataset on proteins detected in their dataset indicate oocyte-specific regulation or where the proteins not detected? This should be clarified.*

Response: Thank you for the comment. In this study, we did not directly compare the proteomes between Gygi dataset and ours. Of 1,842 novel phosphosites, 270 sites are located on oocyte-specific proteins (Author Response Figure 6); and 1,572 sites belong to the proteins that are also widely expressed in somatic cells (Wang & Nishida, 2015), assuming that most of these proteins can be found in Gygi dataset. Therefore, our results suggest the site-specific phosphorylation in mouse oocytes, rather than protein-specific phosphorylation.

Author Response Figure 6

7. *BTG4 phosphorylation sites. Are these sites individually detected or as double and triple phosphorylation sites? Figure 4A makes it look like individual sites. Are these sites localized on the phosphopeptide? The authors generated a BTG4 3A mutant and did RNA-seq and PAT. An additional description of how the constructs were expressed and how their expression was confirmed is needed in the main text. How many oocytes were used per experiment? Same lack of information for MDH1 experimental description.*

Response: Thank you for the suggestions. The three sites localize on the same phosphopeptide (-ATGDCSSGTSSDEESCSR-). Mass spectrometry data showed that they are individually detected (S146/S147) or double detected (T145 and S146, T145 and S147, S146 and S147). To explore the potential function of BTG4 phosphorylation, three phosphosites were simultaneously mutated in current study. To show the phosphorylation dynamics of each site during maturation, they were presented individually in Fig.4A based on the quantification data of phosphosites calculated by MaxQuant.

BTG4 triple mutant was generated by substituting the three serine/threonine residues with alanine. The *in vitro* synthesized mRNA was expressed in oocytes using microinjection techniques (see Experimental Procedures, Overexpression Analysis section, line 760-770). The site-mutant constructs (BTG4 and MDH1) were confirmed by Sanger sequencing (Author Response Figure 7), and their expression was validated by Western blotting (Figures 5J, Figure EV3 and Appendix Figure S3B). The relevant information has been incorporated in the new version of the manuscript as suggested (line 771-778).

Author Response Figure 7

8. In the kinase analyses, the authors interpret the changes in kinase phosphorylation with altered kinase activity. How is this statement supported? Either structural modeling or activity analyses should be performed or cited for support.

Response: Thank you for the insightful comment. We did not perform structural modeling or activity analyses in this study. According to the reviewer's suggestion, in the revised version, we have included three relevant literatures that supports the potential link between phosphorylation changes and kinase activity (Aerts, Craessaerts et al., 2015, Chong, Lee et al., 2001, Johnson, Noble et al., 1996).

"Previous studies have shown that phosphorylation can significantly impact kinase function (Aerts, Craessaerts et al., 2015, Chong, Lee et al., 2001, Johnson, Noble et al., 1996). In the kinase analyses, we observed changes in kinase phosphorylation, which may suggest altered kinase activity." (line 353-356)

9. Phosphoproteome is frequently misspelled in Figure S1.

Response: Thank you for pointing this out. It has been corrected throughout the manuscript.

Referee #3:

With this report, Dr. Sun and collaborators have investigated the changes in protein phosphorylation taking place during mouse oocyte maturation at the genome-wide level. Using high resolution mass spectrometry-based phosphoproteomics, they identify more than 8000 phosphosites, most of which fluctuate during oocyte maturation. The authors have identified several new and never described before phosphorylation sites in proteins critical for oocyte maturation and have explored their function. They use dominant negative experiments with phosphorylation-mutant proteins to investigate the functional significance of these phosphorylation sites. In addition, they have used a pharmacological approach to define the function of novel

kinases identified in their phosphoproteomes.

The study provides a comprehensive database of phosphorylations during oocyte maturation. Undoubtedly, this dataset is an important contribution to the field of meiotic maturation. However, the enthusiasm for the study is tempered by several oversights, by the lack of robust controls in the experiments using phospho-mutant proteins, and the poorly controlled use of inhibitors to investigate the function in oocyte maturation of the novel kinases they have identified.

Response: We thank the reviewer for the supportive and constructive comments.

1. The authors state several times that the pivotal points in oocyte maturation are GV, GVBD and MII. Certainly, GVBD may be a useful transition to study because changes in phosphorylation occur with minimal changes in protein levels. GVBD does not require protein synthesis in the mouse. However, one may point that metaphase one maybe equally important in oocyte maturation. Critical changes in phosphoproteome are likely necessary for chromosome condensation and trafficking and for spindle assembly and function, changes not detected or obscured by doing measurements only in MII. The authors may want to better discuss the rationale for their choices of maturation times. Moreover, are 3 hrs after hCG injection sufficient to produce 100% GVBD? Have the authors discarded oocytes that have not gone GVBD? This should be specified in the methods.

Response: Thank you for the insightful comments. We totally agree with the reviewer's comments that unravelling phosphoproteome in MI is important for understanding oocyte maturation. However, *in vivo* isolation of large number of MI oocytes is difficult. On one hand, following hormonal stimulation, approximately 7-8 hours later, some oocytes could develop to MI stage; and the rest of them are at pre-metaphase I or anaphase I stage (Edwards & Gates, 1959, Sanfins, Plancha et al., 2004), showing the low synchronization in this population. On the other hand, there is no way to select MI oocytes under microscope based on their morphology, unless through chromosome staining. Hence, we just focused on GV/GVBD/MI stage oocytes in the present study.

In addition, germinal vesicle breakdown occurs within 2-4 hours after hCG injection. Considering the time for ovary/oocyte isolation, we chose the 3-hour post-hCG to collect GVBD stage oocytes (Sanfins et al., 2004), and those oocytes that have not gone GVBD were discarded. The related description has been included in the Experimental Procedures (line 580). "When collecting GVBD oocytes, those that did not undergo germinal vesicle breakdown were discarded."

2. Page 5, line 129. The paper cited to document Mapk and CDk1 phosphorylations may not be correct.

Response: Thank you for pointing this out. We have changed the references in the revised manuscript (line 130).

“It has been well recognized that the active sites on MAPK1 (T183/Y185), MAPK3 (T203/Y205), and CDK1 (T161) undergo phosphorylation during oocyte meiosis, while inhibitory sites on CDK1 (T14/Y15) are dephosphorylated upon GVBD occurring (Adhikari & Liu, 2014, Lemonnier, Dupre et al., 2020, Tong, Fan et al., 2003)”

3. Page 5/6 line 144-151. Here the authors state that the phosphoproteome changes are much larger than the changes in proteome during oocyte maturation. Inspection of the proteome data shows significant changes for about 900 proteins. This number is considerably lower than the number predicted by the changes in translation of mRNAs during oocyte maturation (see PMID: 35697785; PMID: 31970406). More in depth inspection of the levels of candidate proteins well known to change during oocyte maturation show some discrepancies. In the included proteome, CcnB1 increases 2.1-fold, whereas WB of numerous publications show up to 10-fold changes in protein level, changes consistent with the large increase in CcnB1 mRNA translation; Cdc20 levels in the proteome increase 2.4fold, whereas 10fold increase have been commonly detected by WB; Tex19 levels do not change from the proteome data, whereas large fold changes in protein levels and mRNA translation are reported in the literature. Even proteins known to undergo major decrease during maturation including Zar1, Zp1, Zp2, Zp3, show more moderate decline in the proteome or called unchanged. Surprisingly, their protein level changes also are considerably lower than in the proteome published in 2023 by the authors (Fig.S1 of PMID: 36496143). This raises the issue of whether the normalization applied to the raw data may have affected the calculations. Have the authors tried to apply different normalization strategies on their raw data? How do they explain the discrepancies with their previous report? Since the raw phosphoproteome data are corrected for the protein abundance, would the data be different if they used their previous proteome for normalization? Perhaps, the normalization strategies used here should be more thoroughly justified. Given these uncertainties, it is not clear whether the statement that phosphoproteome changes are much larger than proteome is correct. The statement should be tempered down.

Response: Thank you for the insightful comments. The differences between mass spectrometry (MS) data and Ribo-seq data in oocytes and early embryos have also been noted by another group (Zhang, Ji et al., 2023). Their proteome data showed much fewer dynamically regulated genes than translatoome. It is attributed to the dominance of FGO-stockpiled proteins in the proteomes of early embryos. Besides, the following reasons may also explain such an inconsistency : (1). Although both

techniques aim to identify protein changes, Ribo-Seq measures the levels of ribosome-bound RNA (mRNA), whereas MS directly sequences protein peptides; (2). Proteins undergo various post-translational modifications (PTMs) that Ribo-Seq cannot detect. Protein modifications, stability, as well as degradation rate, significantly affect its abundance; (3). MS-based proteomics and Ribo-Seq have differential sensitivity. MS may miss low-abundance proteins due to the detection limits, while Ribo-Seq could detect actively-translated mRNAs even if the resultant proteins are in low abundance or rapidly degraded.

In addition, the discrepancy in protein level between proteomic data and WB may be explained by : (1). Phosphoproteomics and western blot are two distinct techniques, and these technical differences may lead to variations in the detected fold changes of proteins; (2). TMT (Tandem Mass Tag) strategy has a limitation of ratio compression because of peptide coelution during the LC-MS quantification (Rauniyar & Yates, 2014, Sun, Poudel et al., 2022). These factors could result in the lower fold changes for certain proteins. This phenomenon occurred not only in oocytes but also in somatic cells (Frese, Mikhaylova et al., 2017); and also, have been explained by other researchers (Mehta, Ahkami et al., 2022).

The expression level of each protein was normalized by dividing the mean expression level of the corresponding protein across samples (Protein Identification and Quantification, line 668-670), and this method was consistently applied in both proteomics analyses. Phosphoproteome calibration should be performed using the same batch of proteome data, as there are different biological variations between two studies, and the phosphoproteome and proteome in this study were generated from the same TMT-labeled mixture.

The main drawback of TMT quantification is ratio compression - a phenomenon that causes fold changes to appear compressed. The degree of compression for each protein is generally unknown, and the contributing factors are complex. The ratio compression degree in proteomics between the two studies may mainly be attributed to the various technical conditions. The number of biological replicates, the type of TMT reagents, and chromatographic separation conditions are different. For instance, the previous study (proteome conducted about 6 years ago) (PMID: 36496143), a 194-minute gradient was used to separate TMT-labeled peptides, whereas a 95-minute gradient was utilized here. In this study, using the same batch of samples and the same TMT-labeled mixture, we observed that both the number of differential phosphosites and the fold changes were greater compared to those in the proteome (Figure 2C and 2D, Figure EV2A and EV2C). The phenomenon, that the dynamic changes in phosphoproteome exceed those in the proteome, is not only observed in our study of oocytes but is also widely reported in various somatic cells (Bouhaddou, Memon et al.,

2020, Bruning, Noya et al., 2019, Robles, Humphrey et al., 2017, Wang, Ma et al., 2018). As the reviewer suggested, we have tempered the statement to: “The overall changes in phosphorylation status appeared to be of a larger magnitude than the changes at the protein level (GVBD/GV and MII/GV; Fig. 2C, 2D).” (line 152)

4. *Page 7 Line 209. YBX2/MSY2 phosphorylations and their functions have been described by R. Schultz and collaborators (PMID: 18606161).*

Response: Thank you for the comment. R. Schultz and collaborators have shown that CDC2A-mediated phosphorylation of MSY2 triggers maternal mRNA degradation during mouse oocyte maturation. The phosphosites of YBX2 identified in their study are not included in the novel phosphosites we discovered here. In the present study, only those novel phosphosites of YBX2 were presented.

5. *Page 8. In the experiments where the authors express wild type and mutant BTG4, they see a considerable decrease of the endogenous BTG4 (supplementary Fig.S5). Is this a consistent finding? If yes, the authors should exclude that the mRNA stabilization phenotype they see is not due to the decrease in the endogenous Btg4. More in general, all the experiments overexpressing a phospho-mutant protein should be better controlled by assessing the effects on the endogenous protein levels.*

Response: We totally agree with the reviewer’s comments. Based on our data (Appendix Figure S3C), it seemed that both exogenous myc-BTG4 wt and myc-BTG4 tm potentially reduce the accumulation of endogenous BTG4. Throughout the entire overexpression experiment, we used exogenously overexpressed BTG4 (wt) as the control group to evaluate the potential impact of the phosphorylation mutations. The exogenous myc-BTG4 wt still mediates the degradation of maternal transcripts and interacts with PABPN1L (Fig. 4G-4L and Appendix Figure S3F). In contrast, the exogenous myc-BTG4 tm was able to block this process (Fig. 4G-4L and Appendix Figure S3F). Thus, these observations promoted us to believe that the reduction of endogenous BTG4 in these two groups of oocytes has little impact on the function of exogenous BTG4.

6. *Page 8 line 217. The authors state that the BTG4 phosphorylation sites are conserved across species. However, in human and chimpanzee there is a cysteine flanking the sequence. Would disulfide bridges hinder phosphorylation at these sites? Please, discuss. Moreover, it is not mentioned whether these phosphorylated sites are found also in the Xenopus oocyte phosphoproteome the authors have analyzed.*

Response: Thank you for these valuable comments. To date, we are not sure whether disulfide bonds affect BTG4 phosphorylation modifications in humans and

chimpanzees. Structural analysis of BTG4 may be helpful for addressing this question.

Due to the low depth of phosphoproteome sequencing in *Xenopus* oocytes (3,129 phosphosites detected, Appendix Figure S4A) (Presler, Van Itallie et al., 2017), we did not find these phosphosites on *Xenopus* BTG4.

7. Page 10, line 301. The statement: "whether and how ribosomal protein phosphorylation influence oocyte maturation remain to be explored" should be toned down. RPS6 phosphorylation and S6kinase activity in the oocyte have been reported.

Response: As the reviewer suggested, this sentence has been rephrased in the revised manuscript.

"However, the role of ribosomal protein phosphorylation during oocyte maturation remains to be investigated." (line 302)

8. Page 11, line 350 Kinase inhibitor studies. A quick search on the web indicates that the reported CSNK1-IN-1 IC50 is 21 uM. However, the authors see a maximum effect already at 0.5 uM (Fig S11). An IC50 of 32 nM is reported for PF-4800567. However, the authors see effects only in the uM range. These raises concern of non-specific effects affecting the measurements. It is also of concern that they use oil in the oocyte incubation; this condition may have a major effect on hydrophobic compounds. The IC50 for all these drugs should be included in Fig.S11 and Fig.6 G. The concentration used to monitor polar body extrusion should be included in the figure or in the figure legend.

Response: Thank you for these great suggestions. Based on MCE website, CSNK1-IN-1 has inhibitory activity against CSNK1A1 in high ATP with IC50 values 1.5 nM. We speculate that the ATP concentration in oocytes might alter the IC50 value, making CSNK1A1 more sensitive to CSNK1-IN-1. Of course, this hypothesis needs to be tested in the future. In addition, we also noted that CSNK1-IN-1 effectively inhibited polar body extrusion at three concentrations (0.5, 2, and 5 μ M), indicating that CSNK1A may be sensitive to low-concentration inhibitors in the oocyte environment.

Oocytes have unique cellular structures and compositions that may affect the penetration and efficacy of inhibitors. Higher concentrations are often necessary to achieve effective inhibition within mammalian oocytes (Su et al., 2003, Swain et al., 2008, Wei et al., 2018, Zhou & Homer, 2022). For example, U0126, an inhibitor of MAPK as an example (IC50 72 nM), 15 μ M of U0126 is required to sufficiently inhibit MAPK activity (phosphorylation) in oocytes (Miyagaki et al., 2014, Tong et al., 2003). In addition, mineral oil has been widely used in oocyte/embryo *in vitro* culture, to protect the medium from evaporative effects. In the present study, Inhibitors were dissolved in appropriate solvents (DMSO) before the dilution to M16 medium. Hence,

we cannot completely rule out the possibility that mineral oil may have an effect on hydrophobic compounds.

As the reviewer indicated, we have incorporated a discussion about the potential off-target effects of inhibitors in the revised manuscript (line 548-552). Meanwhile, the IC50 values available for inhibitors were included in new Figure EV4, and the specific concentrations were included in new Fig. 6G.

9. Page 14, 444. The authors state that there is the comprehensive description of the dynamics of phosphorylation in mammalian oocytes. However, a comparison of phosphorylation in GV and MI has been reported for mouse oocytes (PMID: 36264786). This publication should be mentioned here.

Response: As the reviewer suggested, we have revised our statement in line 450.

“Our study, together with another work (Cheng, Altmepfen et al., 2022), provide a comprehensive description of the dynamics of phosphorylation in mammalian oocytes.”

Referee #4:

In the current study, Qiang Wang and colleagues present a detailed quantitative phosphoproteomic analysis of oocytes across three crucial maturation stages. Additionally, they conduct a functional examination of specific phosphosites, chosen due to their specificity and conservation. The identification of kinases also expands our understanding of the mechanism underlying oocyte maturation. While the article is generally well-written and the figures are of high quality, some of the legends could benefit from further elaboration.

Overall, this study offers valuable insights into the phosphorylation landscape of mouse oocytes. The findings are intriguing, and the experiments are well-designed. However, there are a few issues that need to be addressed to enhance the strength of this manuscript.

Response: We thank the reviewer for the supportive comments.

1. This study utilized a large number of mouse oocytes, resulting in good reproducibility of the data. Therefore, three replicates should be sufficient to meet statistical requirements. The rationale behind the decision to conduct phosphoproteomics profiling on 5 replicates rather than 3 should be clarified.

Response: Thank you for the valuable comment. Based on our preliminary experiments, the more oocytes we use, the more abundant the phosphosites

identified. 10,000 cells for each stage is almost the maximum number we can collect under our system. Therefore, to ensure the reproducibility and statistical validity, we decided to use five replicates, instead of three, in current study.

2. The author conducted phosphoproteomic studies on oocytes during the GV, GVBD, and MII stages. Why were oocytes at the MI stage not collected?

Response: Thank you for the insightful comment. *In vivo* isolation of large number of MI oocytes is difficult. On one hand, following hormonal stimulation, approximately 7-8 hours later, some oocytes could develop to MI stage; and the rest of them are at pre-metaphase I or anaphase I stage (Edwards & Gates, 1959, Sanfins et al., 2004). On the other hand, there is no way to select MI oocytes under microscope based on their morphology, unless through chromosome staining. Hence, we just focused on GV/GVBD/MI stage oocytes in the present study.

3. In line 200, the author mentions numerous phosphorylation sites located on proteins specifically expressed in oocytes. What evidence does the author have to define these oocyte-specific proteins?

Response: We primarily referred to REGULATOR: a database of metazoan transcription factors and maternal factors for developmental studies (Wang & Nishida, 2015). In this study, we highlighted those specific and well-defined maternal proteins (i.e. SCMC complex), instead of presenting all potential maternal factors.

4. In the kinase assay section, what criteria guided the selection of these specific 13 kinases for inhibitor assays? Furthermore, how was the potential toxicity of the inhibitors on oocyte maturation mitigated?

Response: Thank you for the insightful question. We selected kinases based on the following criteria: (1). kinases that have not been studied in oocytes; (2). kinases that have dynamically-regulated phosphorylation sites, as this indicates the potential changes in their activity; (3). we conducted a comparative analysis between regulated kinases and predicted kinases. Kinases that were present in both groups were of particular concern.

5. In metabolomic studies, the reproducibility is not as robust as in proteomic studies. How to interpret and address this issue

Response: Thank you for the constructive comment. In the present study, trace sample metabolomics was used to detect the metabolites in mouse oocytes. This technique has been developed in recent years and applied to the analysis of metabolic patterns in oocytes and early embryos as reported before (Li, Zhu et al., 2020, Zhao, Yao et al.,

2021). However, the reproducibility of data in trace metabolomics is relatively poor in comparison to proteomic data. Here, we performed technical replicates to exclude the variability arising from the instrument and experimental procedures. Rigorous normalization and statistical analysis were used to exclude the batch effects and to improve the reliability of the detected metabolites.

6. immunofluorescence: how many oocytes were pooled, and from how many mice?

Response: Thank you for the question. At least 100 oocytes from 3 mice for each group were evaluated, and the experiments were conducted three times. The relevant information has been included in figure legends (line 1231-1234).

7. What's the rationale behind selecting 70% as the criterion for conservation analysis?

Response: In this part of the analysis, the conservation of the amino acid sequence is greater than 69% (9/13). We apologize for the clerical errors in the manuscript. We have made the corrections in the revised manuscript (line 255, 752, 755). Given the large evolutionary distance among the species used for peptide conservation comparisons, there can be considerable variability in the peptide sequences. Therefore, we chose a very stringent screening criterion: of 13 amino acids, the central site (S/T/Y) is the same and phosphorylated, and at least 9 residues are identical (9/13, >69%). At least three species' phosphorylated peptide sequences with a similarity exceeding 69%, were included for further analysis.

References

- Aerts L, Craessaerts K, De Strooper B, Morais VA (2015) PINK1 kinase catalytic activity is regulated by phosphorylation on serines 228 and 402. *J Biol Chem* 290: 2798-811
- Bortel P, Piga I, Koenig C, Gerner C, Martinez-Val A, Olsen JV (2024) Systematic Optimization of Automated Phosphopeptide Enrichment for High-Sensitivity Phosphoproteomics. *Mol Cell Proteomics* 23: 100754
- Bouhaddou M, Memon D, Meyer B, White KM, Rezelj VV, Correa Marrero M, Polacco BJ, Melnyk JE, Ulferts S, Kaake RM, Batra J, Richards AL, Stevenson E, Gordon DE, Rojc A, Obernier K, Fabius JM, Soucheray M, Miorin L, Moreno E et al. (2020) The Global Phosphorylation Landscape of SARS-CoV-2 Infection. *Cell* 182: 685-712 e19
- Bruning F, Noya SB, Bange T, Koutsouli S, Rudolph JD, Tyagarajan SK, Cox J, Mann M, Brown SA, Robles MS (2019) Sleep-wake cycles drive daily dynamics of synaptic phosphorylation. *Science* 366
- Cheng S, Altmeppen G, So C, Welp LM, Penir S, Ruhwedel T, Menelaou K, Harasimov K, Stutzer A, Blayney M, Elder K, Mobius W, Urlaub H, Schuh M (2022) Mammalian oocytes store mRNAs in a mitochondria-associated membraneless compartment. *Science* 378: eabq4835
- Cheung TK, Lee CY, Bayer FP, McCoy A, Kuster B, Rose CM (2021) Defining the carrier proteome limit for

single-cell proteomics. *Nature Methods* 18: 76-+

Chong H, Lee J, Guan KL (2001) Positive and negative regulation of Raf kinase activity and function by phosphorylation. *EMBO J* 20: 3716-27

Edwards RG, Gates AH (1959) Timing of the stages of the maturation divisions, ovulation, fertilization and the first cleavage of eggs of adult mice treated with gonadotrophins. *J Endocrinol* 18: 292-304

Fan Y, Cheng Y, Li Y, Chen B, Wang Z, Wei T, Zhang H, Guo Y, Wang Q, Wei Y, Chen F, Sha J, Guo X, Wang L (2020) Phosphoproteomic Analysis of Neonatal Regenerative Myocardium Revealed Important Roles of Checkpoint Kinase 1 via Activating Mammalian Target of Rapamycin C1/Ribosomal Protein S6 Kinase b-1 Pathway. *Circulation* 141: 1554-1569

Frese CK, Mikhaylova M, Stucchi R, Gautier V, Liu Q, Mohammed S, Heck AJR, Altelaar AFM, Hoogenraad CC (2017) Quantitative Map of Proteome Dynamics during Neuronal Differentiation. *Cell Rep* 18: 1527-1542

Johnson LN, Noble ME, Owen DJ (1996) Active and inactive protein kinases: structural basis for regulation. *Cell* 85: 149-58

Kronja I, Whitfield ZJ, Yuan B, Dzeyk K, Kirkpatrick J, Krijgsveld J, Orr-Weaver TL (2014) Quantitative proteomics reveals the dynamics of protein changes during *Drosophila* oocyte maturation and the oocyte-to-embryo transition. *Proc Natl Acad Sci U S A* 111: 16023-8

Li H, Chen H, Zhang X, Qi Y, Wang B, Cui Y, Ren J, Zhao Y, Chen Y, Zhu T, Wang Y, Yao L, Guo Y, Zhu H, Li Y, Situ C, Guo X (2022) Global phosphoproteomic analysis identified key kinases regulating male meiosis in mouse. *Cell Mol Life Sci* 79: 467

Li L, Zhu S, Shu W, Guo Y, Guan Y, Zeng J, Wang H, Han L, Zhang J, Liu X, Li C, Hou X, Gao M, Ge J, Ren C, Zhang H, Schedl T, Guo X, Chen M, Wang Q (2020) Characterization of Metabolic Patterns in Mouse Oocytes during Meiotic Maturation. *Mol Cell* 80: 525-540 e9

Mehta D, Ahkami AH, Walley J, Xu SL, Uhrig RG (2022) The incongruity of validating quantitative proteomics using western blots. *Nat Plants* 8: 1320-1321

Miyagaki Y, Kanemori Y, Tanaka F, Baba T (2014) Possible role of p38 MAPK-MNK1-EMI2 cascade in metaphase-II arrest of mouse oocytes. *Biol Reprod* 91: 45

Presler M, Van Itallie E, Klein AM, Kunz R, Coughlin ML, Peshkin L, Gygi SP, Wuhr M, Kirschner MW (2017) Proteomics of phosphorylation and protein dynamics during fertilization and meiotic exit in the *Xenopus* egg. *Proc Natl Acad Sci U S A* 114: E10838-E10847

Rauniyar N, Yates JR (2014) Isobaric Labeling-Based Relative Quantification in Shotgun Proteomics. *Journal of Proteome Research* 13: 5293-5309

Robles MS, Humphrey SJ, Mann M (2017) Phosphorylation Is a Central Mechanism for Circadian Control of Metabolism and Physiology. *Cell Metab* 25: 118-127

Sanfins A, Plancha CE, Overstrom EW, Albertini DF (2004) Meiotic spindle morphogenesis in in vivo and in vitro matured mouse oocytes: insights into the relationship between nuclear and cytoplasmic quality. *Hum Reprod* 19: 2889-99

Su YQ, Denegre JM, Wigglesworth K, Pendola FL, O'Brien MJ, Eppig JJ (2003) Oocyte-dependent

activation of mitogen-activated protein kinase (ERK1/2) in cumulus cells is required for the maturation of the mouse oocyte-cumulus cell complex. *Dev Biol* 263: 126-38

Sun H, Poudel S, Vanderwall D, Lee DG, Li Y, Peng J (2022) 29-Plex tandem mass tag mass spectrometry enabling accurate quantification by interference correction. *Proteomics* 22: e2100243

Sun H, Sun G, Zhang H, An H, Guo Y, Ge J, Han L, Zhu S, Tang S, Li C, Xu C, Guo X, Wang Q (2023) Proteomic Profiling Reveals the Molecular Control of Oocyte Maturation. *Mol Cell Proteomics* 22: 100481

Swain JE, Ding J, Wu J, Smith GD (2008) Regulation of spindle and chromatin dynamics during early and late stages of oocyte maturation by aurora kinases. *Mol Hum Reprod* 14: 291-9

Tong C, Fan HY, Chen DY, Song XF, Schatten H, Sun QY (2003) Effects of MEK inhibitor U0126 on meiotic progression in mouse oocytes: microtubule organization, asymmetric division and metaphase II arrest. *Cell Res* 13: 375-83

Wang K, Nishida H (2015) REGULATOR: a database of metazoan transcription factors and maternal factors for developmental studies. *BMC Bioinformatics* 16: 114

Wang S, Kou Z, Jing Z, Zhang Y, Guo X, Dong M, Wilmut I, Gao S (2010) Proteome of mouse oocytes at different developmental stages. *Proc Natl Acad Sci U S A* 107: 17639-44

Wang Z, Ma J, Miyoshi C, Li Y, Sato M, Ogawa Y, Lou T, Ma C, Gao X, Lee C, Fujiyama T, Yang X, Zhou S, Hotta-Hirashima N, Klewe-Nebenius D, Ikkyu A, Kakizaki M, Kanno S, Cao L, Takahashi S et al. (2018) Quantitative phosphoproteomic analysis of the molecular substrates of sleep need. *Nature* 558: 435-439

Wei Z, Greaney J, Zhou C, H AH (2018) Cdk1 inactivation induces post-anaphase-onset spindle migration and membrane protrusion required for extreme asymmetry in mouse oocytes. *Nat Commun* 9: 4029

Wu R, Dephoure N, Haas W, Huttlin EL, Zhai B, Sowa ME, Gygi SP (2011) Correct interpretation of comprehensive phosphorylation dynamics requires normalization by protein expression changes. *Mol Cell Proteomics* 10: M111 009654

Zhang H, Ji S, Zhang K, Chen Y, Ming J, Kong F, Wang L, Wang S, Zou Z, Xiong Z, Xu K, Lin Z, Huang B, Liu L, Fan Q, Jin S, Deng H, Xie W (2023) Stable maternal proteins underlie distinct transcriptome, translome, and proteome reprogramming during mouse oocyte-to-embryo transition. *Genome Biol* 24: 166

Zhao J, Yao K, Yu H, Zhang L, Xu Y, Chen L, Sun Z, Zhu Y, Zhang C, Qian Y, Ji S, Pan H, Zhang M, Chen J, Correia C, Weiskittel T, Lin DW, Zhao Y, Chandrasekaran S, Fu X et al. (2021) Metabolic remodelling during early mouse embryo development. *Nat Metab* 3: 1372-1384

Zhou C, Homer HA (2022) The oocyte spindle midzone pauses Cdk1 inactivation during fertilization to enable male pronuclear formation and embryo development. *Cell Rep* 39: 110789

Prof. Qiang Wang
Nanjing Medical University
State Key Laboratory of Reproductive Medicine and Offspring Health
Nanjing, Jiangsu
China

5th Aug 2024

Re: EMBOJ-2024-117657R
The global phosphorylation landscape of mouse oocytes during meiotic maturation

Dear Dr. Wang,

Thank you for submitting your revised manuscript for our consideration. It has now been assessed once more by the original referees 2 and 3, whose comments are copied below. I carefully compared their comments to your point-by-point response, and concluded that there are no further experimental concerns to be addressed, but that several presentational issues would still need to be addressed before publication. In particular, referee 3 retains several reservations, which would need to be answered not only in the response letter, but would also require incorporation into the main manuscript text, e.g. by adding caveats, toning down statements, or better acknowledging previous reports.

I am therefore returning the manuscript to you for a final round of revision, in which I would ask you to also take care of the following editorial points:

- As we are switching from a free-text author contribution statement towards a more formal statement based on Contributor Role Taxonomy (CRediT) terms, please remove the present Author Contribution section and instead specify each author's contribution(s) directly in the Author Information page of our submission system during upload of the final manuscript. See <https://casrai.org/credit/> for more information.
- Please use the attached "Reagents and Tools Table" template for entering the information currently provided as "Dataset EV19". (For detail, see <https://www.embopress.org/page/journal/14693178/authorguide#structuredmethods>)
- Please reorganize the uploaded Source Data into one file or folder per figure, and providing one ZIP file for each main figure. Source Data for EV Figures and/or Appendix Figures (if applicable) should be combined in one ZIP for EV and a separate ZIP for Appendix.
- Finally, please provide suggestions for a short 'blurb' text prefacing and summing up the study in two sentences (max. 250 characters), followed by 3-5 one-sentence 'bullet points' with brief factual statements of key results of the paper; they will form the basis of an editor-written 'Synopsis' accompanying the online version of the article. Please also upload a synopsis image, which can be used as a "visual title" for the synopsis section of your paper (maybe based on a condensed/simplified version of Figure 1A?). The image should be in PNG or JPG format with the modest dimensions of EXACTLY 550 pixels wide and 300-600 pixels high.

I would appreciate if you could make these requested changes and upload all finalized files before the end of the week, in order to expedite our final checking and prevent unnecessary delays with acceptance and publication of this Resource Article.

Yours sincerely,

Hartmut Vodermaier

1) Every manuscript requires a Data Availability section (even if only stating that no deposited datasets are included). Primary datasets or computer code produced in the current study have to be deposited in appropriate public repositories prior to

resubmission, and reviewer access details provided in case that public access is not yet allowed. Further information: embopress.org/page/journal/14602075/authorguide#dataavailability

9) To facilitate reproducibility and cross-laboratory adoption of methodologies, please structure the Materials & Methods section as outlined in our guide to authors, including a completed Reagents and Tools Table that can be downloaded from our author guidelines as well (<https://www.embopress.org/page/journal/14602075/authorguide#structuredmethods>).

10) Digital image enhancement is acceptable practice, as long as it accurately represents the original data and conforms to community standards. If a figure has been subjected to significant electronic manipulation, this must be clearly noted in the figure legend and/or the 'Materials and Methods' section. The editors reserve the right to request original versions of figures and the original images that were used to assemble the figure. Finally, we generally encourage uploading of numerical as well as gel/blot image source data; for details see: embopress.org/page/journal/14602075/authorguide#sourcedata

At EMBO Press, we ask authors to provide source data for the main manuscript figures. Our source data coordinator will contact you to discuss which figure panels we would need source data for and will also provide you with helpful tips on how to upload and organize the files.

In the interest of ensuring the conceptual advance provided by the work, we recommend submitting a revision within 3 months (3rd Nov 2024). Please discuss the revision progress ahead of this time with the editor if you require more time to complete the revisions. Use the link below to submit your revision:

Link Not Available

Referee #2:

The authors have addressed most of my concerns. However, I still find the combination of protein-corrected and non-corrected

phosphorylation site values less than ideal. I believe they should be investigated independently, or the low number of non-corrected sites should be removed from the analysis, especially because the authors state that the sites have no effect on the analysis.

Referee #3:

In this revised version of the manuscript, the authors have introduced several changes in response to the reviewers' comments. In their rebuttal, the authors also presented arguments in response to some of the reviewers' points. However, statements were sometimes difficult to understand. They did not perform additional experiments to strengthen their conclusions and to address the reviewers' concerns and did not modify the text of the revised manuscript. Often, the authors' arguments are difficult to accept, since are based on speculative and questionable opinions. As a result, concerns remain that some of the authors' statements and conclusions are not correct.

Examples:

Comment 3. In this comment, Reviewer 3 expresses concern that the statement that phosphoproteome changes are much larger than changes in the proteome may not be correct. The authors engage in a long explanation of the differences in proteomics and translomics sensitivity. However, they themselves admit that the TMT quantification suffers from ratio compression, which opens the possibility that many of the protein levels changes may be underestimated or missed. As a result, the detected 929 changes in protein levels they report may be a fraction of what is happening in the oocytes as suggested by the analysis of other datasets. Based on this reasoning, the authors should be very cautious with their conclusions that changes in phosphorylation exceed changes in protein levels, and they should alert the readers of the limitations of the technique used.

Thus, the authors sentence introduced in the text that "The overall changes in phosphorylation status appeared to be of a larger magnitude than the changes at the protein level" does not provide the reader sufficient information to evaluate the data.

Comment 4. Here the reviewer alerted the authors that YBX2/MSY2 phosphorylation has been reported before (PMID18606161) and its function explored with mutagenesis. The authors acknowledge this previous publication in the responses to reviewers. However, they do not modify the text and state in the rebuttal that "the phosphosites of YBX2 identified in their studies are not included in the novel phosphosites discovered here". The statement is misleading because, in their dataset, the authors include changes in phosphorylation of residue T67 and T78 that have been investigated in the previously published study. The authors should acknowledge this previous publication and clearly state that, in addition to previously identified phosphosites, they observe additional sites of phosphorylation that may or may not have distinct functions from those already published.

Comments 5. The authors' argument that they use WT BTG4 as a control does not alleviate the concerns raised by multiple reviewers. In Fig. S3C, the mutant BTG4 decreases the levels of the endogenous BTG4 more than the WT BTG4, providing an alternative explanation of the data. In addition, mutagenesis only partially disrupts the interaction with PABPN1. Therefore, the authors' arguments to not provide additional controls and not to modify the text are weak.

Comment 7. The modification of the text on line 302 does not include reference to extensive studies investigating the phosphorylation of RPS6 by S6 kinases and the effects on translation. Thus, their statements that phosphorylation of ribosomal proteins needs investigation is misleading.

Comment 8. The authors' response to this comment is mostly speculative and none of the suggestions they make is followed by experiments to prove their point. Their statement that "due to potential off-target effects of these compounds, a cleaner way such as knockdown or knockout targeting the specific kinase is crucial to clarify this question" does not help the reader in evaluating the data and the conclusions. Their suggestion does not take into consideration the numerous reports that have explored the function of casein kinases in the oocyte. For example, knockdown (PMID 23690993, 25927854) or neutralizing antibodies against CSNK1A (PMID: 9365278) have been used to probe the function of this protein during oocyte maturation and embryo development, but the results have been conflicting. The authors should refer to these previous studies and discuss how these proposed strategies would strengthen or not their finding. They should also discuss how their statement that CSNK1A activity decreases during oocyte maturation (line 397) can be reconciled with the effects of the inhibitors, which also decrease kinase activity.

Minor points:

Fig.4 is mislabeled. MII not MI.

Line 130 should be modified.

Referee #2:

The authors have addressed most of my concerns. However, I still find the combination of protein-corrected and non-corrected phosphorylation site values less than ideal. I believe they should be investigated independently, or the low number of non-corrected sites should be removed from the analysis, especially because the authors state that the sites have no effect on the analysis.

Response: Thank you for the comment. The following reason promoted us to include the non-corrected phosphorylation sites in this study. The dynamic changes in the non-corrected phosphosites (515 out of 8,090) during oocyte maturation (from GV to MII) may not be absolutely accurate due to the corresponding proteins undetectable in the proteome. However, the phosphorylation of these sites is highly likely to occur in mouse oocytes (localization probability > 0.75). The present study aims to provide a valuable and comprehensive resource. Considering that some researchers in the field may be just interested in the phosphosite itself, instead of the dynamics, here we decided to keep the non-corrected phosphosites.

Referee #3:

In this revised version of the manuscript, the authors have introduced several changes in response to the reviewers' comments. In their rebuttal, the authors also presented arguments in response to some of the reviewers' points. However, statements were sometimes difficult to understand. They did not perform additional experiments to strengthen their conclusions and to address the reviewers' concerns and did not modify the text of the revised manuscript. Often, the authors' arguments are difficult to accept, since are based on speculative and questionable opinions. As a result, concerns remain that some of the authors' statements and conclusions are not correct.

Examples:

1. *Comment 3. In this comment, Reviewer 3 expresses concern that the statement that phosphoproteome changes are much larger than changes in the proteome may not be correct. The authors engage in a long explanation of the differences in proteomics and translomics sensitivity. However, they themselves admit that the TMT quantification suffers from ratio compression, which opens the possibility that many of the protein levels changes may be underestimated or missed. As a result, the detected 929 changes in protein levels they report may be a fraction of what is happening in the oocytes as suggested by the analysis of other datasets. Based on this reasoning, the authors*

should be very cautious with their conclusions that changes in phosphorylation exceed changes in protein levels, and they should alert the readers of the limitations of the technique used. Thus, the authors sentence introduced in the text that " The overall changes in phosphorylation status appeared to be of a larger magnitude than the changes at the protein level" does not provide the reader sufficient information to evaluate the data.

Response: Thank you for the suggestion.

(1) The explanation of the differences in proteomics and translomics sensitivity aimed to answer another question raised by Reviewer 3.

(2) We agree with the reviewer's comments on the comparison between phosphoproteome and proteome changes. As the reviewer suggested, we have modified the text in the revised manuscript as follows:

(a) The sentence "*The overall changes in phosphorylation status appeared to be of a larger magnitude than the changes at the protein level*" has been changed to "*The broader distribution of the magnitude of changes to the phosphosites was observed in comparison to the proteome*" (line 152-153)

(b) A brief description of ratio compression has been incorporated in the Discussion section (line 552-556). "*On the other hand, our data may not reflect the exact fold changes in proteome and phosphoproteome due to the ratio compression inevitably caused by Tandem Mass Tag (TMT) quantification (Karp, Huber et al., 2010). Technical improvements (i.e., the implementation of a causal model of ion interference) may help resolve this issue (Madern, Reiter et al., 2024).*"

2. Comment 4. Here the reviewer alerted the authors that YBX2/MSY2 phosphorylation has been reported before (PMID18606161) and its function explored with mutagenesis. The authors acknowledge this previous publication in the responses to reviewers. However, they do not modify the text and state in the rebuttal that "the phosphosites of YBX2 identified in their studies are not included in the novel phosphosites discovered here". The statement is misleading because, in their dataset, the authors include changes in phosphorylation of residue T67 and T78 that have been investigated in the previously published study. The authors should acknowledge this previous publication and clearly state that, in addition to previously identified phosphosites, they observe additional sites of phosphorylation that may or may not have distinct functions from those already published.

Response: As the reviewer suggested, we have included a statement in the revised manuscript (line 209-212). *"In addition to the previously identified phosphosites (i.e., YBX2-T67/78) (Medvedev, Yang et al., 2008), substantial novel sites of phosphorylation were detected (i.e., YBX2-S203/206/351, CUL4B-S155/147, CSNK1A1-T321 and SRPK1-S37; Fig. 3L-3O and Appendix Fig. S2). "*

3. *Comments 5. The authors' argument that they use WT BTG4 as a control does not alleviate the concerns raised by multiple reviewers. In Fig. S3C, the mutant BTG4 decreases the levels of the endogenous BTG4 more than the WT BTG4, providing an alternative explanation of the data. In addition, mutagenesis only partially disrupts the interaction with PABPN1. Therefore, the authors' arguments to not provide additional controls and not to modify the text are weak.*

Response: Thank you for the comments. As we responded before, both exogenous myc-BTG4 wt and myc-BTG4 tm potentially reduce the accumulation of endogenous BTG4, and perhaps to varying extents. The potential reasons need to be further clarified in the future. However, as shown in Fig. 4L, exogenous BTG4^{wt} oocytes exhibited a transcriptome profile very similar to that of the control MII oocytes. This indicates that, in current overexpression system, the slight reduction of endogenous BTG4 has little effect on the overall transcript abundance. In contrast, expression of exogenous BTG4^{mut} markedly suppressed the mRNA degradation during oocyte maturation, implying the function of phosphorylation of these sites. In the revised manuscript, we have included the relevant information for the readers to evaluate the data and conclusion (line 477-481). *"Three previously unknown phosphosites (T145/S146/S147) were identified in BTG4, and phosphomutant compromises its ability for mRNA degradation in oocytes (Fig. 4). Nevertheless, expression of exogenous BTG4^{wt} and BTG4^{mut} seemed to reduce the accumulation of endogenous BTG4 (Appendix Fig. S3C), which needs to be clarified in the future."*

In addition, as the reviewer indicated, mutagenesis only partially disrupts the interaction with PABPN1, indicating that other factors or mechanisms may be involved in this process. We have toned down our conclusion to *"Co-immunoprecipitation assay further corroborated that phospho-deficient mutant of BTG4 PARTIALLY disrupts its interaction with PABPN1"* (line 244).

4. *Comment 7. The modification of the text on line 302 does not include reference to*

extensive studies investigating the phosphorylation of RPS6 by S6 kinases and the effects on translation. Thus, their statements that phosphorylation of ribosomal proteins needs investigation is misleading.

Response: Thank you for pointing this out. We have rewritten this sentence according to the reviewer's suggestion in the new version of the manuscript (line 300-304). *"The phosphorylation of ribosome protein RPS6 and S6 kinase activity have been extensively examined in oocytes (Adhikari, Flohr et al., 2009, Adhikari, Zheng et al., 2010, Reddy, Zheng et al., 2010). However, whether and how the phosphorylation of other ribosomal proteins functions during oocyte maturation remains to be explored."*

5. *Comment 8. The authors' response to this comment is mostly speculative and none of the suggestions they make is followed by experiments to prove their point. Their statement that "due to potential off-target effects of these compounds, a cleaner way such as knockdown or knockout targeting the specific kinase is crucial to clarify this question" does not help the reader in evaluating the data and the conclusions. Their suggestion does not take into consideration the numerous reports that have explored the function of casein kinases in the oocyte. For example, knockdown (PMID 23690993, 25927854) or neutralizing antibodies against CSNK1A (PMID: 9365278) have been used to probe the function of this protein during oocyte maturation and embryo development, but the results have been conflicting. The authors should refer to these previous studies and discuss how these proposed strategies would strengthen or not their finding. They should also discuss how their statement that CSNK1A activity decreases during oocyte maturation (line 397) can be reconciled with the effects of the inhibitors, which also decrease kinase activity.*

Response: Thank you for the constructive comments. According to the reviewer's suggestion, a discussion on inhibitor treatment has been included in the revised manuscript (line 557-565). *"In addition, various inhibitors were utilized to evaluate the effects of kinase activity on oocyte maturation in current study. We observed the significantly reduced Pb1 extrusion in oocytes treated with CSNK1A1 inhibitor (CSNK1-IN-1) (Fig. EV4B), consistent with a previous report (Wang, Lu et al., 2013). Nevertheless, knockdown via RNAi or microinjection of CKI α antibodies did not block the maturation progression of mouse oocytes (Gross, Simerly et al., 1997, Qi, Wang et al., 2015). Such a discrepancy may arise from the differential effects of different approaches on enzymatic activity or abundance. Knockout targeting the specific kinase*

is helpful in clarifying its function.”

As shown in Fig. EV5D, the phosphorylation levels of CSNK1A substrates suggested the progressive decline of its activity from GV to MII stage. On one hand, this does not mean the absence of CSNK1A activity in these oocytes. On the other hand, GV oocytes were treated with inhibitor, and CSNK1A activity is high at this stage. Similarly, the activity of PLK1 is high in GV oocytes, and then decreases after meiotic resumption, as indicated by the phosphorylation of its substrates (Fig. EV5C). It has been reported that PLK1 knockout or enzymatic inhibition results in the maturation arrest in mouse oocytes (PMID 32267211 25658810).

Minor points:

Fig.4 is mislabeled. MII not MI.

Response: The mislabeled MI has been corrected to MII in Figure 4A.

Line 130 should be modified.

Response: This sentence has been modified in the revised manuscript as suggested (line 129-130). *“It has been well recognized that the active sites on MAPK1 (T183/Y185), MAPK3 (T203/Y205), and CDK1 (T161) undergo phosphorylation during oocyte meiosis, while the phosphorylation of two inhibitory residues (T14/Y15) on CDK1 is decreased after GVBD.”*

Dear Qiang,

Thank you for addressing the final editorial points. Since Hartmut is currently out of office, I have briefly taken over handling of your manuscript to ensure fast processing of your study. I am now pleased to inform you that your manuscript has been accepted for publication in the EMBO Journal.

Finally, we would like to promote your manuscript among the Chinese readership. Therefore, we would like to invite you to prepare a short summary of the manuscript in Chinese (1500-2000 Chinese characters), which we will promote on the WeChat platform 'BioArt' with more than 610,000 followers.

If you are interested in this opportunity, we recommend covering the article very close to its online publication date. Thus, ideally we would very much appreciate if you could send us a draft within the next 7 working days. Please let us know whether or not you would be interested in contributing such a short summary in Chinese.

I have included below some general guidelines on how to prepare a summary and a link to recent examples for your reference. Please let me know if you have any questions about this.

If you have any questions, please do not hesitate to contact the Editorial Office. Thank you for your contribution to The EMBO Journal, and congratulations on a nice study!

With best wishes,

Ieva

General WeChat Summary Guidelines

1. These summary articles are meant to be targeting general audience so please limit the use of specialized technical terms, acronyms and jargon.
2. A summary usually starts with brief background information of the reported work, which is followed by explaining the findings in some detail, and ends with a short review of the conclusions as well as the implications of the work and future directions for the research.
3. The summary should at least contain one graphical item, such as a scheme or a figure from the paper.
4. Please provide ONE SINGLE document containing all text and graphical materials, ideally as a Word.docx or .doc file. Please DO NOT provide the document as a .pdf file.
5. Please DO NOT publicly release the document before the paper is officially published online.

Summary Examples

EMBO J | 罗招庆/欧阳松应揭示谷酰胺脱氨酶MvcA的去泛素化功能
